# The interferon stimulated gene-encoded protein HELZ2 inhibits human LINE-1 retrotransposition and LINE-1 RNA-mediated type I interferon induction

Ahmad Luqman-Fatah [1,2], Yuzo Watanabe[3], Kazuko Uno[4], Fuyuki Ishikawa[1,2], John V. Moran[5,6] & Tomoichiro Miyoshi [1,2,7] ✉

Some interferon stimulated genes (ISGs) encode proteins that inhibit LINE-1 (L1) retrotransposition. Here, we use immunoprecipitation followed by liquid chromatography-tandem mass spectrometry to identify proteins that associate with the L1 ORF1-encoded protein (ORF1p) in ribonucleoprotein particles. Three ISG proteins that interact with ORF1p inhibit retrotransposition: HECT and RLD domain containing E3 ubiquitin-protein ligase 5 (HERC5); 2′−5′-oligoadenylate synthetase-like (OASL); and helicase with zinc finger 2 (HELZ2). HERC5 destabilizes ORF1p, but does not affect its cellular localization. OASL impairs ORF1p cytoplasmic foci formation. HELZ2 recognizes sequences and/ or structures within the L1 5′UTR to reduce L1 RNA, ORF1p, and ORF1p cytoplasmic foci levels. Overexpression of WT or reverse transcriptase-deficient L1s lead to a modest induction of IFN-α expression, which is abrogated upon HELZ2 overexpression. Notably, IFN-α expression is enhanced upon overexpression of an ORF1p RNA binding mutant, suggesting ORF1p binding might protect L1 RNA from "triggering" IFN-α induction. Thus, ISG proteins can inhibit retrotransposition by different mechanisms.

Sequences derived from Long INterspersed Element-1 (LINE-1 or L1) retrotransposons comprise ~17% of human genomic DNA[1]. The overwhelming majority of L1-derived sequences have been rendered retrotransposition-defective by mutational processes either during or after their integration into the genome[2–4]. However, an average human genome is estimated to contain at least 100 full-length human-specific retrotransposition-competent L1s (RC-L1s)[5–7], with only a small number of human-specific "hot" L1s responsible for the bulk of retrotransposition activity[6,8].

Human RC-L1s are ~6 kb and consist of a 5′ untranslated region (UTR), two open reading frames (ORF1 and ORF2), and a 3′ UTR that ends in a poly(A) tract[4,9,10]. ORF1 encodes a ~40 kDa protein (ORF1p) that has RNA binding and nucleic acid chaperone activities[11–13]. ORF2 encodes a ~150 kDa protein (ORF2p) that has endonuclease (EN) and reverse transcriptase (RT) activities required for canonical L1 retrotransposition[14–17]. RC-L1s mobilize via a "copy-and-paste" mechanism, where an L1 RNA intermediate is reverse transcribed into an L1 cDNA at a new genomic

[1]Department of Gene Mechanisms, Graduate School of Biostudies, Kyoto University, Kyoto 606-8501, Japan. [2]Radiation Biology Center, Graduate School of Biostudies, Kyoto University, Kyoto 606-8501, Japan. [3]Proteomics Facility, Graduate School of Biostudies, Kyoto University, Kyoto 606-8501, Japan. [4]Division of Basic Research, Louis Pasteur Center for Medical Research, Kyoto 606-8225, Japan. [5]Department of Human Genetics, University of Michigan Medical School, Ann Arbor, MI 48109-5618, USA. [6]Department of Internal Medicine, University of Michigan Medical School, Ann Arbor, MI 48109-5618, USA. [7]Laboratory for Retrotransposon Dynamics, RIKEN Center for Integrative Medical Sciences, Yokohama 230-0045, Japan. ✉e-mail: miyoshi.tomoichiro.5e@kyoto-u.ac.jp

integration site by a process termed target-site primed reverse transcription (TPRT)[16,18–20].

L1 retrotransposition begins with transcription of full-length RC-L1 sense strand RNA using an internal RNA polymerase II promoter located within the L1 5′UTR[21–23]. The resultant bicistronic L1 mRNA is exported to the cytoplasm, where its translation leads to the production of ORF1p and ORF2p. ORF1p and ORF2p preferentially associate with their encoding L1 RNA, by a process known as *cis*-preference[24,25], to form a cytoplasmic L1 ribonucleoprotein (RNP) complex that appears necessary, but not sufficient for retrotransposition[26,27]. Components of the L1 RNP gain access to the nucleus by a process that does not strictly require mitotic nuclear envelope breakdown[28], although recent reports suggest that components of the L1 RNP might also gain access to genomic DNA during mitotic nuclear envelope breakdown[29].

Once in the nucleus, ORF2p EN makes a single-strand endonucleolytic nick at a consensus target sequence (e.g., 5′-TTTTT/AA-3′ and related variants of that sequence) in genomic DNA, generating 5′-PO$_4$ and 3′-OH groups[16,17,20,30,31]. Base pairing between the short stretch of thymidine nucleosides in genomic DNA liberated by L1 EN cleavage and the 3′ L1 poly(A) tract is thought to form a primer/template complex[27,32], where the 3′-OH group of genomic DNA serves as a primer to allow ORF2p RT to generate (−) strand L1 cDNA from its associated L1 RNA template[16,17,19,32]. How top strand genomic DNA cleavage and (+) strand L1 cDNA synthesis occurs requires elucidation, but each step likely requires activities contained within ORF2p[4,33–35]. The completion of TPRT results in the integration of an L1 at a new genomic location.

L1 retrotransposition is mutagenic and, on rare occasions, can lead to human genetic diseases[4,36–39]. Besides acting as an insertional mutagen, products generated during the process of L1 retrotransposition (e.g., double-stranded L1 RNAs and single-stranded L1 cDNAs) are hypothesized to trigger a type I interferon (IFN) response that may contribute to inflammation and aging phenotypes[40–46]. However, how L1 expression contributes to the induction of a type I IFN response and whether this process plays a direct role in human diseases require elucidation.

Previous studies revealed that ORF1p, ORF2p, and L1 RNA can localize within cytoplasmic foci that closely associate with stress granule (SG) proteins—dynamic membrane-less cytoplasmic structures that form upon the treatment of cells with certain stressors—although it is unclear what role, if any, cytoplasmic foci play in L1 biology[47–50]. SGs sequester polysomes, host proteins, and cellular RNAs and are proposed to function as regulatory hubs during the cellular stress response[51,52]. Intriguingly, host factors that inhibit L1 retrotransposition (e.g., the zinc-finger antiviral protein [ZAP] or MOV10 RNA helicase) frequently co-localize with L1 cytoplasmic foci[50,53,54].

To further understand the suite of host factors that bind to L1 RNPs, we generated a panel of ORF1p missense mutation and tested them for their ability to: (1) be stably expressed in human cell lines; (2) reduce the formation of cytoplasmic foci; (3) impair the ability to bind L1 RNA; and (4) inhibit L1 retrotransposition. These analyses led to the identification of a triple mutant, R206A/R210A/R222A (a.k.a., M8/RBM), in the ORF1p RNA binding domain[12].

Immunoprecipitation (IP) coupled with liquid chromatography-tandem mass spectrometry (LC-MS/MS) analyses followed by Gene Ontology (GO)[55,56] and Gene Set Enrichment Analysis (GSEA)[57] that compared the proteins associated with WT ORF1p vs. an ORF1p triple mutant that impairs RNA binding (M8/RBM) revealed that a full-length RC-L1 containing a carboxyl-terminal epitope-tagged version of ORF1p (WT ORF1p-FLAG) preferentially associates with proteins encoded by several interferon stimulated genes (ISGs), including HERC5, HELZ2, OASL, DDX60L, and IFIT1. Detailed analyses revealed that HERC5, HELZ2, and OASL overexpression inhibits the retrotransposition of engineered L1s in cultured cells and that each protein appears to act at different steps in the L1 retrotransposition cycle. Finally, we report that HELZ2 preferentially recognizes RNA sequences and/or RNA

structures within the L1 5′UTR to destabilize L1 RNA and that HELZ2 overexpression reduced the ability of engineered L1 RNAs to induce IFN-α expression.

## Results

### Construction of a panel of ORF1p missense mutations

To refine the role of ORF1p domains necessary for L1 retrotransposition and/or cytoplasmic foci formation, we generated a panel of ORF1p alanine missense mutations in a full-length human RC-L1 expression construct that expresses a version of ORF1p containing a FLAG epitope tag at its carboxyl-terminus (Fig. 1a, pJM101/L1.3FLAG)[5,50]. Mutations were generated in the following ORF1p regions: (1) M1: a conserved pair of amino acids (N157A/R159A) important for ORF1p cytoplasmic foci formation and L1 retrotransposition[48]; (2) M2: a pair of amino acids predicted to play a role in ORF1p trimerization[58] (R117A/E122A); (3) M3 and M4: amino acids proposed to mediate the coordination of chloride ions in the coiled-coil domain to stabilize ORF1p homotrimer formation[12] (N142A and R135A, respectively); (4) M5: a putative ORF1p protein–protein interaction surface that may interact with host factors through its acidic patch[12] (E116A/D123A); (5) M6-M9: amino acids required for ORF1p RNA binding activity[12,26,49] (K137A/K140A, R235A, R206A/R210A/R211A, and R261A, respectively); and (6) M10: an amino acid thought to decrease nucleic acid chaperone activity[49] (Y282A). The relative position of each mutation in the ORF1p crystal structure[12] and the putative functions of the WT ORF1p amino acids are shown in Supplementary Figs. 1 and 2a.

### ORF1p RNA binding is critical for ORF1p cytoplasmic foci formation

Western blot analyses, using an antibody that recognizes the ORF1p FLAG epitope tag, revealed that each of the ORF1p mutant constructs could be expressed in human U-2 OS osteosarcoma, HeLa-JVM cervical cancer, and HEK293T embryonic kidney cell lines (Fig. 1b and Supplementary Fig. 2b). We observed a severe reduction in the steady state level of ORF1p in the M1 mutant, as well as an alteration in the electrophoretic mobility of ORF1p in the M5 mutant, when compared to the WT ORF1p-FLAG control, in each cell line (Supplementary Fig. 2b). The steady state levels of ORF1p in the M9 mutant were reduced in each cell line and the steady state level of ORF1p in the M10 mutant was more mildly reduced in U-2 OS, but not HEK293T and HeLa-JVM cells, when compared to the WT ORF1p-FLAG control. These results are in general agreement with a previous ORF1p alanine scanning mutational analyses[59]. Similar results were obtained in Western blots using an anti-ORF1p antibody, which can also detect endogenous ORF1p (Supplementary Fig. 2b).

We next assayed whether the ORF1p mutations affected L1 retrotransposition efficiency. Briefly, each of the full-length WT pJM101/L1.3FLAG and mutant ORF1p derivative constructs (mutants M1 to M10) contain an *mneoI* retrotransposition indicator cassette within their 3′UTR, ensuring the G418-resistant foci will only arise upon the completion of a single round of retrotransposition[17]. The L1 retrotransposition efficiency was calculated by counting the resultant number of G418-resistant foci, which was normalized to the transfection efficiency, upon completion of the assays[17,60,61] (Fig. 1c, d; see "Methods").

The M1, M2, M5, M8, and M9 mutants exhibited a severe reduction in L1 retrotransposition efficiencies when compared to the positive control (i.e., retrotransposition decreased by >90% of the level of pJM101/L1.3FLAG). By comparison, the M6, M7, and M10 mutants only exhibited a -60 to 70% decrease in L1 retrotransposition efficiency, whereas the M3 and M4 mutants had no discernable effect on L1 retrotransposition efficiency, when compared to the pJM101/L1.3FLAG positive control (Supplementary Fig. 2c). A construct harboring a missense mutation within the ORF2p reverse transcriptase domain (D702A) served as a negative control. The above data suggest that the putative trimerization,

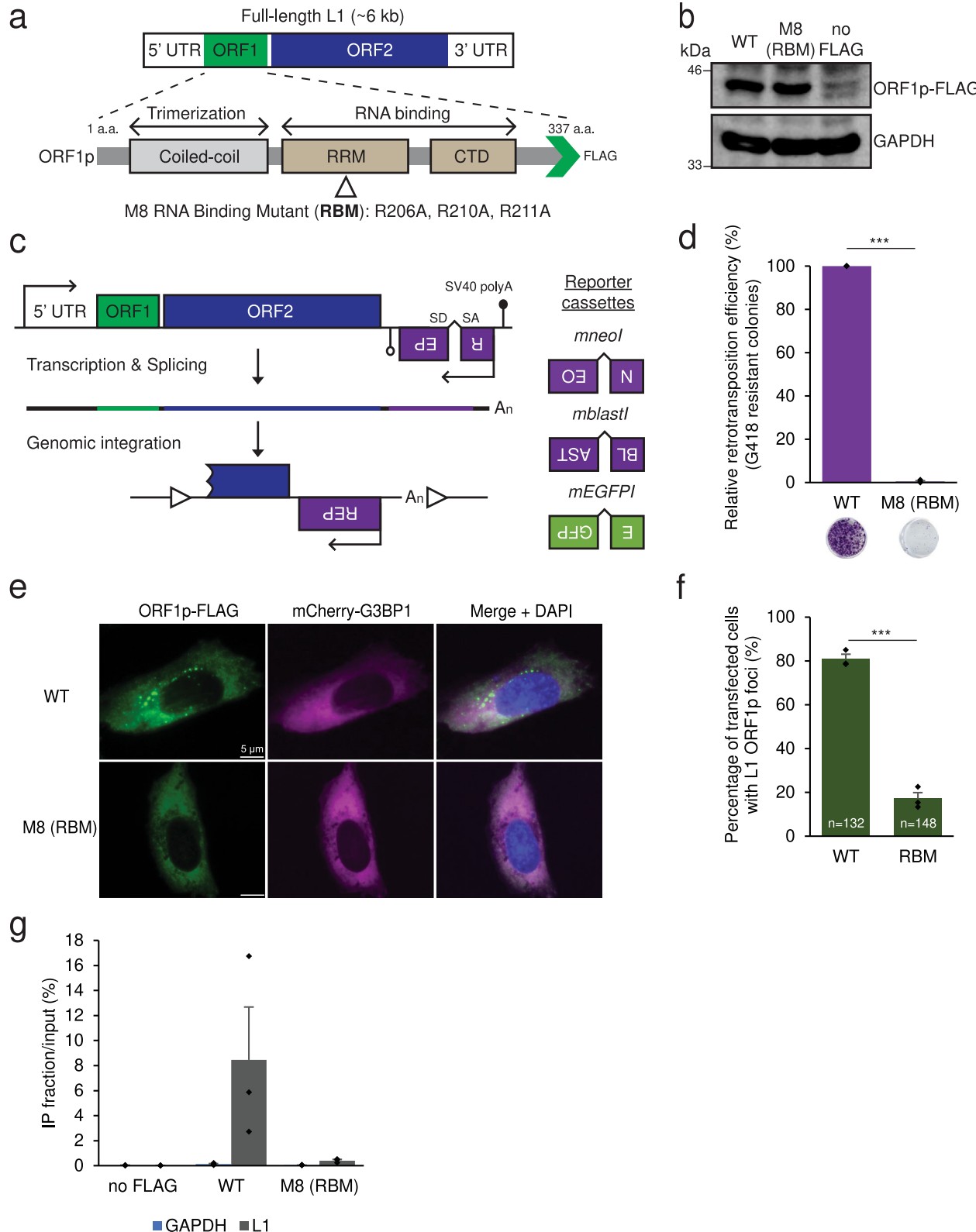

RNA binding, nucleic acid chaperone, and ORF1p protein-binding domains are important for L1 retrotransposition[11,12,17,25,49]. Because the M3 and M4 mutants did not show a reduction in L1 retrotransposition efficiency, these data suggest that single missense mutations in the putative chloride-ion coordinating sites (R135A or N142A) are not sufficient to destabilize ORF1p trimerization when compared to either the M2 mutant or the G132I/R135I/N142I triple mutant used in a previous study[12].

We next focused our analyses on the M2, M5, and M8 mutants because their respective versions of ORF1p are stably expressed in HeLa-JVM cells despite severely reducing L1 retrotransposition efficiency. To determine whether the M2, M5, and M8 mutant ORF1p proteins localize to cytoplasmic foci and associate with stress granules, we established a U-2 OS cell line that expresses a doxycycline-inducible stress granule protein, G3BP1, which is a widely used stress granule marker, that is tagged at its amino terminus with an mCherry

**Fig. 1 | Identification of an ORF1p RNA binding mutant critical for L1 retrotransposition and ORF1p cytoplasmic foci formation. a** Schematic of a full-length RC-L1 (L1.3: Genbank Accession #L19088). ORF1p functional domains are noted below the schematic and include the coiled-coil domain, the RNA recognition motif (RRM), and carboxyl-terminal domain (CTD). Green arrowhead, position of the in-frame FLAG epitope tag. Open triangle, relative position of a triple mutant (R206A/R210A/R211A) in the RRM domain. **b** WT ORF1p and the ORF1p-FLAG R206A/R210A/R211A mutant are stably expressed in HeLa-JVM cells. Western blot with an anti-FLAG antibody. A construct lacking the FLAG epitope tag (pJM101/L1.3 [no FLAG]) served as a negative control. GAPDH served as a sample processing control. **c** Schematics of the retrotransposition indicator cassettes used in this study. A retrotransposition indicator cassette (*REP*) was inserted into the 3′UTR of an L1 in the opposite orientation relative to sense strand L1 transcription. The *REP* gene contains its own promoter (upside down arrow) and polyadenylation signal (open lollipop). The *REP* gene is interrupted by intron in the same orientation relative to sense strand L1 transcription. This arrangement ensures that *REP* expression only will occur if the sense strand L1 transcript is spliced and successfully integrated into genomic DNA by retrotransposition (bottom schematic, open triangles, target site duplications that typically are generated upon L1 retrotransposition). Three retrotransposition indicator cassettes are shown at the right of the figure: *mneoI*, which confers resistance to G418; *mblastI*, which confers resistance to blasticidin; and *mEGFPI*, which leads to enhanced green fluorescent protein (EGFP) expression. **d** Results of a representative mneoI-based retrotransposition assay. HeLa-JVM cells were co-transfected with phrGFP-C (transfection control) and either pJM101/L1.3FLAG (WT) or pALAF008 (M8 [RBM]). *X*-axis, L1 construct names, and representative retrotransposition assay results. *Y*-axis,

relative retrotransposition efficiency; the number of G418 resistant (retrotransposition-positive) foci was normalized to the transfection efficiency (i.e., the percentage of hrGFP-positive cells). Pairwise comparison relative to the WT control: $p = 2.1 \times 10^{-12}$***. **e** The ORF1p-FLAG R206A/R210A/R211A mutant (M8 [RBM]) reduces the number of ORF1p cytoplasmic foci. Representative immunofluorescence microscopy images of U-2 OS cells expressing either WT ORF1p-FLAG (pJM101/L1.3FLAG) or ORF1p-FLAG R206A/R210A/R211A mutant (pALAF008 [M8 (RBM)]). The U-2 OS cells also expressed a doxycycline-inducible (Tet-On) mCherry-G3BP1 protein. White scale bars, 5 μm. **f** Quantification of immunofluorescence assays in U-2 OS cells. *X*-axis, L1 construct names. *Y*-axis, percentage of transfected cells containing ORF1p cytoplasmic foci. The number (n) inside the green bars indicates the number of individual cells counted in the assay. Pairwise comparisons relative to the WT control: $p = 7.5 \times 10^{-11}$***. **g** RNA-immunoprecipitation (RNA-IP) reveals an L1 RNA binding defect in the ORF1p-FLAG R206A/R210A/R211A mutant (M8 [RBM]). HeLa-JVM cells were transfected with either pJM101/L1.3 (no FLAG), WT ORF1p-FLAG (pJM101/L1.3FLAG), or the ORF1p-FLAG R206A/R210A/R211A mutant (pALAF008 [M8 (RBM)]). An anti-FLAG antibody was used to immunoprecipitate ORF1p-FLAG; reverse transcription-quantitative PCR (RT-qPCR) using a primer set (L1 [SV40]) that amplifies RNAs derived from the transfected L1 plasmid was used to quantify L1 RNA. *X*-axis, constructs name. *Y*-axis, the enrichment of L1 RNA levels between the IP and input fractions. Blue rectangles, relative levels of control GAPDH RNA (primer set: GAPDH). Gray rectangles, relative levels of L1 RNA. In panels (**d**), (**f**), and (**g**), values represent the mean ± the standard error of the mean (SEM) of three independent biological replicates. The *p*-values were calculated using a one-way ANOVA followed by Bonferroni–Holm post-hoc tests; *** $p < 0.001$.

---

fluorescent protein (mCherry-G3BP1)[62]. A previous study revealed that U-2 OS cells allow the ready detection of L1 ORF1p cytoplasmic foci[49] and G3BP1-containing stress granules (Supplementary Fig. 3a). The U-2 OS cells then were transfected with either the WT (pJM101/L1.3FLAG), M2, M5, or M8 mutant ORF1p derivatives and ORF1p-FLAG was visualized ~48 h post-transfection using an anti-FLAG primary antibody and Alexa Fluor 488-conjugated anti-mouse IgG secondary antibody (see Methods). The M2 and M5 mutants were able to form ORF1p cytoplasmic foci at comparable numbers and intensities relative to the WT ORF1p control (Supplementary Fig. 3b); however, we did not observe the formation of mCherry-G3BP1 foci. Thus, we next treated the U-2 OS cells with sodium arsenite, which strongly induces stress granule formation[48]. Sodium arsenite treatment resulted in both an increase in size of the ORF1p cytoplasmic foci and co-localization of ORF1p with mCherry-G3BP1 foci (Supplementary Fig. 3c). By comparison, the M8 ORF1p RNA binding mutant exhibited a severe reduction in the percentage of cells containing ORF1p cytoplasmic foci (~15% of cells) when compared to U-2 OS cells expressing either the WT, M2, or M5 constructs (~80% of cells) even though the M8 mutant was stably expressed in HeLa-JVM, U-2 OS, and HEK293T cells (Figs. 1b, e, and f; Supplementary Figs. 2b, 3b, c, and d). RNA-immunoprecipitation (RNA-IP) experiments confirmed that the M8 mutant was impaired for its ability to bind L1 and other cellular RNAs[63] when compared to WT ORF1p, suggesting the M8 mutant exhibited a general loss in its ability to bind RNA (Fig. 1g, Supplementary Fig. 3e, and see below), which is consistent with the previous study[12].

To confirm the M8 ORF1p protein exhibited reduced RNA binding, we transfected the pJM101/L1.3FLAG (ORF1p-FLAG) or pALAF008_L1.3FLAG_M8 (M8/RBM-FLAG) expression constructs into HeLa-JVM cells and immunoprecipitated (IP) the resultant ORF1p complexes using an anti-FLAG antibody (Fig. 2a). Control western blot experiments revealed a similar level of WT and M8/RBM ORF1p-FLAG in whole cell extracts and immunoprecipitates from the HeLa-JVM whole cell extracts cells, but not in a negative control transfected with an L1 expression vector lacking the FLAG epitope tag (Fig. 2b). Moreover, the Poly(A) Binding Protein Cytoplasmic 1 (PABPC1) was robustly detected in IP reactions conducted with cell extracts derived from WT ORF1p-FLAG L1 transfected cells, but was severely reduced in IP reactions conducted with cell extracts derived from M8/RBM ORF1p-FLAG

L1 transfected cells (Fig. 2b), which is consistent with previous studies that found the association between ORF1p and PABPC1 requires RNA[50,64]. Thus, the above data suggest that the M2, M5, and M8 mutants each produce similar steady state levels of ORF1p and reduce L1 retrotransposition efficiencies. However, cytoplasmic foci formation depends on the ability of ORF1p to bind RNA, which is reduced in the M8 mutant. Given these data, we focused our subsequent studies on the WT ORF1p-FLAG and M8/RBM-FLAG proteins (herein called the RNA Binding Mutant [RBM]).

**Immune-related proteins associate with the WT ORF1p complex**
To identify cellular proteins that differentially interact with the WT ORF1p-FLAG and M8/RBM-FLAG protein complexes, we conducted immunoprecipitation coupled with liquid chromatography-tandem mass spectrometry (IP/LC-MS/MS), followed by label-free quantification (LFQ) analyses (Fig. 2c). We used the Database for Annotation, Visualization and Integrated Discovery (DAVID)[55,56] to conduct gene ontology (GO) analyses using proteins that have >0.5 log₂ abundance ratio in WT ORF1p-FLAG vs. M8/RBM-FLAG mutant IP/LC-MS/MS experiments (see Source data). These analyses revealed an enrichment of viral-related GO terms, including "host-virus interaction," "innate immunity," and "antiviral defense" (Fig. 2d and Supplementary Data 1), associated with the WT ORF1p-FLAG vs. M8/RBM-FLAG protein complexes, suggesting a cohort of antiviral proteins preferentially associates with WT L1 RNPs. We also observed an enrichment in the following GO terms: "nonsense-mediated mRNA decay" and "RNA-mediated gene silencing." Proteins within these pathways, such as UPF1[65] and let-7 miRNA[66], respectively, previously were implicated in the regulation of L1 retrotransposition.

We next performed a preranked Gene Set Enrichment Analysis (GSEA) using the log₂ abundance ratio of WT ORF1p-FLAG vs. M8/RBM-FLAG IP/LC-MS/MS protein hits to determine if there was an enrichment of hallmark gene set signatures in the Molecular Signatures Database (MsigDB) (see "Methods"). These analyses identified two interferon-related gene sets—the interferon alpha and interferon gamma responses—among the top six most significantly enriched gene sets (Fig. 2e and Supplementary Data 2, see "Methods").

Because the overexpression of engineered L1s previously was reported to modestly induce type I IFN response[42,45,46,67], we next tested

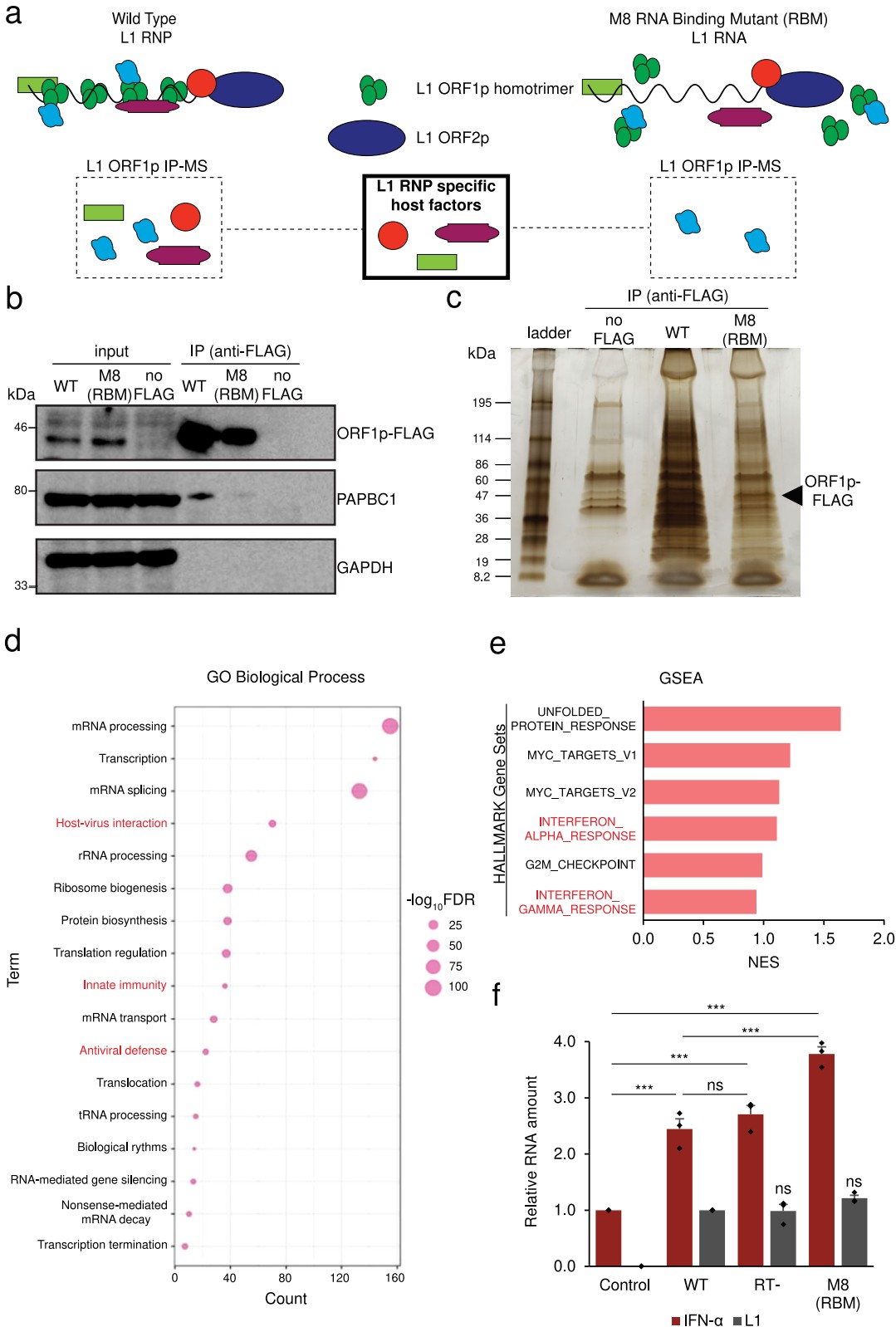

whether there was a difference in IFN-α induction in HEK293T cells transfected with WT and mutant L1 constructs. Notably, HEK293T cells previously were reported to have a low amount of cyclic GMP–AMP synthase (cGAS, a DNA sensor), which can prevent a strong innate immune response by plasmid-based transfections and immunogenic DNAs[46].

HEK293T cells were transfected with either pJM101/L1.3FLAG (WT ORF1p-FLAG), pJM105/L1.3 (a reverse transcriptase-deficient [RT-]

mutant lacking an epitope tag), or pALAF008_L1.3FLAG_M8 (M8/RBM-FLAG). Expression of the WT ORF1p-FLAG or RT-deficient mutant construct each led to a modest induction (~2.5-fold increase) of IFN-α transcription (Fig. 2f). By comparison, M8/RBM-FLAG expression induced a modest, but more significant ~4-fold increase in IFN-α transcription, when compared to a mock control (Fig. 2f). Controls revealed the L1 RNA levels of the RT-deficient and M8/RBM-FLAG mutants were similar to the WT L1 using a primer set that amplified the

**Fig. 2 | The proteins encoded by interferon-responsive genes are enriched in WT ORF1p-FLAG, but not ORF1p-FLAG (M8 [RBM]) mutant complexes.**
**a** Experimental rationale for identifying host factors enriched in WT ORF1p-FLAG vs. ORF1p-FLAG (M8 [RBM]) immunoprecipitation reactions. Hypothetical diagrams of the proteins associating with WT and M8 (RBM) mutant RNP particles. Green circles, ORF1p-FLAG. Blue Oval, ORF2p. Red circle, purple squared oval, and green rectangle, host factors that might associate with ORF1p-FLAG and/or L1 RNPs. **b** The ORF1p (M8 [RBM]) mutant does not efficiently interact with Poly(A) Binding Protein Cytoplasmic 1 (PABPC1). HeLa-JVM cells were transfected with either pJM101/L1.3 (no FLAG), pJM101/L1.3FLAG (WT ORF1p-FLAG), or pALAF008 (ORF1p-FLAG [M8 [RBM]] mutant). An anti-FLAG antibody was used to immunoprecipitate ORF1p-FLAG. Western blots detected ORF1p (anti-FLAG), PABPC1 (anti-PABC1), and GAPDH (anti-GAPDH) in the input and IP fractions. GAPDH served as a sample processing control for the input fractions and a negative control in the IP experiments. **c** Separation of proteins associated with the WT and mutant ORF1p-FLAG proteins. The WT and M8 (RBM) mutant ORF1p-FLAG IP complexes were separated by SDS-PAGE using a 4-15% gradient gel and silver staining visualized the proteins. Protein size standards (kDa) are shown at the left of the gel. Black arrowhead, the expected molecular weight of ORF1p-FLAG. **d** Gene Ontology (GO) analysis identifies cellular proteins enriched in IP WT ORF1p-FLAG vs. the mutant ORF1p-FLAG complex. Cellular proteins present in the WT ORF1p and (M8 [RBM])-FLAG mutant IP complexes were identified using LC-MS/MS. Proteins having a >0.5 $\log_2$ abundance ratio at any p-value in the WT ORF1p vs. M8 [RBM]) complexes were subjected to DAVID gene ontology analysis. Listed are the "functional annotation of UniProt Keyword GO biological process" terms. X-axis, protein count, the number of proteins identified by mass spectrometry that are included in each respective GO term. Y-axis, GO term. Circle size, $-\log_{10}$FDR. Larger circles indicate higher

confidence based on the FDR for each GO term. Red lettering, viral related GO terms. **e** GSEA preranked analysis identifies interferon-related gene sets enriched upon WT ORF1p-FLAG immunoprecipitation. Gene Set Enrichment Analysis (GSEA) of $\log_2$ abundance ratio of cellular proteins immunoprecipitated in the WT ORF1p-FLAG vs. (M8 [RBM])-FLAG IP complexes was performed using hallmark gene sets in the Molecular Signatures Database (MSigDB: https://www.gsea-msigdb.org/gsea/msigdb/), followed by Leading Edge Analysis to determine gene set enrichment scores. The top six hallmark gene sets with the highest normalized enrichment score (NES) are sorted in descending values. X-axis, NES. Y-axis, hallmark gene sets. **f** The expression of engineered L1s modestly up-regulates IFN-α expression. HEK293T were transfected with either pCEP4 (an empty vector control), pJM101/L1.3FLAG (WT), pJM105/L1.3 (RT-), or pALAF008 (M8 [RBM]). RT-qPCR was used to quantify IFN-α (primer set: IFN-α, which amplifies IFN-α1 and IFN-α13) and L1 expression (primer set: *mneoI* [Alu or L1]) ~96 h post-transfection. IFN-α and L1 expression levels were normalized using β-actin (*ACTB*) as a control (primer set: Beta-actin). X-axis, name of constructs. Control, pCEP4. Y-axis, relative RNA expression levels normalized to the pCEP4 empty vector control. Red bars, normalized IFN-α expression levels. Gray bars, normalized L1 expression levels. Values from three independent biological replicates ± SEM are depicted in the graph. The p-values were calculated using a one-way ANOVA followed by Bonferroni-Holm post-hoc tests: pairwise comparisons of IFN-α relative to the pCEP4 control, $p = 0.00028^{***}$ (WT); $0.00011^{***}$ (RT-); $3.14 \times 10^{-6***}$ (M8 [RBM]). Pairwise comparisons of IFN-α: WT vs. RT-, $p = 0.21^{ns}$; WT vs. M8 (RBM), $p = 0.00036^{***}$. Pairwise comparisons of L1 relative to WT, $p = 0.87^{ns}$ (RT-), $p = 0.10^{ns}$ (M8 [RBM]); ns: not significant; $^{***} p < 0.001$. For (**d**) and (**e**), the Source data are provided as a Source Data file.

---

*mneoI* retrotransposition reporter cassette, thereby avoiding the amplification of endogenous L1 transcripts (Note: although HEK293T cells contain a neomycin resistant gene, the endogenous neomycin RNA amount is negligible in comparison to the plasmid-based L1 transcripts [see L1 RNA amounts in pCEP4 vs. L1-transfected cells in Fig. 2f]). Because the expression of each construct upregulates IFN-α expression independently of retrotransposition, these data suggest that L1 RNA, but not L1 cDNAs or ssDNA intermediates generated during L1 TPRT, per se, are primarily responsible for the modest induction of type I IFN expression in HEK293T cells.

Finally, we conducted a Bio-Plex assay that allows the simultaneous assessment of 37 different cytokines and chemokines (see Methods). These analyses revealed that the M8/RBM mutant L1, in particular, exhibited a modest, but overall increase in secreted cytokines and chemokines, as well as other IFNs, when compared to a mock control. As a control, we also included polyinosinic:polycytidylic acid (poly[I:C]) in this assay, which is a double-stranded RNA analog known to strongly induce the innate immune response[46]. We found that several cytokines exhibited a comparable level of upregulation in the poly(I:C) and M8/RBM transfected cells (Supplementary Data 3). WT or RT-deficient L1 transfected cells exhibited an increase in the secretion of several cytokines, including IFN-β, IL-27, and MMP-3, when compared to the controls; however, those levels were generally lower than those in the L1 M8/RBM-transfected cells. These results are consistent with the IFN-α RT-qPCR results (Fig. 2f), which demonstrated the M8/RBM L1-transfected cells induced a higher IFN-α expression when compared to either the WT or RT-deficient L1 transfected cells (Supplementary Data 3).

## Proteins produced by Interferon-Stimulated Genes (ISGs) as potential L1 regulators

A number of proteins expressed from interferon stimulated genes (ISGs) have been reported to influence L1 and/or Alu retrotransposition. These proteins include: (1) MOV10, an RNA helicase[53,68,69]; (2) ADAR1, a double-stranded RNA-specific adenosine deaminase[70]; (3) APOBEC3A, 3B, 3C, 3F, and, for Alu, APOBEC3G, paralogs of the apolipoprotein B editing complex enzyme catalytic polypeptide-like 3 containing cytidine deaminase activity[71-78];

(4) TREX1, a three prime repair exonuclease 1[41,79,80]; (5) ZAP, a zinc-finger antiviral protein[50,54]; (6) SAMHD1, a sterile alpha motif (SAM) domain and histidine-aspartate (HD) domain-containing protein 1[81-83]; (7) RNase H2[84,85]; and (8) RNaseL, a protein that is activated by 2′,5′-oligoadenylate (2–5 A) synthetase (OAS) to enzymatically degrade L1 RNA[86]. Thus, we hypothesized that the ISG proteins associated with L1 RNPs may directly regulate L1 retrotransposition and/or L1-mediated IFN-α expression.

To test the above hypothesis, we screened for proteins that have >0.5 $\log_2$ abundance ratio in WT ORF1p-FLAG vs. M8/RBM-FLAG in our IP/LC-MS/MS analyses using the interferome database (www.interferome.org)[87]. We focused our analysis on identified proteins exhibiting a >5-fold change in expression upon the induction by type I, II, and III IFNs (Supplementary Data 4). STRING database (https://string-db.org/)[88] then was used to test for possible associations among the putative interferon-inducible proteins that preferentially associated with WT ORF1p-FLAG[45,46].

STRING analysis identified several interferon-inducible proteins that exhibited associations and most (i.e., ZC3HAV1 [also known as ZAP], APOBEC3B, ADAR, ADARB1, DDX60, HERC5, TRIM25, TRIM56, EIF2AK2, IFIT1, IFIT2, IFIT3, OASL, DHX58, and IFI16), with the exception of HELZ2 and DDX60L, MOV10, LGALS3BP, and PARP12, were annotated as antiviral defense (red) and/or innate immunity (blue) proteins in UniProt (https://www.uniprot.org/) (Fig. 3a, middle dotted box: ISG network that might regulate L1, red circles: FDR, $1.2 \times 10^{-14}$, interaction strength, 1.51; blue circles: FDR, $5.2 \times 10^{-12}$, interaction strength, 1.16; see "Methods"). Importantly, four proteins: MOV10, APOBEC3B, ADAR, and ZAP, previously were reported to inhibit L1 retrotransposition[50,53,54,68-70,73,75]. Thus, we reasoned that other ISG proteins in the identified ISG network also might be involved in the regulation of L1 retrotransposition (Fig. 3a). Five of these unreported proteins (i.e., HELZ2, IFIT1, DDX60L, OASL, and HERC5) were selected for further analyses; their respective p-values and $\log_2$ abundance ratios (WT ORF1p-FLAG vs. M8/RBM-FLAG) are annotated on the volcano plot shown in Fig. 3b.

To confirm that HELZ2, IFIT1, DDX60L, OASL, and HERC5 associated with WT ORF1p-FLAG, we conducted additional co-IP experiments. Briefly, pJM101/L1.3FLAG (WT ORF1p-FLAG) was co-transfected

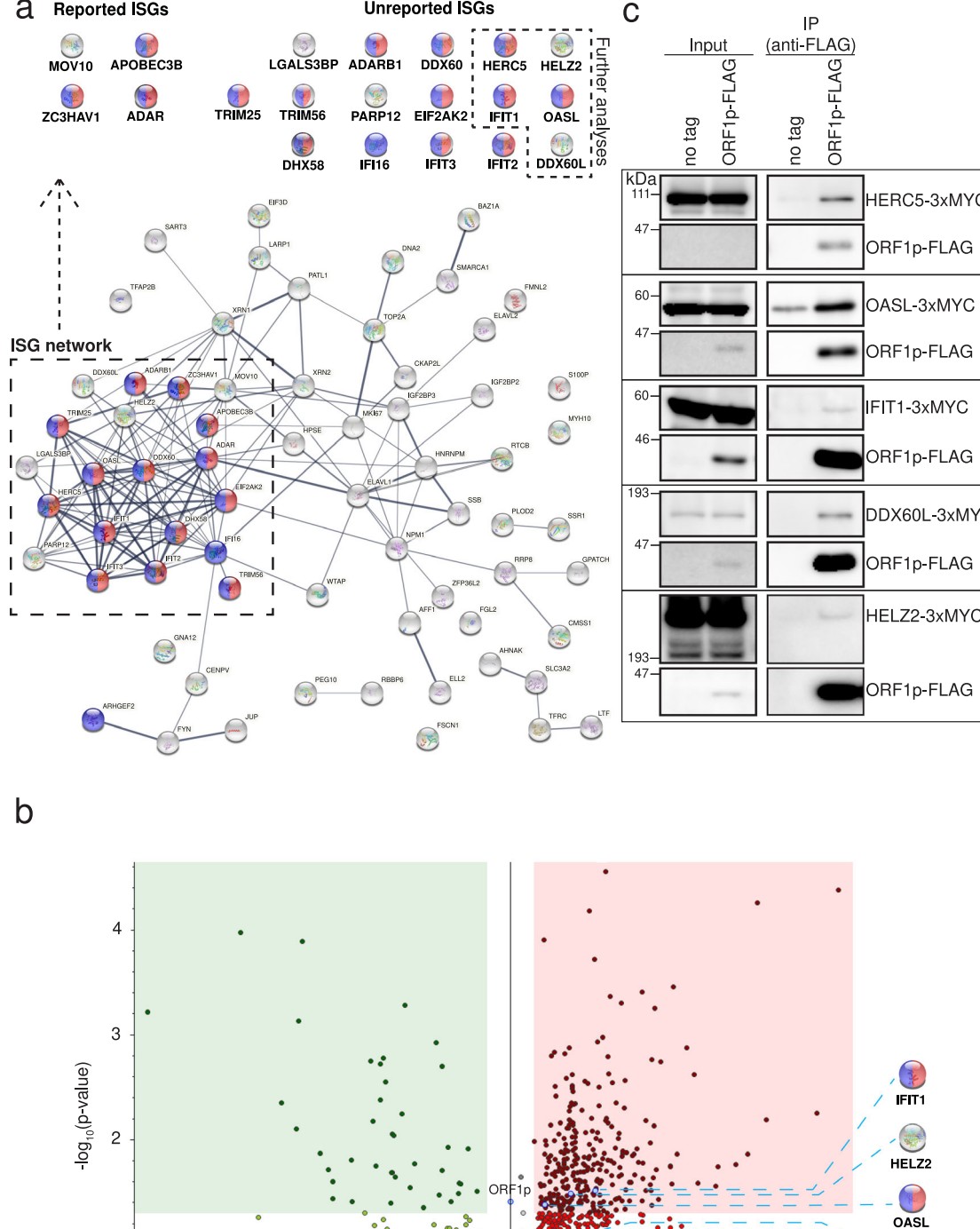

into HEK293T cells with individual ISG protein expression vectors (HELZ2, IFIT1, DDX60L, OASL, and HERC5) containing three copies of a MYC epitope tag at their respective carboxyl-termini. An anti-FLAG primary antibody then was used to immunoprecipitate associated proteins from HEK293T whole cell extracts and an anti-MYC antibody was used to confirm associations between WT ORF1p-FLAG and the candidate ISG proteins. WT ORF1p-FLAG co-immunoprecipitated HERC5, OASL, IFIT1, DDX60L, and HELZ2 (Fig. 3c). Thus, a network of

antiviral ISG proteins may affect L1 RNA, L1 RNP, and/or L1 retrotransposition dynamics.

### The ISG proteins, HELZ2, OASL, and HERC5 inhibit L1 retrotransposition

To determine whether ectopic overexpression of the ISG proteins identified above affect L1 retrotransposition, we co-transfected HeLa-JVM or HEK293T cells with a WT human L1 expression construct

**Fig. 3 | A network of ISGs that potentially affect WT L1 retrotransposition.**
**a** STRING database analysis of WT ORF1p-FLAG associated proteins. Proteins with >0.5 log$_2$ abundance ratios in the WT ORF1p-FLAG vs. M8/RBM-FLAG complexes that exhibited a >5-fold change in expression upon induction by type I, II, and III IFNs were subjected to STRING analysis. Red and blue spheres, proteins annotated in UniProt as antiviral defense and innate immunity proteins, respectively. Thickness of the inter-connecting lines, the strength of association based on the number of independent channels supporting the putative interactions. The black dotted box indicates a group of proteins that closely associate (i.e., a putative ISG network); the majority are annotated as antiviral defense proteins in UniProt. The proteins in the box are listed at the top of the figure (follow dotted arrow) based upon whether they have been reported to regulate L1 retrotransposition (left, Reported ISG), or not (right, Unreported ISG). The top black dotted hexagon shows the proteins used for further analyses. **b** Volcano plot of WT ORF1p-FLAG vs. M8 (RBM) ORF1p-FLAG label-free quantitative mass spectrometry analysis. Data from the WT ORF1p-FLAG vs. M8 (RBM) ORF1p-FLAG IP-LC/MS experiments were analyzed to identify cellular proteins that preferentially associate with ORF1p-FLAG. X-axis, log$_2$ abundance ratios of WT ORF1p-FLAG vs. M8 (RBM) ORF1p-FLAG. Y-axis, −log$_{10}$ p-values of the abundance ratios. The ORF1p amounts obtained in the WT and M8 (RBM), which is indicated in the middle of the plot (0 abundance ratio) were used to normalize protein abundance ratios. Cutoffs of >0.5 log$_2$ abundance ratio and <0.05 p-values are shown as references for the enrichment of proteins in the WT ORF1p-FLAG fraction (red rectangle) or M8 (RBM) ORF1p-FLAG fraction (green rectangle). Blue dotted lines, proteins enriched in WT ORF1p-FLAG complexes (i.e., HELZ2, HERC5, DDX60L, IFIT1, and OASL). The p-values of the abundance ratios were calculated using Tukey Honestly Significant Difference test (post hoc) after an analysis of variance (ANOVA) test. **c** Independent confirmation that ISG proteins interact with WT ORF1p-FLAG. HEK293T cells were co-transfected with either pJM101/L1.3 (no tag) or pJM101/L1.3FLAG (ORF1p-FLAG) and the following individual carboxyl-terminal 3xMYC epitope-tagged ISG expression vectors: pALAF015 (HELZ2), pALAF016 (IFIT1), pALAF021 (DDX60L), pALAF022 (OASL), or pALAF023 (HERC5). The input and anti-FLAG IP reactions were analyzed by western blotting using an anti-FLAG (to detect ORF1p-FLAG) or an anti-MYC (to detect ISG proteins) antibody. For (**a**) and (**c**), the Source data are provided as a Source Data file.

containing either a *mblastI* (pJJ101/L1.3) or *mEGFPI* (cepB-gfp-L1.3) retrotransposition indicator cassette and the carboxy-terminal 3× MYC epitope-tagged HELZ2, IFIT1, DDX60L, OASL, or HERC5 expression vectors. L1 retrotransposition efficiencies then were determined by counting the resultant number of blasticidin-resistant foci or EGFP-positive cells (Fig. 1c and Supplementary Fig. 4a, see "Methods"). A MOV10 expression vector containing a carboxyl-terminal 3x MYC epitope tag served as a positive control. The overexpression of DDX60L and IFIT1 did not significantly inhibit L1 retrotransposition in HeLa-JVM (Fig. 4a) or HEK293T cells (Supplementary Fig. 4b), although we note the expression of DDX60L was barely detected by western blot in either cell line (Fig. 4b and Supplementary Fig. 4c). By comparison, overexpression of HERC5, HELZ2, and OASL reduced retrotransposition by at least 2-fold in the *mblastI*-based L1 retrotransposition assay conducted in HeLa-JVM cells (Fig. 4a) and by ~90% in the *mEGFPI*-based L1 retrotransposition assay conducted in HEK293T cells (Supplementary Fig. 4b).

### Some ISG proteins affect ORF1p and L1 mRNA levels
To further understand how ISG proteins might inhibit L1 retrotransposition, we co-transfected a full-length RC-L1 (pTMF3), and either the HELZ2, IFIT1, DDX60L, OASL, HERC5, or MOV10 expression vectors into HeLa-JVM or HEK293T cells and examined whether the ISG proteins affected ORF1p and/or L1 RNA expression. Western blot analysis revealed a similar data trend in HeLa-JVM and HEK293T cells: the steady state ORF1p levels were significantly decreased by co-expression of HERC5, HELZ2, and MOV10, were modestly reduced by the co-expression of OASL, but were not changed significantly by the co-expression of IFIT1 or DDX60L (Fig. 4b and Supplementary Fig. 4c). RT-qPCR analyses, using a probe set that specifically recognizes the SV40 poly(A) signal of the plasmid-expressed L1 RNA, revealed that HELZ2 co-expression significantly reduced L1 RNA levels in HeLa-JVM cells (Fig. 4c, ~90% reduction of the WT L1 control). By comparison, MOV10 co-expression resulted in a ~70% reduction in L1 RNA when compared to the WT L1 control, which is consistent with previous reports[69,89].

We next tested whether the co-transfection of pJM101/L1.3FLAG (WT ORF1p-FLAG) with the individual ISG protein expression vectors (i.e., HELZ2, HERC5, OASL, and MOV10) affected ORF1p-FLAG cytoplasmic foci formation in HeLa-JVM cells (Fig. 4d, e). Greater than 70% of transfected cells expressing WT ORF1p-FLAG exhibited cytoplasmic foci (Fig. 4e), which is consistent with previous results[49] (see Supplementary Fig. 3d). Co-expression of HERC5 did not dramatically affect ORF1p cytoplasmic foci formation in HeLa-JVM cells (Fig. 4e, ~55% of cells contained ORF1p cytoplasmic foci that associated with HERC5). By comparison, the co-expression of HELZ2, OASL, and MOV10 resulted in a decrease in ORF1p-FLAG cytoplasmic foci (Fig. 4e, ~30%, ~15%,

and ~5% of cells, respectively) and very few of these foci co-associated with the relevant ISG protein (Fig. 4d). In aggregate, these data suggest: (1) HERC5 destabilizes ORF1p, but does not affect its cellular localization; (2) OASL mainly impairs ORF1p cytoplasmic foci formation; and (3) HELZ2 reduces the levels of L1 RNA, ORF1p, and ORF1p cytoplasmic foci formation. Thus, different ISGs appear to affect different steps of the L1 retrotransposition cycle.

### The HELZ2 helicase activity is important for inhibition of L1 retrotransposition
HELZ2 contains two putative helicase domains (helicase 1 and helicase 2) that flank a putative exoribonuclease RNase II/R (RNB) domain (Fig. 5a and Supplementary Fig. 5a). Because proteins containing a RNB domain often possess 3′ to 5′ single-strand exoribonuclease activity[90,91], we aligned the protein sequences of RNB-containing proteins from human, yeast, and *E. coli* to identify evolutionarily conserved aspartic acid residues, which when mutated, are predicted to impair exoribonuclease activity[90–92] (Supplementary Fig. 5b), which led to the creation of a HELZ2 triple mutant (D1346N/D1354N/D1355N, a.k.a. dRNase mutant).

To examine whether the HELZ2 RNB domain has exoribonuclease activity, we purified the WT HELZ2-3xFLAG and dRNase HELZ2-3xFLAG mutant proteins from HEK293T cells (Supplementary Fig. 5c) and performed a ribonuclease assay using a poly(A)$_{30}$ RNA oligonucleotide labeled with IRDye800 at its 5′ end as an RNA substrate. The WT HELZ2-3xFLAG protein, but not the dRNase HELZ2-3xFLAG mutant protein, degraded the single-strand RNA substrate in a 3′ to 5′ direction (Supplementary Fig. 5d). However, the dRNase mutant generally only had minor effects (i.e., less than 2-fold) on L1 retrotransposition efficiency in HeLa-JVM and HEK293T cells when compared to the WT HELZ2 control (Supplementary Fig. 5e, f, see "Discussion").

We next tested whether mutations in the putative HELZ2 helicase domains affect L1 retrotransposition. We mutated conserved amino acids in the Walker A and Walker B boxes thought to be required for ATP binding (WA1 [K550A] in the helicase 1 domain and WA2 [K2180A] in the helicase 2 domain) (Fig. 5a) or ATP hydrolysis (WB1 [E668A] in the helicase 1 domain and WB2 [E2361A] in the helicase 2 domain), respectively[93–95] (Supplementary Fig. 5a). The WA1 mutant was able to inhibit L1 retrotransposition almost as effectively as WT HELZ2 in HEK293T (Fig. 5b), but not HeLa-JVM (Fig. 5c), cells. The WA2 and WA1&2 double mutants were significantly impaired in their ability to inhibit L1 retrotransposition in both HEK293T (Fig. 5b) and HeLa-JVM cells (Fig. 5c). A similar data trend was observed for the Walker B box mutations in HEK293T and HeLa-JVM cells (Supplementary Fig. 5g, h, respectively). In sum, mutations in the helicase 2 (WA2 and WB2) domains generally alleviated the HELZ2-mediated repression of L1 retrotransposition to a greater extent than mutations in the helicase 1

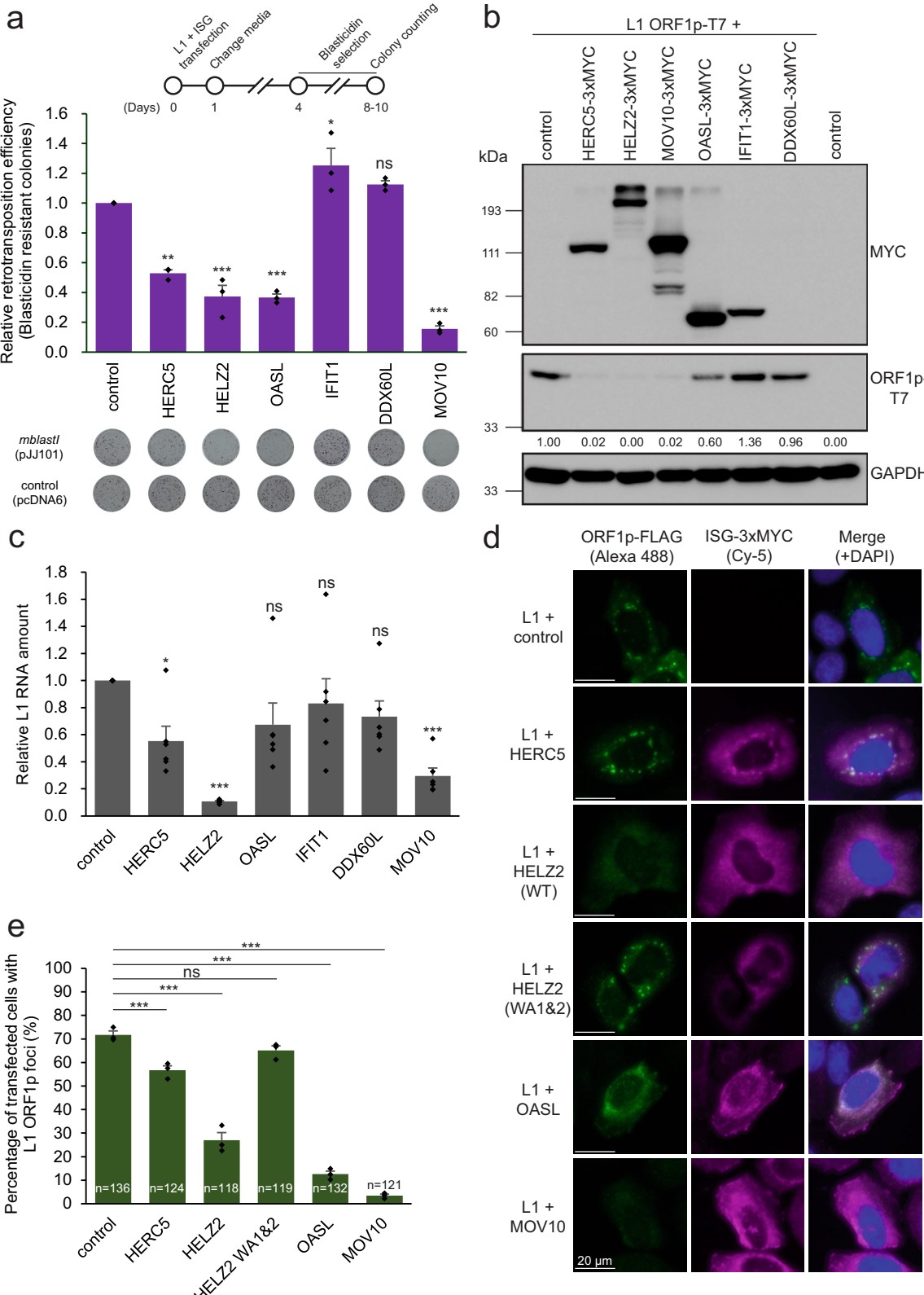

(WA1 and WB1) mutants, indicating the importance of the helicase 2 domain in the inhibition of L1 retrotransposition.

Additional experiments revealed that the WA1 mutant reduced both ORF1p-T7 and L1 RNA levels almost as effectively as WT HELZ2 in HeLa-JVM cells (Fig. 5d), whereas the WA2 and WA1&2 double mutant partially reduced L1 RNA levels when compared to the WT control, but did not dramatically affect the steady state levels of the ORF1p-T7

protein (Fig. 5d). Importantly, we did not observe a noticeable reduction in the steady state levels of the HELZ2 mutant proteins (Fig. 5d, bottom panel), suggesting that the effects on L1 retrotransposition are not due to mutant HELZ2 protein instability (Fig. 5d). Finally, the WA1&2 double mutant did not affect the ability of ORF1p-FLAG to localize to cytoplasmic foci when compared to WT HELZ2 (Fig. 4d, e). A union of the above data suggest that the HELZ2 helicase activity has a

**Fig. 4 | A subset of ISG proteins affect steady state L1 RNA levels, ORF1p cytoplasmic foci formation, and/or L1 retrotransposition. a** Overexpression of HERC5, HELZ2, or OASL inhibit L1 retrotransposition. HeLa-JVM cells were co-transfected with pJJ101/L1.3, which contains the *mblastI* retrotransposition indicator cassette, and either pCMV-3Tag-8-Barr or one of the following carboxyl-terminal 3xMYC epitope-tagged ISG protein expression plasmids: pALAF015 (HELZ2), pALAF016 (IFIT1), pALAF021 (DDX60L), pALAF022 (OASL), pALAF023 (HERC5), or pALAF024 (MOV10) according to the timeline shown at the top of the figure. A blasticidin expression vector (pcDNA6) was co-transfected into cells with either pCMV-3Tag-8-Barr or an individual ISG protein expression plasmid (see plates labeled control [pcDNA6]) to assess cell viability. The retrotransposition efficiencies then were normalized to the respective toxicity control. *X*-axis, name of the control (pCMV-3Tag-8-Barr) or ISG protein expression plasmid. Y-axis, relative retrotransposition efficiency normalized to the pJJ101/L1.3 + pCMV-3Tag-8-Barr co-transfected control. Representative results of the retrotransposition (see plates labeled *mblastI* [pJJ101]) and toxicity (see plates labeled *control* [pcDNA6]) assays are shown below the graph. Pairwise comparisons relative to the pJJ101/L1.3 + pCMV-3Tag-8-Barr control: $p = 8.0 \times 10^{-5**}$ (HERC5); $4.4 \times 10^{-6***}$ (HELZ2); $4.9 \times 10^{-6***}$ (OASL); 0.011* (IFIT1); 0.12 ns (DDX60L); and $1.7 \times 10^{-7***}$ (MOV10). MOV10 served as a positive control in the assay. **b** Expression of the ISG proteins in HeLa-JVM cells. HeLa-JVM cells were co-transfected with pTMF3, which expresses a version of ORF1p containing a T7 gene 10 carboxyl epitope tag (ORF1p-T7), and either a pCMV-3Tag-8-Barr (control) or the individual ISG-expressing plasmids used in panel (**a**). Whole cell extracts were subjected to western blot analysis 48 h post-transfection. ISG proteins were detected using an anti-MYC antibody. ORF1p was detected using an anti-T7 antibody. GAPDH served as a sample processing control. The relative band intensities of ORF1p-T7 are indicated under the ORF1p-T7 blot;

they were calculated using ImageJ software and normalized to the respective GAPDH bands. **c** HELZ2 expression leads to a reduction in the steady state level of L1 RNA. HeLa-JVM cells were transfected as in panel (**b**). L1 RNA levels were determined by performing RT-qPCR using a primer set specific to RNAs derived from the transfected L1 (primer set: L1 [SV40]) and then were normalized to *ACTB* RNA levels (primer set: Beta-actin). *X*-axis, name of the constructs. *Y*-axis, relative level of L1 RNA normalized to the ORF1-T7 + pCMV-3Tag-8-Barr control. Pairwise comparisons relative to the control: $p = 0.032^*$ (HERC5); $1.7 \times 10^{-5***}$ (HELZ2); 0.14ns (OASL); 0.29ns (IFIT1); 0.20ns (DDX60L); and $4.4 \times 10^{-4**}$ (MOV10). **d** Differential effects of ISG proteins on ORF1p-FLAG cytoplasmic foci formation. HeLa-JVM cells were co-transfected with pJM101/L1.3FLAG (WT ORF1p-FLAG) and either a pCEP4 empty vector (control) or one of the following carboxyl-terminal 3xMYC epitope-tagged ISG protein expression plasmids: pALAF015 (HELZ2); pALAF027 (HELZ2 WA1&2); pALAF022 (OASL); pALAF023 (HERC5); or pALAF024 (MOV10) to visualize WT ORF1p-FLAG cytoplasmic foci and co-localization between WT ORF1p-FLAG and the candidate ISG protein. Shown are representative fluorescent microscopy images. White scale bars, 20 μm. **e** Quantification of L1 cytoplasmic foci formation. *X*-axis, name of the constructs co-transfected with pJM101/L1.3FLAG (WT ORF1p-FLAG); control, pCEP4. *Y*-axis, percentage of transfected cells with visible ORF1p signal exhibiting ORF1p-FLAG cytoplasmic foci. The numbers (*n*) within the green rectangles indicate the number of analyzed cells in each experiment. Pairwise comparisons relative to the pJM101/L1.3FLAG (WT ORF1p-FLAG) + pCEP4 control: $p = 8.6 \times 10^{-4***}$ (HERC5); $1.2 \times 10^{-7***}$ (HELZ2); 0.098ns (HELZ2 WA1&2); $1.0 \times 10^{-10***}$ (OASL); $2.7 \times 10^{-9***}$ (MOV10). Values represent the mean ± SEM from three (in panels [**a**] and [**e**]) or six (in panel [**c**]) independent biological replicates. The *p*-values were calculated using one-way ANOVA followed by Bonferroni−Holm post-hoc tests; ns: not significant; * $p < 0.05$; ** $p < 0.01$; *** $p < 0.001$.

more pronounced effect than the HELZ2 RNase activity on L1 retrotransposition and that mutations in the HELZ2 helicase domains affect L1 RNA stability, ORF1p levels, and ORF1p cytoplasmic localization to different extents.

### Knockdown of endogenous HELZ2 enhances L1 retrotransposition

To determine whether endogenous HELZ2 could inhibit L1 retrotransposition, we used small interfering RNAs (siRNAs) to reduce HELZ2 and MOV10 levels in HeLa-JVM cells. Control RT-qPCR experiments revealed a ~70% and ~80% knockdown of HELZ2 and MOV10 RNAs, respectively, when compared to a non-targeting siRNA control (Fig. 5e, middle panel); *mEGFPI*-based assays revealed a ~1.5-fold and ~3-fold increase in L1 retrotransposition efficiency in the siHELZ2 and siMOV10 treated cells, respectively (Fig. 5e, bottom panel). Thus, endogenous HELZ2 may also suppress L1 retrotransposition.

### HELZ2 and HERC5 recognize L1 RNA independent of RNP formation

We further investigated the mechanism of association between ORF1p-FLAG and HELZ2. Treatment of the WT ORF1p RNP complex with RNase A abolished the ORF1p-FLAG and HELZ2 interaction, suggesting that HELZ2, like PABPC1, associates with ORF1p in an RNA-dependent manner[50,64] (Fig. 6a).

To test whether L1 RNP formation is required for the association between WT ORF1p-FLAG and HELZ2, we compared the effects of HELZ2 overexpression on L1 RNA and ORF1p protein abundance in HeLa-JVM cells transfected with either pJM101/L1.3FLAG (ORF1p-FLAG) or pALAF008_L1.3FLAG_M8 (M8/RBM-FLAG). RT-qPCR using a probe set that specifically recognizes the SV40 poly(A) signal of plasmid expressed L1 RNA and western blot experiments conducted with an anti-FLAG antibody revealed a marked reduction in L1 RNA (~80% reduction) and ORF1p levels in both the WT ORF1p-FLAG or M8/RBM ORF1p-FLAG transfected cells upon HELZ2 overexpression when compared to controls (Fig. 6b). We observed a similar reduction in L1 WT ORF1p-FLAG and M8/RBM ORF1p-FLAG protein levels upon the co-expression of HERC5 in HeLa-JVM cells (Supplementary Fig. 6a). Thus, both HELZ2 and HERC5 overexpression appear to

destabilize L1 RNA and ORF1p, respectively, independent of WT L1 RNP formation.

### HELZ2 overexpression modestly inhibits Alu retrotransposition

To examine whether HELZ2 overexpression affects Alu retrotransposition, HeLa-HA cells[74] were transfected with an expression plasmid that contains both an engineered Alu-element harboring a *neo*-based retrotransposition indicator cassette (*neo*[Tet])[96] and a monocistronic L1 ORF2p-3xFLAG expression cassette[97]. HELZ2 overexpression reduced Alu retrotransposition by ~2-fold when compared to the respective controls (Fig. 6c). Additional experiments revealed that HELZ2 overexpression reduced L1 ORF2p and Alu RNA levels by ~80% and ~35%, respectively (Fig. 6d); the reduction in L1 RNA levels led to a corresponding decrease of L1 ORF2p protein levels (see below). Notably, the reductions in the levels of monocistronic and full-length L1 RNAs upon HELZ2 overexpression were quite similar (i.e., Fig. 6b vs. Fig. 6d), suggesting that the observed decrease in Alu retrotransposition may result mainly from the HELZ2-dependent destabilization of L1 RNA. That being stated, the co-expression of an Alu only expression plasmid (Alu_*neo*[Tet]) and HELZ2 in HeLa-HA cells still exhibited a ~40% reduction in Alu RNA levels, although this reduction was not as significant as observed with L1 RNA (Supplementary Fig. 6b vs. Fig. 6d).

### HELZ2 recognizes the 5′UTR of L1 RNA to reduce both L1 RNA levels and IFN-α induction

HELZ2 overexpression adversely affects L1 and Alu retrotransposition. Intriguingly, the sequences of the L1 5′UTR and L1 3′UTR are shared between the full-length L1 and monocistronic ORF2p expression constructs used in these assays. However, the monocistronic ORF2p expression cassette that drives Alu retrotransposition, but not the full-length bicistronic L1, contains a deletion of a conserved polypurine tract (Δppt) in the L1 3′UTR, which does not dramatically affect L1 retrotransposition[17]. Thus, we hypothesized that HELZ2 may recognize either RNA sequences or RNA structures in the L1 5′UTR and/or 3′UTR to destabilize L1 RNA.

To test the above hypothesis, we deleted the L1 5′UTR sequence from a WT L1 expression construct (pTMF3) that also

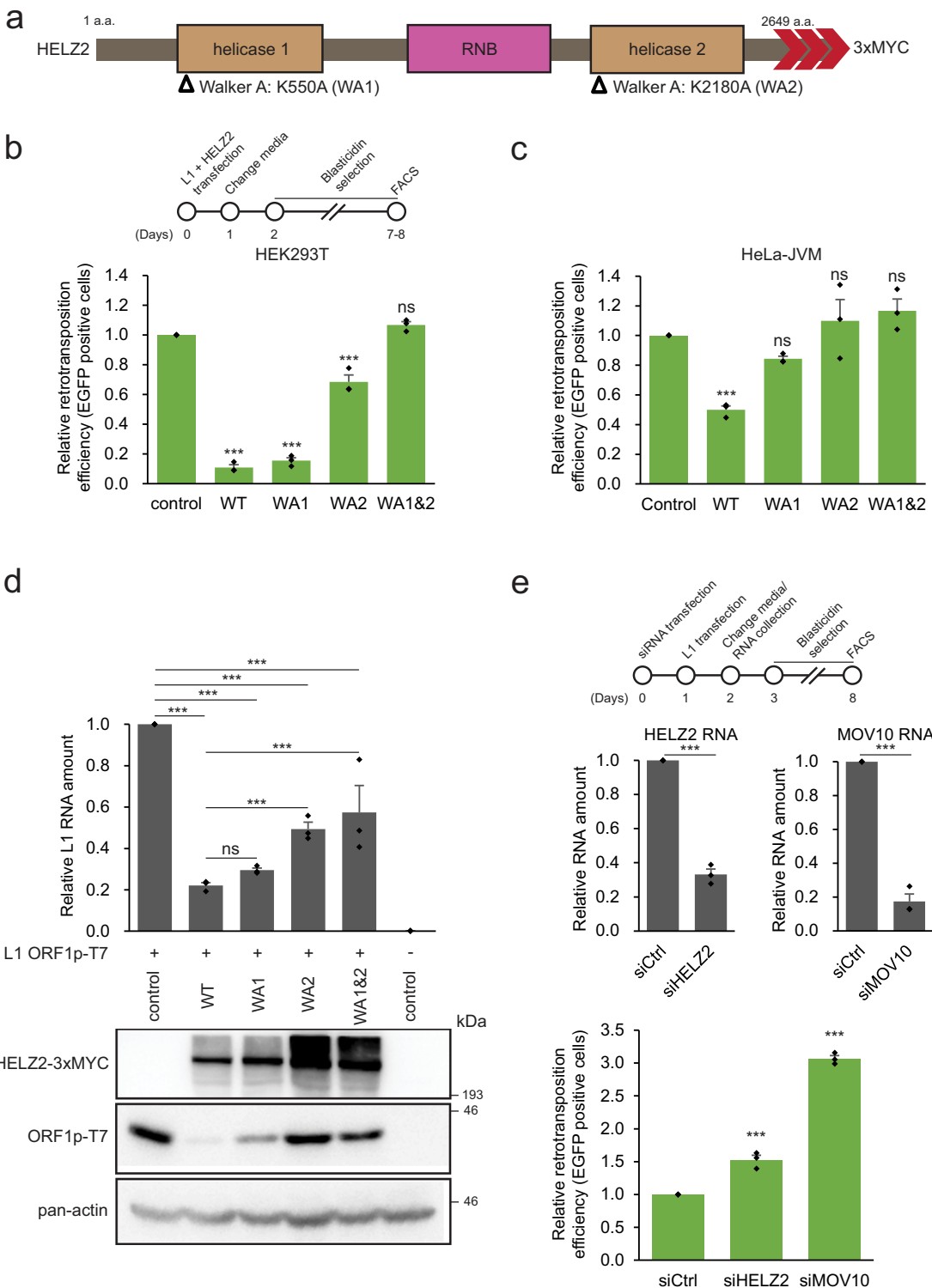

contains the 3'UTRΔppt sequence and drove L1 expression solely from the cytomegalovirus immediate-early (CMV) promoter (Fig. 6e, L1 [Δ5'UTR]; a.k.a. pTMF3_Δ5UTR). As an additional control, we also replaced the L1.3 coding sequences (*ORF1* and *ORF2*) with a firefly luciferase gene, creating a construct that has the L1 5'UTR and L1 3'UTRΔppt sequences surrounding the luciferase gene (Fig. 6e, Fluc; a.k.a pL1_[5&3UTRs]_Fluc). Co-transfection of HeLa-JVM cells with either pTMF3, pTMF3_Δ5UTR, or Fluc and HELZ2 followed by RT-qPCR (i.e., using probe sets that specifically recognize the SV40 poly(A)

signal of the plasmid, pTMF3, pTMF3_Δ5UTR, or Fluc RNAs; see Methods) revealed that HELZ2 overexpression significantly reduced the RNA levels derived from the L1 5'UTR-containing constructs irrespective of their downstream sequences (Fig. 6e). Consistently, HELZ2 overexpression did not significantly affect L1 retrotransposition efficiency in the L1 [Δ5'UTR] construct (Supplementary Fig. 6c). Independent control experiments in HeLa-JVM cells revealed that HELZ2 overexpression does not affect steady state RNA or protein levels produced from an inducible Tet-On firefly luciferase or human L1 ORFeus construct that lack

**Fig. 5 | The HELZ2 helicase activity is critical for L1 inhibition. a** Schematic of the HELZ2 protein domains. HELZ2 contains two putative helicase domains (helicase 1 and helicase 2), which surround a putative RNB exonuclease domain. Open triangles, positions of missense mutations in conserved amino acids within the Walker A (WA) boxes in the helicase 1 and helicase 2 domains (K550A [WA1] and K2180A [WA2], respectively). Red arrowheads, relative positions of the 3xMYC carboxyl-terminal epitope tag in the HELZ2 expression constructs. **b** The effect of mutations in the Walker A box on L1 retrotransposition in HEK293T cells. HEK293T cells were co-transfected with cepB-gfp-L1.3, which contains an *mEGFPI* retrotransposition indicator cassette, and either pCMV-3Tag-8-Barr (control), pALAF015 (WT HELZ2), or one of the following HELZ2 expression plasmids that contain a mutation(s) in the Walker A box: pALAF025 (WA1); pALAF026 (WA2); or pALAF027 (WA1&2). Cells co-transfected with cepB-gfp-L1.3RT(-) intronless and either pCMV-3Tag-8-Barr, pALAF015 (WT HELZ2), or a mutant HELZ2 plasmid served as transfection normalization and toxicity controls. Top, timeline of the assay for experiments shown in panels (**b**) and (**c**). *X*-axis, name of HELZ2 expression constructs co-transfected into cells with cepB-gfp-L1.3; control, pCMV-3Tag-8-Barr. *Y*-axis, relative retrotransposition efficiency normalized to the cepB-gfp-L1.3 + pCMV-3Tag-8-Barr control. Pairwise comparisons relative to the cepB-gfp-L1.3 (*mEGFPI*) + pCMV-3Tag-8-Barr control: $p = 2.5 \times 10^{-11}$*** (WT HELZ2); $3.5 \times 10^{-11}$*** (WA1); $1.7 \times 10^{-6}$*** (WA2); and $0.070$ns (WA1&2). **c** The effect of mutations in the Walker A box on L1 retrotransposition in HeLa-JVM cells. HeLa-JVM cells were co-transfected as in panel (**b**). Retrotransposition efficiencies were calculated as described in panel (**b**). Pairwise comparisons relative to the control: $p = 0.00087$*** (WT HELZ2); $0.26$ns (WA1); $0.32$ns (WA2); and $0.32$ns (WA1&2). **d** Mutations in the HELZ2 helicase domains reduce the ability to inhibit L1 ORF1p and RNA. HeLa-JVM cells were transfected with pTMF3 (L1 ORF1p-T7), denoted by + symbol, and either pCMV-3Tag-8-Barr (control), pALAF015 (WT HELZ2), or an individual HELZ2 expression plasmid containing a mutation(s) in the Walker A box: pALAF025 (WA1), pALAF026 (WA2), or pALAF027 (WA1&2). Top: L1 RNA levels were determined by RT-qPCR using primers directed against sequences in the transfected L1 RNA (primer set: L1 [SV40]) and then were normalized to *ACTB* RNA levels (primer set: Beta-actin). Pairwise comparisons relative to the pTMF3 (L1 ORF1p-T7) + pCMV-3Tag-8-Barr control: $p = 9.5 \times 10^{-9}$*** (WT); $1.9 \times 10^{-8}$*** (WA1); $7.3 \times 10^{-7}$*** (WA2); and $1.5 \times 10^{-6}$*** (WA1&2). Pairwise comparisons relative to the pTMF3 (L1 ORF1p-T7) + WT HELZ2: $p = 0.56$ns (WA1); $5.9 \times 10^{-4}$*** (WA2); $1.9 \times 10^{-4}$*** (WA1&2). Bottom: western blot image displaying ORF1p-T7 bands. HELZ2 expression was detected using an anti-MYC antibody. ORF1p was detected using an anti-T7 antibody. Pan-actin served as a sample processing control. **e** Small-interfering RNA (siRNA)-mediated knockdown of endogenous HELZ2 increases L1 retrotransposition. Top, timeline of the assay conducted in HeLa-JVM cells. Cells were transfected with a non-targeting siRNA control (siCtrl), siRNA targeting HELZ2 (siHELZ2), or siRNA targeting MOV10 (siMOV10). Middle left panel, HELZ2 RNA levels in siRNA treated cells. Middle right panel, MOV10 RNA levels in siRNA treated cells. *X*-axes, name of the siRNA. HELZ2 and MOV10 RNA levels were determined using RT-qPCR (primer sets: HELZ2 and MOV10, respectively) and then were normalized to *ACTB* RNA levels (primer set: Beta-actin). Y-axes, relative HELZ2 or MOV10 RNA levels normalized to the siCtrl. A two-tailed, unpaired Student's t-test was used to calculate the *p*-values relative to the siRNA control: $p = 3.1 \times 10^{-5}$*** (siHELZ2); and $5.2 \times 10^{-5}$*** (siMOV10). Bottom panel, HeLa-JVM cells were transfected with either siCtrl, siHELZ2, or siMOV10, followed by transfection with either cepB-gfp-L1.3 or cepB-gfp-L1.3RT(-) intronless, which was used to normalize transfection efficiencies. *X*-axis, name of the siRNA. *Y*-axis, relative retrotransposition efficiency. Pairwise comparisons relative to the non-targeting siRNA control: $p = 2.9 \times 10^{-4}$*** (siHELZ2); and $2.0 \times 10^{-7}$*** (siMOV10). All the reported values represent the mean ± SEM from three independent biological replicates. The *p*-values, except for the RT-qPCR experiment shown in panel (**e**), were calculated using a one-way ANOVA followed by a Bonferroni−Holm post hoc tests. ns: not significant; * $p < 0.05$; *** $p < 0.001$.

the L1 5′UTR[45] (Supplementary Fig. 6d, e). Together, these data suggest that HELZ2 destabilizes L1 RNA by recognizing RNA sequences and/or RNA structure(s) within the L1 5′UTR.

Because previous experiments reported that L1 RNA induces a type I IFN response[40,42,46] (Fig. 2f), we next tested whether the destabilization of L1 RNA by HELZ2 leads to a decrease in L1-mediated IFN-α induction. Strikingly, HELZ2 overexpression reduced the level of L1-dependent IFN-α induction to less than 5% of the control pJM101/L1.3FLAG construct (Fig. 6f, compare the leftmost and middle data graphs). Notably, this level of IFN-α induction was even lower than that observed in cells transfected with only the pCEP4 empty vector (Fig. 6f). Co-expression of HELZ2 with the pCEP4 empty vector also reduced the IFN-α level when compared to the pCEP4 only empty vector (Fig. 6f), raising the possibility that HELZ2 overexpression may also reduce the stability of endogenous immunogenic RNAs, thereby reducing basal levels of IFN-α induction.

## Discussion

Previous studies identified antiviral factors involved in innate immune responses that inhibit L1 retrotransposition by destabilizing L1 RNA, L1 proteins, L1 RNPs, and perhaps L1 (-) strand cDNAs (see "Results": *"Proteins produced by Interferon-Stimulated Genes (ISGs) as potential L1 regulators"* for a complete list). In this study, we uncovered 16 additional ISG proteins that are enriched in IP/LC-MS/MS experiments conducted with WT ORF1p, but not the M8/RBM ORF1p mutant, which exhibits both attenuated RNA binding and L1 cytoplasmic foci formation. We focused on three of these ISG proteins (HELZ2, HERC5, and OASL) in more detail.

HELZ2, HERC5, and OASL were predominantly localized in the cytoplasm, and upon overexpression, inhibited the retrotransposition of an engineered wild-type L1 (Fig. 4a; Supplementary Fig. 4b). Overexpression experiments further revealed that HELZ2 interacts with ORF1p in an RNA-dependent manner (Figs. 3c and 6a) and reduces the steady state levels of engineered L1 RNA, ORF1p, and ORF1p cytoplasmic foci formation (Fig. 4b−e; Supplementary Fig. 4c). By comparison, HERC5 destabilizes ORF1p, but does not affect its cellular

localization, whereas OASL mainly impairs ORF1p cytoplasmic foci formation. Thus, ISG proteins that predominantly act in the cytoplasm have the potential to inhibit L1 retrotransposition by acting at various steps in the L1 retrotransposition cycle (Fig. 7).

HELZ2 is a poorly characterized protein containing two putative RNA helicase domains that surround a centrally-located exoribonuclease (RNB) domain[98]. Other RNB-containing proteins are known to function in RNA quality control (e.g., the yeast and human RNA exosome component Dis3, and prokaryotic cold shock inducible protein RNase R[99]). A more in-depth analysis of HELZ2 revealed mechanistic similarities to other RNB-containing proteins, which can degrade highly structured RNAs through its concerted helicase and 3′ to 5′ exoribonuclease activities[100,101]. Thus, it is tempting to suggest that HELZ2 might function in a similar stepwise manner, where its helicase activity initially unwinds L1 RNA secondary structures, allowing the subsequent degradation of L1 RNA by the HELZ2 3′ to 5′ exoribonuclease activity (Fig. 7 and Supplementary Fig. 5d). Indeed, a HELZ2 helicase double mutant (WA1&2), but not a putative RNase-deficient mutant (dRNase), severely impaired the ability of HELZ2 to inhibit L1 retrotransposition (Fig. 5b, c; Supplementary Fig. 5e, f, g, and h), suggesting the HELZ2 helicase activity likely functions upstream of the single-strand 3′ to 5′ exoribonuclease activity to degrade L1 RNA. Because ORF1p-binding to L1 RNA is proposed to stabilize L1 RNAs[12,102], we speculate that some regions of L1 RNA might be protected from HELZ2 degradation due to ORF1p RNA binding, whereas other regions of L1 RNA that have complex RNA secondary structures may be preferential HELZ2 targets. If so, HELZ2 might primarily destabilize these regions of L1 RNA to inhibit retrotransposition.

Previous studies demonstrated CpG DNA methylation of the L1 5′ UTR is a potent means to inhibit endogenous L1 transcription[103–105]. DNA sequences within the 5′UTRs of older L1s (e.g., members of the L1PA3 and L1PA4 subfamilies) also bind repressive Krüppel-associated Box-containing Zinc-Finger Protein 93 (ZNF93) to repress their transcription and deletion of these repressive sequences allowed the subsequent amplification of the L1PA2 and human-specific L1PA1 subfamilies in the human genome[106,107]. Notably, HELZ2 was

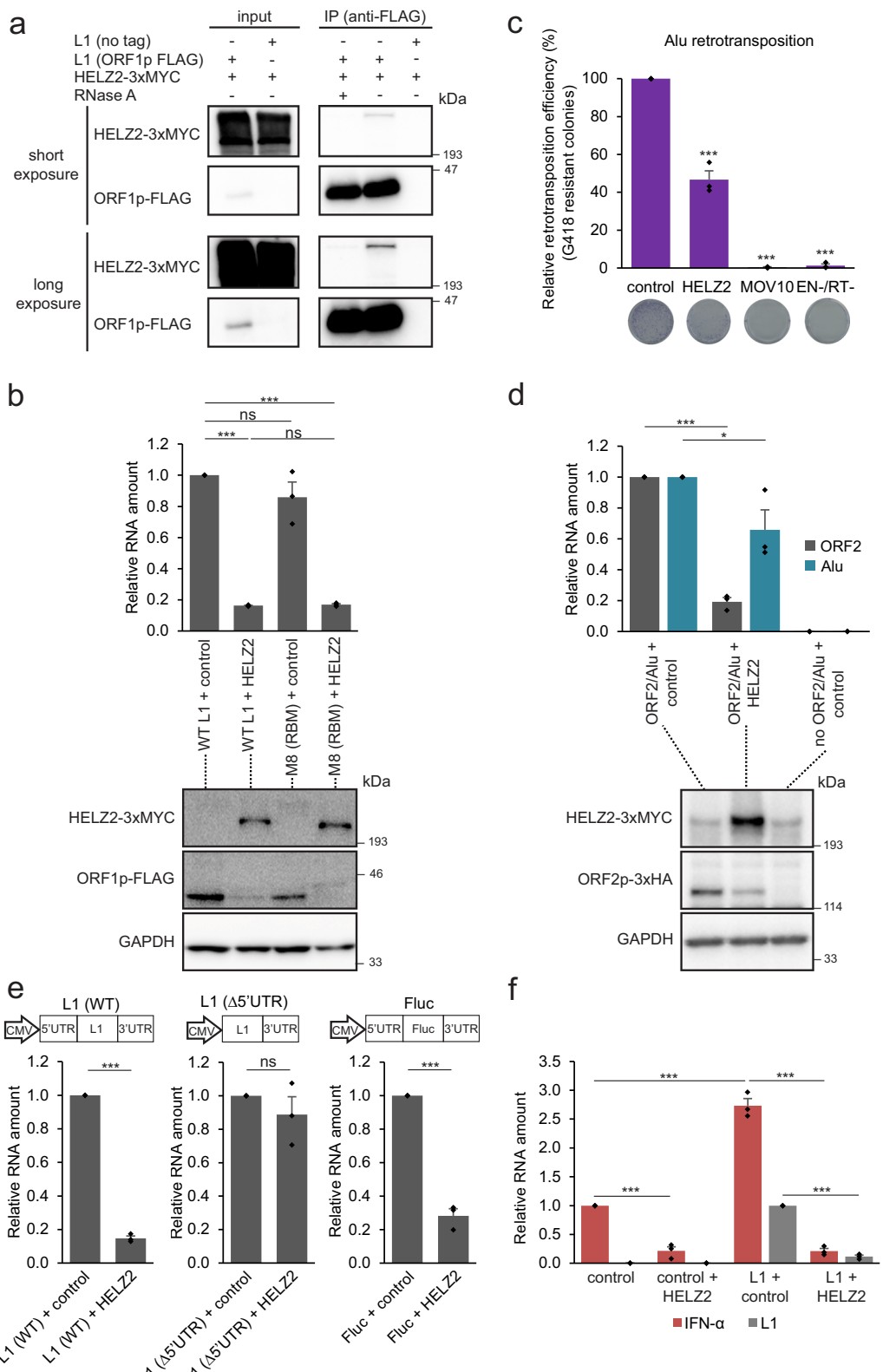

discovered as a transcriptional co-activator of peroxisome proliferator-activated receptor alpha (PPAR-α)[108] and PPAR-γ[109]. Because the L1 5′UTR also contains multiple transcription factor binding sites that drive L1 expression, it remains possible that HELZ2 might repress L1 transcription[21,22,110–112].

Intriguingly, we found that L1 RNAs containing 5′UTR sequences appear to be particularly susceptible to HELZ2-mediated RNA degradation (Fig. 6e; Supplementary Fig. 6d and e), thereby representing a potential post-transcriptional mechanism by which RNA sequences and/or RNA structures within the 5′UTR might be targeted by host proteins to inhibit L1 retrotransposition. RNB domains typically are flanked by cold shock and S1 domains that form an RNA binding channel. However, HELZ2 appears to lack these domains, as well as conserved amino acids associated with these domains;[98]

**Fig. 6 | HELZ2 destabilizes L1 RNA through recognition of the L1 5′UTR sequence, leading to attenuation of L1-mediated IFN-α induction. a** The association between ORF1p and HELZ2 is RNA-dependent. HEK293T cells were co-transfected with pALAF015 (HELZ2-3xMYC) and either pJM101/L1.3FLAG (WT ORF1p-FLAG) or pJM101/L1.3 (no tag). The input and anti-FLAG IP fractions were analyzed by western blot using an anti-FLAG antibody to detect ORF1p-FLAG or an anti-MYC antibody to detect HELZ2-3xMYC. Shown are short (top blots) and longer (bottom blots) chemiluminescence western blot exposures. **b** HELZ2 expression reduces steady state levels of L1 RNA and ORF1p independent of ORF1p RNA binding. HeLa-JVM cells were co-transfected with pJM101/L1.3FLAG (WT ORF1p-FLAG) or the pALAF008 ORF1p-FLAG (M8 [RBM]) mutant expression plasmid and either pCEP4 (control) or pALAF015 (HELZ2). Top: L1 RNA amounts were determined by RT-qPCR (primer set: L1 [SV40]) and then were normalized to *ACTB* RNA levels (primer set: Beta-actin). The L1 RNA values were normalized to the WT L1 or ORF1p-FLAG (M8 [RBM]) + pCEP4 control transfections. Pairwise comparisons (in parentheses) relative to the (WT L1 + control) are shown: $p = 7.1 \times 10^{-7}$*** (WT L1 + HELZ2); $0.090^{ns}$ (M8 [RBM] + control); $6.7 \times 10^{-7}$*** (M8 [RBM] + HELZ2). Pairwise comparisons of (WT L1 + HELZ2) vs. (M8 [RBM] + HELZ2), $p = 0.92^{ns}$. Bottom: ORF1p-FLAG and HELZ2 protein levels were detected by western blot using anti-MYC and anti-FLAG antibodies, respectively. GAPDH served as a sample processing control. **c** HELZ2 expression inhibits Alu retrotransposition. HeLa-HA cells were co-transfected with pTMO2F3_Alu (which expresses an Alu element marked with *neo*-based retrotransposition indicator cassette and monocistronic version of L1 ORF2p [see Methods]), pTMO2F3D145AD702A_Alu (which expresses an Alu element marked with *neo*-based retrotransposition indicator cassette and an EN-/RT-mutant version of L1 ORF2 [see Methods]), or phrGFP-C (a transfection normalization control) and either pCMV-3Tag-8-Barr (control), pALAF015 (WT HELZ2), or pALAF024 (WT MOV10). *X*-axis, name of constructs. *Y*-axis, the percentage of G418-resistant foci, indicative of Alu retrotransposition, relative to the pTMO2F3_Alu + pCMV-3Tag-8-Barr control (see "Methods" for more detail). Representative images of G418-resistant foci are shown below the graph. Pairwise comparisons relative to the pTMO2F3_Alu + pCMV-3Tag-8-Barr control: $p = 7.8 \times 10^{-5}$*** (HELZ2); $1.8 \times 10^{-7}$*** (MOV10); and $1.6 \times 10^{-7}$*** (EN-/RT-). **d** HELZ2 expression leads to a reduction in monocistronic ORF2 L1 RNA and ORF2p levels. HeLa-HA cells were co-transfected with pTMO2H3_Alu (ORF2p-3xHA and Alu) and either pCMV-3Tag-8-Barr (control)

or pALAF015 (HELZ2). Top: ORF2 (gray bars) and Alu RNA (blue bars) levels were determined using RT-qPCR (primer sets: L1 [SV40] and *mneoI* [Alu or L1], respectively) and normalized to *ACTB* RNA levels (primer set: Beta-actin). *X*-axis, co-transfected constructs name. *Y*-axis, relative RNA level normalized to the pTMO2-H3_Alu (ORF2p-3xHA and Alu) + pCMV-3Tag-8-Barr control. L1 ORF2 RNA pairwise comparison (ORF2/Alu + control vs. ORF2/Alu + HELZ2), $p = 7.2 \times 10^{-8}$***. Alu RNA pairwise comparison (ORF2/Alu + control vs. ORF2/Alu + HELZ2), $p = 0.018$*. Bottom: Western blotting using an anti-HA antibody was used to detect ORF2p. GAPDH served as a sample processing control. **e** The L1 5′UTR is required for HELZ2-mediated reduction of L1 RNA levels. HeLa-JVM cells were co-transfected with L1 (WT), L1 (Δ5′UTR), or Fluc (a firefly luciferase gene flanked by the L1 5′ and 3′UTRs) and either pCMV-3Tag-8-barr (control) or pALAF015 (HELZ2). Schematics of the constructs are above the bar charts. RNA levels were determined by RT-qPCR using the following primer sets: L1 (SV40) (for L1 WT and L1[Δ5′UTR]) or Luciferase (for Fluc) and then were normalized to *GAPDH* RNA levels (primer set: GAPDH). *X*-axis, name of respective constructs co-transfected with pCMV-3Tag-8-Barr (control) or pALAF015 (HELZ2); *Y*-axis, the relative amount of L1 or Fluc-based RNA relative to the relevant pairwise control (e.g., the L1 expression plasmid + pCMV-3Tag-8-Barr or the Fluc-based plasmid + pCMV-3Tag-8-Barr). Two-tailed, unpaired Student's t-tests: $p = 3.9 \times 10^{-7}$*** (left plot); $0.35^{ns}$ (middle plot); $7.1 \times 10^{-5}$*** (right plot). **f** HELZ2 expression represses L1-induced IFN-α expression. HEK293T cells were transfected with only the pCEP4 empty vector (control); or co-transfected with pCEP4 empty vector and pALAF015 (control + HELZ2); pJM101/L1.3FLAG and a pCEP4 empty vector (L1 + control); or pJM101/L1.3FLAG and pALAF015 (L1 + HELZ2). IFN-α (red bars) and L1 RNA (gray bars) levels were determined by RT-qPCR (using a primer set against IFNs [IFN-α, which amplifies IFN-α1 and IFN-α13] or the primer set *mneoI* [Alu or L1], respectively) and normalized to *ACTB* RNA levels (primer set: Beta-actin). The RNA levels then were normalized to the pCEP4 only (control) for IFN-α, and (L1 + control) for L1. L1 RNA pairwise comparison: (L1 + control vs. L1 + HELZ2), $p = 1.4 \times 10^{-10}$***. IFN-α RNA pairwise comparisons: (control vs. control + HELZ2), $p = 1.4 \times 10^{-4}$***; (control vs. control + L1), $p = 7.2 \times 10^{-7}$***; (L1 + control vs. L1 + HELZ2), $p = 5.7 \times 10^{-8}$***. All values are reported as the mean ± SEM of three independent biological replicates. With the exception of panel (**e**), the *p*-values were calculated using a one-way ANOVA followed by Bonferroni–Holm post hoc tests. ns: not significant; * $p < 0.05$; *** $p < 0.001$.

thus, it remains unclear which domain of HELZ2 recognizes the L1 5′ UTR. Future studies also are needed to critically test whether HELZ2 acts to destabilize the polypurine tract within the L1 3′UTR (which is absent from most of our expression vectors)[113].

The overexpression of a WT L1 construct led to a modest upregulation of IFN-α expression, which previously was reported to contribute to inflammation, autoimmunity, and aging phenotypes[40,42–46]. A similar upregulation of type I IFN expression was observed upon the overexpression of an RT-deficient L1 and was slightly more pronounced upon the overexpression of the ORF1p M8/RBM mutant. These data suggest that L1 RNA and or the L1-encoded proteins, but not intermediates generated during TPRT (e.g., L1 cDNAs), are responsible for the modest type I IFN upregulation observed in HEK293T cells. Consistently, HELZ2 overexpression impaired L1-mediated type I IFN upregulation and/or reduced type I IFN expression compared to a pCEP4 empty vector control below baseline levels (Fig. 6f), raising the possibility that HELZ2 also may reduce the expression of endogenous immunogenic RNA(s). Because we observed a modest increase in some cytokines in M8/RBM mutant-transfected cells (Supplementary Data 3), our data are in general agreement with other reports, which demonstrated that higher L1 RNA expression levels can lead to the upregulation of the type I IFN response[42,45,46].

As L1s lack an extracellular phase in their replication cycle, one can posit that L1s would benefit from not triggering an innate immune response. That being stated, why the overexpression of the ORF1p M8/RBM mutant led to a more robust, yet modest, induction of type I IFN expression than the WT and RT-deficient L1s (Fig. 2f) requires further study. It is possible that ORF1p L1 RNA binding and/or the sequestration of L1 RNPs within cytoplasmic foci effectively shields L1 RNAs from

eliciting an interferon response and that the attenuated ability of the ORF1p M8/RBM mutant to bind L1 RNA could lead to higher levels of unprotected L1 RNA substrates that act as "triggers", contributing to type I IFN expression (Fig. 7). Our working model further suggests that ORF1p binding to L1 RNA may attenuate the type I interferon response, which, in turn, might reduce the expression of inhibitory ISG proteins. Indeed, these data are consistent with a recent study, which reported that depletion of the Human Silencing Hub (HUSH complex) correlates with the de-repression of primate-specific L1s and that the resultant L1 double-stranded RNAs may drive physiological or autoinflammatory responses in human cells[46]. Clearly, future studies are necessary to elucidate how and if L1 double-stranded RNAs, or perhaps single-stranded cDNAs, play important contributory roles to innate immune activation and human autoimmune diseases[114,115].

## Methods
### Cell lines and cell culture conditions
The human HeLa-JVM cervical cancer-derived[17], U-2 OS osteosarcoma-derived, and HEK293T embryonic carcinoma-derived cell lines were grown in Dulbecco's Modified Eagle Medium (DMEM) (Nissui, Tokyo, Japan) supplemented with 10% (volume/volume [v/v]) fetal bovine serum (FBS) (Gibco, Amarillo, Texas, United States or Capricorn Scientific, Ebsdorfergrund, Germany), 0.165% (weight/volume [w/v]) NaHCO$_3$, 100 U/mL penicillin G (Sigma-Aldrich, St. Louis, MO, United States), 100 μg/mL streptomycin (Sigma-Aldrich), and 2 mM L-glutamine (Sigma-Aldrich). HeLa-HA cells[74] were grown in Minimum Essential Medium (MEM) (Gibco) supplemented with 10% (v/v) FBS (Capricorn Scientific), 0.165% (w/v) NaHCO$_3$, 100 U/mL penicillin G, 100 μg/mL streptomycin, 2 mM L-glutamine, and 1× MEM Non-Essential Amino Acids Solution (Nacalai, Kyoto, Japan). The cell lines

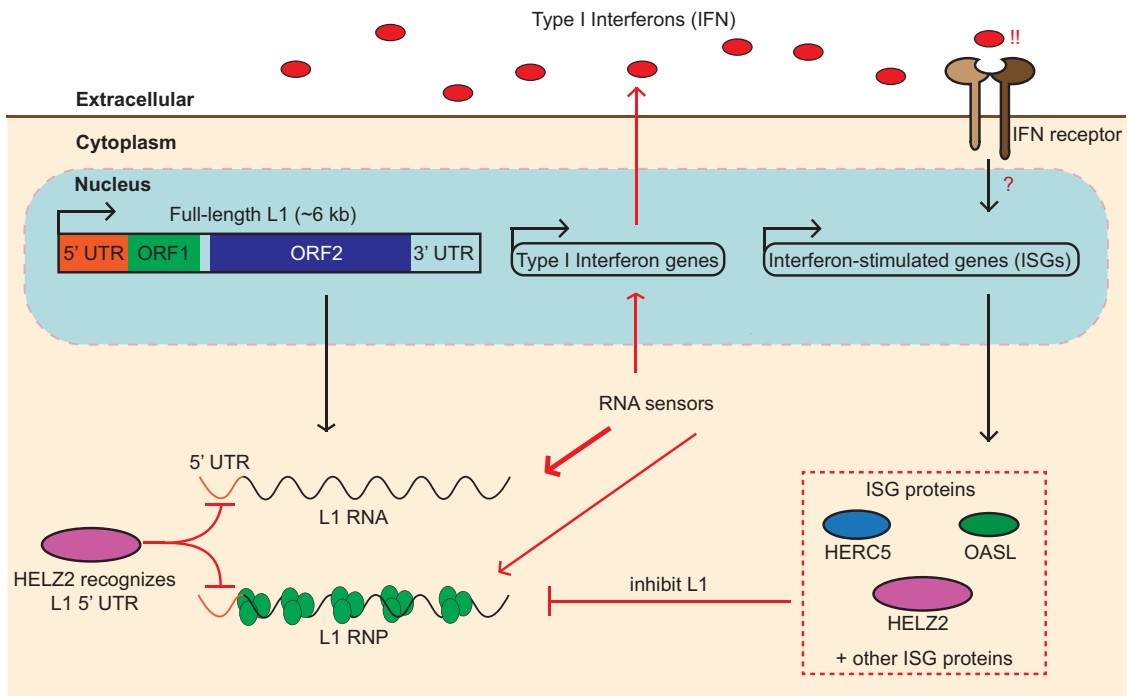

**Fig. 7 | A working model hypothesizing a negative feedback loop between L1 RNA levels and ISG proteins.** L1 RNAs and/or RNPs can be detected by cytoplasmic RNA sensors, which elicit the secretion of type I interferons (IFNs); ORF1p RNA binding might shield L1 RNA from the sensors. IFN-binding to the extracellular IFN cell surface receptors then activates a signaling cascade, which induces the expression of ISGs, including *HELZ2*, *HERC5*, and *OASL*. These ISG proteins appear to inhibit L1 retrotransposition at different steps in the L1 retrotransposition cycle. HELZ2 appears to recognize RNA sequences and/or RNA structures within the L1 5′ UTR, independently of ORF1p RNA binding, leading to the degradation of L1 RNA and subsequent blunting of the IFN response.

were grown at 37 °C in 100% humidified incubators supplied with 5% CO$_2$. The cell lines tested negative for mycoplasma contamination using a PCR-based method using the VenorGeM Classic Mycoplasma Detection Kit (Sigma-Aldrich). STR-genotyping was performed to confirm the identity of HeLa-JVM, HeLa-HA, U-2 OS, and HEK293T cells.

## Plasmid construction

**Creation of the ORF1p-FLAG mutant constructs.** Briefly, the pJM101/L1.3FLAG[7,50] plasmid was used, unless otherwise indicated, to construct the plasmids in this study. Briefly, pJM101/L1.3FLAG DNA was used as a PCR template in conjunction with oligonucleotide primers containing the respective ORF1p mutations to generate the ORF1 mutants. The amplified PCR products and pJM101/L1.3FLAG plasmid DNA then were digested with *Not*I and *Age*I and ligated using the DNA Ligation Kit Mighty Mix (TaKaRa Bio, Shiga, Japan) at 16 °C for 30 min. The resultant ligation products were transformed into *E. coli* XL1-Blue cells and plated on Luria Broth (LB) agar plates containing 50 µg/mL ampicillin. The resultant plasmids then were sequenced from the *Not*I to *Age*I restriction sites to ensure the integrity of the mutants.

**Creation of the mCherry/G3BP1, ISG fusion protein, and HELZ2 mutant constructs.** The *G3BP1* cDNA sequence was amplified from a HeLa-JVM total cDNA library and concurrently inserted in-frame with an *mCherry*-coding sequence into a lentiviral vector (pCW)[116] using the in-Fusion Cloning Kit (TaKaRa Bio). The *HERC5*, *HELZ2*, *OASL*, *MOV10*, *IFIT1*, and *DDX60L* cDNAs were amplified from either a HeLa-JVM or HEK293T total cDNA library and inserted into the *Not*I and *Hind*III restriction sites in the pCMV-3Tag-9 vector (Agilent Technologies, Santa Clara, CA, United States) using either the Gibson Assembly Cloning Kit (New England Biolabs, Ipswich, MA, United States) or in-Fusion Cloning Kit. To generate *HELZ2* mutations, whole plasmid DNAs were amplified using primers harboring the intended mutations in separate reactions to avoid the formation of primer dimers. The

template DNA then was digested with *Dpn*I at 37 °C for 1 h followed by heat inactivation at 80 °C for 20 min. The PCR amplified DNA fragments were, mixed, annealed, and transformed into *E. coli* XL1-Blue cells.

## Plasmids used in this study

For mammalian cell experiments, plasmids were purified using the GenElute HP Plasmid Midiprep Kit (Sigma-Aldrich). All of the L1 expression plasmids contain a retrotransposition-competent L1 (L1.3, Genbank: L19088). The amino acid residues of ORF1p or ORF2p were counted from the first methionine of the L1.3 ORF1p and L1.3 ORF2p, respectively. The plasmids used in the study are listed below:

*pCEP4 (Invitrogen)*: the mammalian expression vector backbone used for cloning pJM101/L1.3 and pJJ101/L1.3 variants.

*phrGFP-C (Agilent technology)*: contains a humanized Renilla *GFP* gene whose expression is driven by a cytomegalovirus immediate-early (CMV) promoter.

*pJM101/L1.3*: was described previously[5,60]. This plasmid contains the full-length L1.3, cloned into the pCEP4 vector plasmid. L1 expression is driven by the CMV and L1.3 5′UTR promoters. The *mneoI* retrotransposition cassette was inserted into the L1.3 3′UTR as described previously[17].

*pJM101/L1.3FLAG*: was described previously[50]. This plasmid is a derivative of pJM101/L1.3 that contain a single copy of the *FLAG* epitope tag fused in-frame to the 3′ end of the L1.3 *ORF1* sequence.

*pJM105/L1.3*: was described previously[25]. This plasmid is a derivative of pJM101/L1.3 that contains a D702A mutation in the ORF2p reverse transcriptase active site.

*pTMF3*: was described previously[97]. This plasmid is a derivative of pJM101/L1.3. A *T7 gene10* epitope tag was fused in-frame to the 3′ end of the *ORF1* sequence and three copies of a *FLAG* epitope tag were fused to the 3′ end of the *ORF2* sequence. This plasmid lacks the polypurine sequence in the L1 3′UTR.

*pTMF3_ΔSUTR*: is a derivative of pTMF3 that contains a deletion of the L1.3 5′UTR sequence.

*pTMF3_M8_ORF1*: is a derivative of pTMF3 that contains the M8 (RBM) mutations: R206A, R210A, and R211A in ORF1p, which impairs the ability of ORF1p to bind RNA[12].

*pL1(S&3UTRs)_Fluc*: is a derivative of pTMF3 that contains a firefly luciferase gene in place of the L1.3-coding region.

*pJJ101/L1.3*: was described previously[117]. This plasmid is similar to pJM101/L1.3, but contains an *mblastI* retrotransposition indicator cassette within the L1.3 3′UTR.

*pJJ105/L1.3*: was described previously[117]. This plasmid is a derivative of pJJ101/L1.3 that contains a D702A mutation in the ORF2p reverse transcriptase active site.

*pALAF001_L1.3FLAG_M1*: is a derivative of pJM101/L1.3FLAG that contains the N157A and R159A mutations in ORF1p, which abolished ORF1p cytoplasmic foci formation[48].

*pALAF002_L1.3FLAG_M2*: is a derivative of pJM101/L1.3FLAG that contains the R117A and E122A mutations in ORF1p, which are proposed to adversely affect ORF1p trimerization[58].

*pALAF003_L1.3FLAG_M3*: is a derivative of pJM101/L1.3FLAG that contains the N142A mutation in ORF1p, which is proposed to bind a chloride ion to stabilize ORF1p trimerization[12].

*pALAF004_L1.3FLAG_M4*: is a derivative of pJM101/L1.3FLAG that contains the R135A mutation in ORF1p, which is proposed to bind a chloride ion to stabilize ORF1p trimerization[12].

*pALAF005_L1.3FLAG_M5*: is a derivative of pJM101/L1.3FLAG that contains the E116A and D123A mutations in ORF1p, which are proposed to act as a binding site for host factors[12].

*pALAF006_L1.3FLAG_M6*: is a derivative of pJM101/L1.3FLAG that contains the K137A and K140A mutations in ORF1p, which reduces the ability of ORF1p to bind L1 RNA[12].

*pALAF007_L1.3FLAG_M7*: is a derivative of pJM101/L1.3FLAG that contains the R235A mutation in ORF1p, which reduces the ability of ORF1p to bind L1 RNA[49].

*pALAF008_L1.3FLAG_M8 (RBM)*: is a derivative of pJM101/L1.3FLAG that contains the R206A, R210A, and R211A mutations in ORF1p, which severely impair the ability of ORF1p to bind RNA[12].

*pALAF009_L1.3FLAG_M9*: is a derivative of pJM101/L1.3FLAG that contains the R261A mutation in ORF1p, which reduces the ability of ORF1p to bind L1 RNA[49].

*pALAF010_L1.3FLAG_M10*: is a derivative of pJM101/L1.3FLAG that contains the Y282A mutation in ORF1p, which is proposed to reduce nucleic chaperone activity[49].

*pALAF012_mCherry-G3BP1_pCW*: contains the *mCherry* sequence fused in frame to a human *G3BP1* cDNA in a lentiviral expression vector, pCW[116]. The puromycin resistant gene and reverse tetracycline-controlled trans-activator (rtTA) coding regions are in-frame and are expressed by a human PGK promoter; puromycin and rtTA are separated by a self-cleaving T2A peptide so that each protein can be expressed from the bicistronic transcript. The *mCherry-G3BP1* cDNA is expressed from a doxycycline inducible (Tet-On) promoter. In the presence of doxycycline, rtTA can adopt an altered confirmation that allows it to bind the Tet-On promoter to allow *mCherry-G3BP1* expression.

*pALAF015_hHELZ2L-3xMYC*: contains the canonical human *HELZ2* long isoform cDNA (2649 bps) cloned into pCMV-3Tag-9 (Agilent Technologies), which allows the expression of a HELZ2-3xMYC fusion protein. The CMV promoter drives *HELZ2-3xMYC* expression.

*pALAF016_hIFIT1-3xMYC*: contains the human *IFIT1* cDNA cloned into pCMV-3Tag-9, which allows the expression of a hIFIT1-3xMYC fusion protein. The CMV promoter drives *IFIT1-3xMYC* expression.

*pALAF021_hDDX60L-3xMYC*: contains the human *DDX60L* cDNA cloned into pCMV-3Tag-9, which allows the expression of a hDDX60L-3xMYC fusion protein. The CMV promoter drives *DDX60L-3xMYC* expression.

*pALAF022_hOASL-3xMYC*: contains the human *OASL* cDNA cloned into pCMV-3Tag-9, which allows the expression of the OASL-3xMYC fusion protein. The CMV promoter drives *OASL-3xMYC* expression.

*pALAF023_hHERC5-3xMYC*: contains the human *HERC5* cDNA cloned into pCMV-3Tag-9, which allows the expression of a HERC5-3xMYC fusion protein. The CMV promoter drives *HERC5-3xMYC* expression.

*pALAF024_hMOV10-3xMYC*: contains the human *MOV10* cDNA cloned into pCMV-3Tag-9, which allows the expression of a MOV10-3xMYC fusion protein. The CMV promoter drives *MOV10-3xMYC* expression.

*cepB-gfp-L1.3*: was described previously[97]. The plasmid contains the full-length L1.3 with an *EGFP* retrotransposition reporter cassette, *mEGFPI*. L1.3 expression is augmented by the L1 5′UTR promoter. The plasmid backbone also contains a *blasticidin S-deaminase* (*BSD*) selectable marker driven by the SV40 early promoter.

*cepB-gfp-L1.3RT(-) intronless*: was described previously[97]. The plasmid is similar to cepB-gfp-L1.3RT(-) except that the intron in the *mEGFPI* retrotransposition cassette was removed, allowing EGFP expression in the absence of L1.3 retrotransposition.

*cep99-gfp-L1.3*: was described previously[97]. The plasmid is similar to cepB-gfp-L1.3 but contains the puromycin resistant gene instead of the blasticidin resistance gene as a selectable marker.

*cep99-gfp-L1.3RT(-) intronless*: was described previously[97]. The plasmid is similar to cep99-gfp-L1.3 except that it contains the D702A mutation in the ORF2p reverse transcriptase domain and the intron in the *mEGFPI* retrotransposition cassette was removed, allowing EGFP expression in the absence of L1.3 retrotransposition.

*pALAF025_hHELZ2L-3xMYC_WA1*: is a derivative of pALAF015_hHELZ2L-3xMYC that contains the K550A mutation in the Walker A motif of the N-terminal HELZ2 helicase domain, which is predicted to inactivate the ATP binding ability of the helicase domain[93].

*pALAF026_hHELZ2L-3xMYC_WA2*: is a derivative of pALAF015_hHELZ2L-3xMYC that contains the K2180A mutation in the Walker A motif of the carboxyl-terminal HELZ2 helicase domain, which is predicted to inactivate the ATP binding ability of the helicase domain[93].

*pALAF027_hHELZ2L-3xMYC_WA1&2*: is a derivative of pALAF015_hHELZ2L-3xMYC that contains the K550A and K2180A mutations in the Walker A motifs of both HELZ2 helicase domains[93].

*pALAF028_hHELZ2L-3xMYC_WB1*: is a derivative of pALAF015_hHELZ2L-3xMYC that contains the E668A mutation in the Walker B motif of the N-terminal helicase domain of HELZ2, which is predicted to inactivate the ATP hydrolysis activity of the helicase domain[93].

*pALAF029_hHELZ2L-3xMYC_WB2*: is a derivative of pALAF015_hHELZ2L-3xMYC that contains the E2361A mutation in the Walker B motif of the C-terminal helicase domain of HELZ2, which is predicted to inactivate the ATP hydrolysis activity of the helicase domain[93].

*pALAF030_hHELZ2L-3xMYC_dRNase*: is a derivative of pALAF015_hHELZ2L-3xMYC that contains the D1346N, D1354N, and D1355N mutations in the RNB domain of HELZ2, which is predicted to inactivate the RNase activity of the RNB domain[92].

*pALAF071_hHELZ2L-3xFLAG*: is a derivative of pALAF015_hHELZ2L-3xMYC where the *3xMYC* epitope tag was replaced with a *3xFLAG* epitope tag.

*pALAF073_hHELZ2L-3xFLAG_dRNase*: is a derivative of pALAF030_hHELZ2L-3xMYC_dRNase where the *3xMYC* epitope tag was replaced with a *3xFLAG* epitope tag.

*psPAX2*: is a lentivirus packaging vector that was a gift from Didier Trono (Addgene plasmid # 12260). The plasmid expresses the HIV-1 gag and pol proteins.

*pMD2.G*: is a lentivirus envelope expression vector that was a gift from Didier Trono (Addgene plasmid # 12259). The plasmid expresses a viral envelope protein and the vesicular stomatitis virus G glyco-protein (VSV-G).

*pcDNA6*: was described previously[97]. It is a derivative of pcDNA6/TR (Invitrogen, Carlsbad, CA, United States) and contains the *blasticidin S-deaminase* (*BSD*) selectable marker but lacks the TetR gene. This plasmid was made by Dr. John B. Moldovan (University of Michigan Medical School).

*pCMV-3Tag-8-Barr*: is a human *β-Arrestin* expression plasmid. The human *ARRB2* cDNA was cloned into pCMV-3Tag-8 (Agilent Technologies). The plasmid contains three copies of a *FLAG* epitope tag fused in-frame to the 3′ end of the *ARRB2* cDNA. The CMV promoter drives *ARRB2-3xFLAG* expression.

*Alu-neo^Tet*: was described previously[96]. The plasmid contains an AluY element with the *neo*^Tet retrotransposition indicator cassette, which was inserted upstream of the Alu poly(A) tract. Alu expression is augmented by a 7SL promoter.

*pTMO2F3_Alu*: is a plasmid that co-expresses Alu and a monocistronic version of L1 ORF2p that contains the L1 5′UTR. The monocistronic *ORF2* coding sequence contains three copies of an in-frame *FLAG* epitope tag sequence at its 3′ end; the CMV promoter augments the expression of *ORF2-3xFLAG*. The plasmid also contains an AluY element whose expression is driven by a 7SL promoter. The Alu element contains the *neo*^Tet retrotransposition indicator cassette[96], which was inserted upstream of the Alu poly(dA) tract. This arrangement allows the quantification of Alu retrotransposition efficiency by counting the resultant number of G418-resistant foci. This plasmid lacks the polypurine sequence in the L1 3′UTR.

*pTMO2F3D145AD702A_Alu*: is identical to pTMO2F3_Alu but contains the D145A and D702A mutations, which inactivate the ORF2p endonuclease and reverse transcriptase activities, respectively.

*pTMO2H3_Alu*: is a derivative of pTMO2F3_Alu plasmid where the *3xFLAG* epitope tag was replaced with three copies of *HA* epitope tag sequence.

*pSBtet-RN*: was a gift from Eric Kowarz[45,118] (Addgene plasmid # 60503). The plasmid contains a firefly luciferase (*Fluc*) gene with an upstream Tet-On inducible promoter.

*pDAO93*: was a gift from Kathleen Burns[45] (Addgene plasmid # 131390). This plasmid is similar to pSBtet-RN but the luciferase gene was replaced with the human L1 *ORFeus* (*ORF1* and *ORF2*) sequence lacking the 5′ or 3′UTR.

*pCMV(CAT)T7-SB100*: was a gift from Zsuzsanna Izsvak[119] (Addgene plasmid # 34879). This plasmid contains a hyperactive variant of the *Sleeping Beauty* transposase, whose expression is driven by the CMV promoter.

## Western blots

HeLa-JVM, U-2 OS, or HEK293T cells were seeded in a 6-well tissue culture plate (Greiner, Frickenhausen, Germany) at $2 \times 10^5$ cells per well. On the following day, the cells were transfected with 1 μg of DNA (1 μg of an L1-expressing plasmid or 0.5 μg of the L1-expressing plasmid and 0.5 μg of either a pCMV-3Tag-8-Barr control or ISG-expressing plasmid) using 3 μL of FuGENE HD transfection reagent (Promega, Madison, WI, United States) and 100 μL of Opti-MEM (Gibco) according to the protocol provided by the manufacturer. The medium was replaced with fresh DMEM approximately 24 h post-transfection (day 1). The cells were harvested using 0.25% (v/v) trypsin (Gibco) at days 2 through 9 post-transfection (depending on the specific experiment). The transfected cells were enriched using 100 μg/mL of hygromycin B (Wako, Osaka, Japan), which was added to the media two days post-transfection and replaced with fresh DMEM containing hygromycin B daily. After collection by trypsinization, the cells were pelleted by centrifugation at $300 \times g$ for 5 min. Then, the cells were washed twice with cold 1× PBS, flash-frozen in liquid nitrogen, and kept at −80 °C.

For cell lysis, the cells were incubated in Radio-ImmunoPrecipitation Assay (RIPA) buffer (10 mM Tris-HCl [pH 7.5], 1 mM EDTA, 1% [v/v] TritonX-100, 0.1% [w/v] sodium deoxycholate, 0.1% [w/v] SDS, 140 mM NaCl, 1× cOmplete EDTA-free protease inhibitor cocktail [Roche, Mannheim, Germany]) at 4 °C for 30 min. The cell debris was pelleted at $12,000 \times g$ for 5 min and the supernatant was collected. The protein concentration was measured using the Protein Assay Dye Reagent Concentrate (Bio-Rad, Richmond, CA, United States) and all of the samples for each experiment were normalized to the same concentration. The protein lysate was mixed at an equal volume with 3x SDS sample buffer (187.5 mM Tris-HCl [pH 6.8], 30%[v/v]) glycerol, 6% [w/v] SDS, 0.3 M DTT, 0.01% [w/v] bromophenol blue) and boiled at 105 °C for 5 min. Twenty micrograms of total protein lysates for all samples were separated using sodium dodecyl sulfate-polyacrylamide gel electrophoresis (SDS-PAGE). Proteins on the gel were transferred onto Immobilon-P, 0.45 μm pore, polyvinylidene difluoride (PVDF) transfer membranes (Merck Millipore, Billerica, MA, United States) using 10 mM CAPS buffer (3-[cyclohexylamino]−1-propanesulfonic acid [pH 11]) in a Mini Trans-Blot Electrophoretic Transfer Cell tank (Bio-Rad) according to protocol provided by the manufacturer. The transfer was performed at 4 °C at 50 V for 16 h. After the transfer was completed, the membrane was incubated with Tris-NaCl-Tween (TNT) buffer (0.1 M Tris-HCl [pH 7.5], 150 mM NaCl, 0.1% [v/v] Tween 20) containing 3% skim milk (Nacalai) for 30 min. The membranes then were washed with TNT buffer, cut into strips, and incubated with the relevant primary antibodies in TNT buffer at 4 °C overnight. The next day, the membranes were washed four times with TNT buffer with five minutes interval at room temperature and incubated with HRP-conjugated secondary antibodies in TNT buffer containing 0.01% (w/v) SDS at room temperature for an hour. The membranes were washed four times with TNT buffer with five minutes interval at room temperature and the signals were detected with the Chemi-Lumi One L (Nacalai) or Chemi-Lumi One Super (Nacalai) chemiluminescence reagent using a LAS-3000 Imager (Fujifilm, Tokyo, Japan), LAS-4000 Imager (Fujifilm), or a FUSION Solo S Imager (Vilber-Lourmat, Marne-la-Vallee, France). Loading controls were run on the same gel, while sample processing controls were run on a separate gel with the same amount of protein loaded from the same samples as indicated in the figure legends. All uncropped blots are available in the Source Data file.

**Primary antibodies and dilutions (in parentheses).** Please note: we tested two different anti-HELZ2 antibodies (Abcam [AB129781] and Affinity Biosciences [DF4285]), but the antibodies were not able to detect the endogenous HELZ2 protein in our experimental conditions.

Mouse monoclonal anti-FLAG M2 antibody (1/5000), (Sigma-Aldrich, F1804, 1.0 mg/mL, RRID: AB_262044)

Rabbit polyclonal anti-FLAG antibody (1/5000), (Sigma-Aldrich, F7425, ~0.8 mg/mL, RRID: AB_439687)

Mouse monoclonal anti-MYC antibody (1/5000), (Cell Signaling Technology, 9B11, RRID: AB_331783)

Rabbit polyclonal anti-PABPC1 antibody (1/5000), (Abcam, ab21060, 0.9 mg/mL, RRID: AB_777008)

Mouse monoclonal anti-GAPDH antibody (1/5000), (Millipore, MAB374, 1.0 mg/mL, RRID: AB_2107445)

Mouse monoclonal anti-Actin antibody (1/5000, diluted to 0.2 times of the original concentration), (Millipore, MAB1501R, RRID: AB_2223041)

Rabbit polyclonal anti-T7-tag antibody (1/5000), (Cell Signaling Technology, D9E1X, RRID: AB_2798161)

Goat polyclonal anti-Luciferase antibody (1/2000), (Promega, G7451, 1.0 mg/mL, RRID: AB_430862)

Mouse monoclonal anti-ORF1p (4H1) antibody (1/2000), (Millipore, MABC1152, 0.5 mg/mL)

Mouse monoclonal anti-eIF3 p110 (B-6) antibody (1/5000), (Santa Cruz Biotechnology, sc-74507, 0.2 mg/mL, RRID: AB_1122487)

**Secondary antibodies and dilutions (in parentheses).** Sheep polyclonal anti-mouse HRP-conjugated Whole antibody (1/5000), (GE Healthcare, NA931-1ML, RRID: AB_772210)

Goat polyclonal anti-rabbit HRP-conjugated Whole antibody (1/5000), (Cell Signaling Technology, 7074, RRID: AB_2099233)

Donkey polyclonal anti-rabbit HRP-conjugated Whole antibody (1/5000), (GE Healthcare, NA934-1ML, RRID: AB_772206)

Donkey polyclonal anti-goat HRP-conjugated Whole antibody (1/5000), (Santa Cruz Biotechnology, sc-2020, 0.4 mg/mL, RRID: AB_631728)

## Immunofluorescence

**Cell transfection and fixation.** HeLa-JVM or U-2 OS cells were plated on 18 mm glass coverslips (Matsunami Glass, Osaka, Japan) coated with Alcian Blue 8GX (Sigma-Aldrich) in 12-well tissue culture plates (Greiner) at $2.5 \times 10^4$ cells per well in DMEM (with 1.0 µg/mL of doxycycline in mCherry-G3BP1-expressing U-2 OS cells). After 24 h, the cells were transfected with 0.5 µg of plasmid DNA (0.5 µg of the L1-expressing plasmid [pJM101/L1.3FLAG, pALAF002, pALAF005, or pALAF008] or 0.25 µg of pJM101/L1.3FLAG and 0.25 µg of either a pCMV-3Tag-8-Barr control or ISG-expression plasmid) using 1.5 µL of FuGENE HD transfection reagent and 50 µL of Opti-MEM according to protocol provided by the manufacturer. Approximately 24 h post-transfection, the medium was replaced with fresh DMEM and 1.0 µg/mL of doxycycline was added into the medium for mCherry-G3BP1-expressing U-2 OS cells. Approximately 48 h post-transfection, the cells were washed with 1× PBS and fixed with 4% paraformaldehyde (PFA) at room temperature for 15 min. Prior to cell fixation, the cells were treated with DMSO (Sigma-Aldrich) or 0.5 mM sodium meta-arsenite (Sigma-Aldrich) for one hour. The fixed cells then were washed with 1× PBS three times and kept at 4 °C until cell permeabilization.

**Immunostaining.** The resultant cells were permeabilized with 0.2% (v/v) Triton X-100 and 0.5% (v/v) normal donkey serum (NDS) for 5 min. The cells were washed once with 1× PBS and twice with PBST (1× PBS and 0.1% [v/v] Tween 20) following permeabilization. The primary antibodies (1/1000 dilution in PBST) containing 0.5% (v/v) NDS were applied onto the coverslip and incubated for 45 min at room temperature. The cells were washed with PBST three times after the primary antibody incubation. The secondary antibodies (1/250 dilution in PBST) containing 0.5% (v/v) NDS and 0.1 µg/mL of 4′,6-diamidino-2-phenylindole (DAPI) were applied onto the coverslip and incubated for 45 min at room temperature. The cells were washed with PBST three times followed by multiple rinses with water. The excess liquid was removed, and the glass coverslips were fixed on glass slides with 3 µL of VECTASHIELD (Vector Laboratories, Burlingame, CA, United States).

**Immunofluorescence.** Images were captured using the DeltaVision Elite microscope with DeltaVision softWoRx 5.5 software (Cytiva, Marlborough, MA, United States). Six z-stack images with 1 µm thickness difference were captured and projected into a single image with the max intensity for each image. For ORF1p-FLAG probed with the Alexa 488-conjugated antibody or MYC-tagged proteins probed with the Cy5-conjugated antibody, the FITC/AF488 or Cy5/AF647 channel was used, respectively. mCherry-G3BP1 fluorescence was detected through the mCherry/AF594 channel. In the ORF1p foci counting experiments, the same signal intensity threshold was applied to all samples and only cells with visible ORF1p signals were counted as positive cells. Only cells that displayed clear cytoplasmic ORF1p signals with foci distinguishable from the background were counted as an L1 foci-positive cells.

## Primary antibodies and dilutions (in parentheses)

Mouse monoclonal anti-FLAG M2 antibody (1/1000), (Sigma-Aldrich, F3165, 3.8–4.2 mg/mL, RRID: AB_259529)

Rabbit polyclonal anti-FLAG antibody (1/1000), (Sigma-Aldrich, F7425, ~0.8 mg/mL, RRID: AB_439687)

Mouse monoclonal anti-MYC antibody (1/1000), (Cell Signaling Technology, 9B11, RRID: AB_331783)

## Secondary antibodies and dilutions (in parentheses)

Donkey anti-mouse polyclonal Alexa Fluor 488 IgG (H + L) (1/250), (Thermo Fisher Scientific, A-21202, 2.0 mg/mL, RRID: AB_141607)

Donkey anti-rabbit polyclonal Alexa Fluor 488 IgG (H + L) (1/250), (Thermo Fisher Scientific, A-21206, 2.0 mg/mL, RRID: AB_2535792)

Goat polyclonal anti-mouse Cy5 (1/250), (Jackson ImmunoResearch Labs, 115-175-146, RRID: AB_2338713)

## Lentiviral transduction

HEK293FT cells were plated in a 10-cm tissue culture dish at $1 \times 10^6$ cells per plate. On the following day, the cells were transfected with 5 µg plasmid DNA (2.5 µg of pALAF012, 1.875 µg of psPAX2, and 0.625 µg of pMD2.G) using 15 µL of 1 mg/mL transfection grade linear polyethylenimine hydrochloride (MW 40,000) (PEI-MAX-40K) (Polysciences, Warrington, PA, United States) in 500 µL of Opti-MEM. Approximately 24 h post-transfection, the medium was replaced with fresh DMEM. The medium containing the virus was collected 48 h post-transfection and filtered through a 0.45 µm polyethersulfone (PES) filter (Merck Millipore).

To generate the inducible mCherry-G3BP1-expressing U-2 OS cell line, $2 \times 10^5$ cells per well were plated in a six-well tissue culture plate. On the next day, the medium was replaced with virus-containing medium supplemented with 8 µg/mL of polybrene (Sigma-Aldrich). Approximately 24 h post-viral treatment, the medium was replaced with fresh DMEM. From the second day post-viral treatment onwards, the media was replaced with fresh DMEM containing 1 µg/mL puromycin every three days until the non-transduced cells were dead.

## Construction of cell lines expressing Tet-On Luciferase and human L1 ORFeus

HeLa-JVM cells were plated in six-well plates at $2 \times 10^5$ cells per well. On the following day, the cells were transfected with 500 ng of plasmid DNA (pSBtet-RN or pDA093) and 50 ng of a sleeping beauty plasmid (pCMV[CAT]T7-SB100) using 2.0 µL of FuGENE HD transfection reagent and 100 µL of Opti-MEM according to the protocol provided by the manufactures. After ~24 h, the medium was replaced with fresh DMEM. G418 (Nacalai) selection (500 µg/mL) began ~48 h post-transfection for 1 week; the G418 containing media was replaced daily. Five percent of the total living cells were transferred into 10-cm tissue culture dishes and the media was replaced daily with 500 µg/mL G418 until the cells reached ~90% confluency. The cells then were trypsinized and resuspended in PBS containing 2% (v/v) FBS and dTomato-positive cells were sorted using a BD FACSAria III flow cytometer with BD FACSDiva Software v.6.1.3 (BD Biosciences, San Jose, CA, United States) to obtain clonal cell lines. Western blotting was used to screen the resultant cell lines for doxycycline dosage-dependent expression of Luciferase or human L1 ORFeus.

## L1 and Alu Retrotransposition Assays

L1 or Alu cultured cell retrotransposition assays were performed as described with modifications[17,54,60,61,96,120].

In retrotransposition assays using the *mneoI* retrotransposition indicator cassette, $2 \times 10^5$ HeLa-JVM or HeLa-HA cells per well were seeded in six-well tissue culture plates. On the following day, the cells were transfected with 1 µg of DNA (0.5 µg of pJM101L1.3/FLAG or its variants and 0.5 µg of phrGFP-C for the L1 retrotransposition assay) or 1 µg of DNA (0.5 µg of pTMO2F3_Alu or phrGFP-C and 0.5 µg of pCMV-3Tag-8-Barr control, pALAF015 [HELZ2], or pALAF024 [MOV10] for the Alu retrotransposition assay) using 3 µL FuGENE HD and 100 µL of Opti-MEM according to the protocol provided by the manufacturer. The

medium was replaced with fresh DMEM (HeLa-JVM) or MEM (HeLa-HA), respectively ~24 h post-transfection (day 1). On day 3 post-transfection, to check transfection efficiency, each duplicate was collected, fixed with 0.5% paraformaldehyde, and subjected to flow cytometry analysis using BD Accuri C6 Plus Flow Cytometer (BD Biosciences). The FITC channel was used to determine the number of hrGFP-expressing cells out of 10,000 cells as a transfection efficiency control. The medium in the remaining transfectants was replaced daily with fresh DMEM or MEM containing 500 μg/mL G418 from day 3 onwards. The resultant colonies were fixed at day 10–14 post-transfection using the fixation solution (1× PBS containing 0.2% [v/v] glutaraldehyde and 2% [v/v] formaldehyde). The cells were stained with 0.1% (w/v) crystal violet. The resultant number of foci were counted and normalized to the transfection efficiency. Please note: the HEK293T cells are G418-resistant and could not be used in *mneoI* based retrotransposition assays.

In retrotransposition assays using the *mblastI* retrotransposition indicator cassette, $5 \times 10^4$ HeLa-JVM cells per well were seeded in six-well tissue culture plates. After ~24 h, the cells were transfected with 1 μg of DNA (0.5 μg of pJJ101/L1.3 and 0.5 μg of an ISG-expressing plasmid or pCMV-3Tag-8-Barr) using 3 μL of FuGENE HD in 100 μL of Opti-MEM. For the viability control, $5 \times 10^3$ HeLa-JVM cells per well were seeded in 6-well tissue culture plates. After ~24 h, the cells were transfected with 1 μg of DNA (0.5 μg of pcDNA6 and 0.5 μg of an ISG-expressing plasmid or pCMV-3Tag-8-Barr) using 3 μL of FuGENE HD in 100 μL of Opti-MEM. Approximately 24 h post-transfection (day 1), the medium was changed with fresh DMEM. Blasticidin selection (10 μg/mL of blasticidin S HCl) began from day 4 post-transfection and the media containing blasticidin was replaced every three days until day 8-10. The resultant colonies were fixed using the fixation solution and stained with 0.1% (w/v) crystal violet. The resultant number of foci were counted and normalized to the resultant number of pcDNA6-transfected foci.

In retrotransposition assays using the *mEGFPI* retrotransposition indicator cassette, $2 \times 10^5$ HeLa-JVM or HEK293T cells per well were seeded in six-well tissue culture plates. On the next day, the cells were transfected with 1 μg of DNA (0.5 μg of cepB-gfp-L1.3 or cepB-gfp-L1.3RT[-] intronless and 0.5 μg of a pCMV-3Tag-8-Barr control or ISG-expressing plasmid) using 3 μL of FuGENE HD in 100 μL of Opti-MEM. Approximately 24 h post-transfection (day 1), the medium was replaced with fresh DMEM. Transfected cells were selected using 10 μg/mL blasticidin S HCl from day 2 post-transfection, changing the media every three days. The cells were collected on day 7–8 post-transfection and the resultant EGFP positive cells were analyzed using BD Accuri C6 Plus Software v.1.0.23.1 (BD Biosciences). The FITC channel was used to count the EGFP positive cells out of 30,000 cells. The number of the EGFP-positive cells was normalized to the transfection efficiency measured by counting the number of cepB-gfp-L1.3RT(-) intronless GFP-positive cells.

## siRNA treatment

HeLa-JVM cells were plated in six-well tissue culture plates at $1 \times 10^5$ cells per well. After ~24 h, 25 nM of a Dharmacon siRNA mixture (non-targeting control: ON-TARGETplus Non-targeting Pool, D-001810-10-0020; HELZ2: ON-TARGETplus HELZ2 siRNA SMARTpool, L-019109-00-0005; or MOV10: ON-TARGETplus MOV10 siRNA SMARTpool, L-014162-00-0005) were transfected using 3.75 μL of Lipofectamine RNAiMAX (Thermo Fisher Scientific, Waltham, MA, United States). Approximately 24 h post-siRNA treatment (day 1), the medium was replaced with fresh DMEM and the cells were transfected with 0.5 μg of cepB-gfp-L1.3 or cepB-gfp-L1.3RT(-) intronless using 1.5 μL of FuGENE HD in 100 μL of Opti-MEM. Transfected cells were selected using 10 μg/mL blasticidin S HCl from day 3 post-transfection with media changes every three days. On day 8 post-transfection, the cells were harvested, washed with cold 1× PBS twice, and analyzed for EGFP

expression using BD Accuri C6 Plus Flow Cytometer out of 30,000 cells. The number of the EGFP-positive cells was normalized to the transfection efficiency measured by counting the number of cepB-gfp-L1.3RT(-) intronless GFP-positive cells.

## Immunoprecipitation of L1 ORF1p

**Immunoprecipitation for IP-MS.** HeLa-JVM cells were plated in 15 cm tissue culture dishes containing DMEM medium at $2.5 \times 10^6$ cells per dish. Three 15 cm tissue culture dishes were used for each sample preparation. After ~24 h, the cells were transfected with 10 μg of an L1-expressing plasmid (pJM101/L1.3, pJM101/L1.3FLAG, or pALAF008) using 30 μL of FuGENE HD (Promega) in 1000 μL of Opti-MEM. On the following day (day 1), the medium was replaced with fresh DMEM. From day 2 post-transfection onwards, the medium was replaced daily with fresh DMEM containing 100 μg/ml hygromycin B. On day 6 post-transfection, the cells were harvested using trypsin, washed with 1× cold PBS twice, flash-frozen with liquid nitrogen, and stored at −80 °C.

For IP reactions, one hundred fifty microliters of Dynabeads Protein G (Invitrogen) was washed twice with PBS containing 0.5% (w/v) BSA and 0.1% (v/v) Triton X-100. For each sample, the beads were incubated with 15 μg of mouse monoclonal anti-FLAG M2 antibody (Sigma-Aldrich, F1804, RRID: AB_262044) in 1 mL of PBS containing 0.5% (w/v) BSA and 0.1% (v/v) Triton X-100 at 4 °C for 2 h. After incubation, the antibody-conjugated beads were washed with PBS containing 0.5% (w/v) BSA and 0.1% (v/v) Triton X-100 twice. The beads were resuspended in Lysis150 buffer (20 mM Tris-HCl [pH 7.5], 2.5 mM MgCl$_2$, 150 mM KCl, 0.5% (v/v) IGEPAL CA-630, 1 mM DTT) containing 0.2 mM phenylmethylsulfonyl fluoride (PMSF) and 1× cOmplete EDTA-free protease inhibitor cocktail before immunoprecipitation. Each cell pellet was lysed using the Lysis150 buffer containing 0.2 mM PMSF and 1× cOmplete EDTA-free protease inhibitor cocktail. The resuspended cell pellets were incubated at 4 °C for 30 min and centrifuged at $12,000 \times g$ for 5 min to pellet the cell debris. The supernatant was collected and incubated with antibody non-conjugated Dynabeads Protein G at 4 °C for 2 h with gentle rotation to remove non-specific protein binding. The Dynabeads were removed and the protein concentration in the pre-cleared cell lysates was quantified using Protein Assay Dye Reagent Concentrate. The same total amount of protein was used for each immunoprecipitation. Dynabeads Protein G conjugated to the anti-FLAG antibody was added to the supernatant and incubated at 4 °C for 3 h with gentle rotation. The beads were then washed five times with 200 μL of the Lysis150 buffer. The ORF1p-FLAG protein complex bound was eluted using 200 μg/mL of 3xFLAG peptide (Sigma-Aldrich) in the Lysis150 buffer containing 0.2 mM PMSF and 1× cOmplete EDTA-free protease inhibitor cocktail by incubation at 4 °C for 1 h with gentle rotation. This step was repeated once, and the protein was precipitated overnight by adding three times the volume of cold acetone to the resultant eluate. The protein was pelleted at $12,000 \times g$ at 4 °C for 30 min, resuspended in 1× SDS sample buffer and boiled at 105 °C for 5 min.

**Immunoprecipitation for western blotting.** HEK293T cells were plated in 10 cm tissue culture dishes at $3 \times 10^6$ cells per dish. Approximately 24 h after plating, the cells were transfected with 4 μg of pJM101/L1.3FLAG or pJM101/L1.3 and 2 μg of ISG-expressing plasmid (pALAF015, pALAF016, pALAF021, pALAF022, pALAF023, or pALAF024) using 18 μL of 1 mg/mL PEI-MAX-40K in 500 μL of Opti-MEM. Approximately 24 h post-transfection, the media was changed with fresh DMEM. From day 2 post-transfection onwards, the medium was replaced daily with fresh DMEM containing 100 μg/ml hygromycin B. On day 4 post-transfection, the cells were harvested with pipetting, washed with 1× cold PBS twice, flash-frozen with liquid nitrogen, and stored at −80 °C for subsequent experiments.

For each sample, ten microliters of the Dynabeads Protein G were incubated with 1 µg of anti-FLAG M2 antibody in 50 µL of PBS containing 0.5% (w/v) BSA and 0.1% (v/v) Triton X-100 at 4 °C for 2 h. After incubation, the antibody-conjugated beads were washed with PBS containing 0.5% (w/v) BSA and 0.1% (v/v) Triton X-100 twice. The beads were resuspended in Lysis150 buffer containing 0.2 mM PMSF and 1× cOmplete EDTA-free protease inhibitor cocktail before immunoprecipitation. Each cell pellet was lysed in 500 µL of the Lysis150 buffer containing 0.2 mM PMSF and 1× cOmplete EDTA-free protease inhibitor cocktail. The resuspended cell pellets were incubated at 4 °C for 1 h and centrifuged at 12,000 × $g$ for 5 min to pellet the cell debris. The supernatant was collected and 10 µL of the supernatant was saved as input. Anti-FLAG antibody-conjugated Dynabeads were added to the samples and incubated at 4 °C for 4 h with gentle rotation.

The RNase treatment for HELZ2-expressed samples was performed after removal of the cell lysate using 20 µg/mL of RNase A (Nippongene, Tokyo, Japan) in 100 µL of the Lysis150 buffer for five minutes at 37 °C. The beads then were washed four times with 100 µL of the Lysis150 buffer. The beads were resuspended directly in 1× SDS sample buffer and boiled at 105 °C for 5 min except for the HELZ2-expressed samples, where the ORF1p-FLAG protein complex was eluted using 20 µL of the Lysis150 buffer containing 0.2 mM PMSF, 1× cOmplete EDTA-free protease inhibitor cocktail, and 200 µg/mL 3xFLAG peptide by incubation at 4 °C for 1 h with gentle rotation. The eluted protein was resuspended in 1× SDS sample buffer and boiled at 105 °C for 5 min.

## Label-free quantification (LFQ) of LC-MS/MS results

Mass spectrometry analysis was performed by the proteomics facility in the Graduate School of Biostudies at Kyoto University. After SDS-PAGE and visualization of the gel using PlusOne Silver Staining Kit, Protein (Cytiva) according to the protocol provided by the manufacturer, the entire gel lane from each sample was excised into 15 components. The silver stain was then removed, and the excised gel slices were incubated with sequencing-grade modified trypsin (Promega) to extract the peptides. The purified peptides then were subjected to liquid chromatography-tandem mass spectrometry (LC-MS/MS) on nano-Advance (AMR, Tokyo, Japan) and Q Exactive Plus (Thermo Fisher Scientific) using Xcalibur 3.1 (Thermo Fisher Scientific), Paradigm Home v.2.0.4 R4 B22 (Bruker Daltonics, Billerica, MA, United States), and Cycle Composer v.1.6.0 (CTC Analytics AG, Zwingen, Switzerland) for mass spectrometry acquisition. LFQ analyses on the resultant datasets were performed using Proteome Discoverer 2.3 (Thermo Fisher Scientific) with the peptide hits identified in the No Tag (pJM101/L1.3), WT (pJM101/L1.3FLAG), and M8 (RBM) (pALAF008) samples (see Source data for more details). Data are available via ProteomeXchange with identifier PXD038851. Briefly, the human Uniprot Knowledgebase (UniProtKB: https://www.uniprot.org/help/uniprotkb) database was used for protein identification and the Mascot Server 2.7.0 database (Matrix Science: https://www.matrixscience.com) was used as the search engine. The Protein Validator Node of Proteome Discoverer 2.3 calculated high (<0.01), medium (0.01≤ and <0.05), or low (0.05≤) false discovery rates (FDRs) with the peptide hits to generate the protein FDR confidence score. Both unique and razor peptides were used for identification of the best associated protein group with those peptides in the analysis. Razor peptides are shared in multiple protein groups and assigned to the protein group with the largest number of total peptides when combined with the unique peptides. Triplicate data from 15 gel strips of WT L1 and M8 (RBM) L1 protein lists were grouped respectively to obtain each group abundance using the Precursor Ions Quantifier nodes of Proteome Discoverer 2.3. The abundances were normalized with the ORF1p peptides. This analysis was followed by a comparison of the grouped abundances between WT L1 (grouped) vs. M8 (RBM) L1 (grouped) to calculate the abundance ratio, where the upper and lower limits of the

ratios were set to 1000 and 0.001, respectively. The $p$-values of the abundance ratios were calculated using the Tukey Honestly Significant Difference test (post hoc) after an analysis of variance (ANOVA) test. The volcano plot depicts the resultant $\log_2$ abundance ratios (WT ORF1p-FLAG vs. M8 ORF1p-FLAG) on the $x$-axis and the $-\log_{10}$ $p$-values of the abundance ratios on the $y$-axis. We used a threshold of >0.5 for $\log_2$ abundance ratios for the GO term analysis (Figs. 2d and 3b, see Source data).

## ORF1p crystal structure analysis

The crystal structure images of ORF1p and the mutations were created using UCSF ChimeraX software 1.2.5 for Windows[121] based on the 2ykp pdb file[12].

## GO term analysis

The proteins were first filtered before the analyses; we removed protein groups with medium and low FDR confidence scores and those that lacked detectable peptide peaks in either the WT ORF1p-FLAG or M8 ORF1p-FLAG samples. The remaining proteins with UniProt accession numbers were converted to official gene symbols using the UniProt Retrieve/ID mapping tool (https://www.uniprot.org/id-mapping); the unmapped UniProt accession numbers were converted to gene symbols manually. UniProt accession numbers that do not map to any gene symbols were excluded from the analysis. In the case of different UniProt accession numbers that map to the same gene symbol, only the UniProt accession number with the highest $\log_2$ abundance ratio value (WT vs. M8 [RBM] L1) was included in the analysis. This filtration process resulted in a total number of 1437 genes (see Source data, Comparisons tab). Among the 1437 genes, genes with values of a >0.5 $\log_2$ abundance ratio (WT vs. M8 [RBM] L1) were used in the DAVID[55,56] 2021 gene ontology (https://david.ncifcrf.gov/) analyses to obtain the "Functional annotations of UniProt Keyword Biological Processes" GO terms that are shown in Fig. 2d and Supplementary Data 1.

## GSEA preranked analysis

GSEA 4.2.3 for Windows software was used for the analysis[57] (http://www.broad.mit.edu/GSEA). The 1437 genes described in the GO term analysis paragraph were included in the GSEA Preranked analysis using the $\log_2$ abundance ratios of WT vs. M8 (RBM) of the respective protein hits (see Source data, Comparisons tab). The GSEA preranked analysis was performed using the hallmark gene sets from GSEA Molecular Signatures Database v7.5.1 on Human Gene Symbol with Remapping v7.5 Chip platform.

## ImageJ quantification of western blot band intensity

Using the ImageJ 1.5.2a for Windows software tool[122], identical sized rectangles were drawn for each band. The area of intensity of the bands were generated using Plot Lanes function and calculated using a wand (tracing) tool. The intensity of each ORF1p-T7 band was normalized to that of the GAPDH band with respective samples. The values were displayed as ratios in comparison to the leftmost band in the western blot image (pTMF3 and pCMV-3Tag-8-Barr control co-transfected cells).

## Bio-Plex cytokine assay

To collect culture supernatants, $2 \times 10^5$ HEK293T cells per well were seeded in six-well plates. Approximately 24 h after seeding, the cells were transfected with 1 µg of an L1-expressing plasmid or pCEP4 using 3 µL of FuGENE HD transfection reagent and 100 µL of Opti-MEM according to the manufacturer's instruction. The media was replaced with fresh DMEM at -24 h (day 1) and -72 h (day 3) post-transfection. The culture supernatants were collected at -96 h (24 h post-day 3 media change) and -120 h (48 h post-day 3 media change) post-transfection. For the polyinosinic:polycytidylic acid (poly[I:C])

transfection, HEK293T cells were transfected with 5 μg/mL of High Molecular Weight Poly(I:C) (InvivoGen, San Diego, California, United States) using 3.75 μL of Lipofectamine RNAiMax (Thermo Fisher Scientific) in 1 mL of culture media in 6-well plates. The culture supernatants were collected at ~24 h post poly(I:C) transfection. All culture supernatants were centrifuged at $500 \times g$ for 5 min to remove cell debris, flash-frozen in liquid nitrogen, and stored at −80 °C. The same batch of DMEM was used in all cell cultures for this cytokine assay. Bio-Plex 200, a multiplex cytokine array system (Bio-Rad), was used to quantify the basal levels of cytokines in DMEM medium and the secreted cytokines and chemokines in the collected culture supernatants according to the protocol provided by the manufacturer. The Bio-Plex Pro Human Inflammation Panel 1 37-Plex includes 37 cytokines and chemokines (APRIL, BAFF, CD30, CD163, Chitinase-3, gp130, IFN-α2, IFN-β, IFN-γ, IL-2, IL-6Ra, IL-8, IL-10, IL-11, IL-12 (p40), IL-12 (p70), IL-19, IL-20, IL-22, IL-26, IL-27 (p28), IL-28A, IL-29, IL-32, IL-34, IL-35, LIGHT, MMP-1, MMP-2, MMP-3, Osteocalcin, Osteopontin, Pentraxin-3, TNF-R1, TNF-R2, TSLP, TWEAK). Data acquisition and analyses were performed using Bio-Plex Manager software version 5.0 (Bio-Rad).

## RNA extraction and RT-qPCR

HeLa-JVM or HEK293T at $2 \times 10^5$ cells per well were seeded in 6-well tissue culture plates. On the following day, the cells were transfected with 1 μg of DNA (1 μg of an L1-expressing plasmid or 0.5 μg of the L1-expressing plasmid and 0.5 μg of a pCMV-3Tag-8-Barr control or an ISG-expressing plasmid). Approximately 24 h post-transfection (day 1), the medium was replaced with fresh DMEM. On day 2 (HeLa-JVM and HeLa-HA) or day 4 (HEK293T) post-transfection, the cells were washed with 1× PBS and 0.9 mL TRIzol was added directly to each well. The RNA extractions were performed according to the protocol provided by the manufacturer. The cells were lysed with TRIzol and transferred into new 1.5 mL tubes. One hundred eighty microliters of chloroform was added into each tube and shaken vigorously for 15 s. After incubation at room temperature for 5 min, the samples were centrifuged at $12,000 \times g$ for 15 min at 4 °C. Three hundred sixty microliters of the upper layer were transferred into a new 1.5 mL tube and 400 μL of 100% isopropanol was added to precipitate the RNA. The samples were incubated at room temperature for 10 min. Next, RNA was pelleted at $12,000 \times g$ for 30 min. The purified RNA then was washed with 75% cold ethanol and centrifuged at $10,000 \times g$ for 5 min. The RNA pellet was dried at room temperature. Once dried, 30 μL of RNase-free $H_2O$ was added and incubated at 55 °C for 10 min to dissolve RNA. The resultant RNA was then treated with RNase-free DNase Set (QIAGEN) according to the protocol provided by the manufacturer with some minor modifications. Five microliters of DNase I (15 K units, TaKaRa Bio), 0.2 U/μL of ribonuclease inhibitor (porcine liver) (TaKaRa Bio) in 44.5 μL of the RNase-free Buffer RDD was added to each sample. The samples were incubated at room temperature for 15 min and the RNA then was pelleted after ethanol precipitation (incubation at −20 °C overnight in 240 μL of 100% ethanol and 8 μL of 3 M NaOAc [pH 5.2]). The RNA pellets were washed with 75% cold ethanol, dried at room temperature, resuspended in RNase-free water, and incubated at 75 °C for 10 min to inactivate the DNase I. One microgram of total RNA was used as a template in reverse transcription reactions using 0.2 mM dNTP (TaKaRa Bio), 1 U/μL ribonuclease inhibitor (porcine liver) (TaKaRa Bio), 0.25 U/μL AMV reverse transcriptase XL (TaKaRa Bio), and 0.125 μM of an oligo (dT) primer (Invitrogen) according to the protocol provided by the manufacturer unless stated otherwise. Two negative controls were included for all instances: no reverse transcriptase (reverse transcriptase was excluded during cDNA synthesis) and no template (cDNA was replaced with RNase-free water). The reverse transcription reaction was performed as follows: 30 °C for 10 min, 42 °C for 30 min, and 95 °C for 5 min. Prime Script MMLV reverse transcriptase (TaKaRa Bio) and 0.125 μM of the oligo (dT) primer for RNA-IP experiments (see below) or a HELZ2 specific primer

(HELZ2_R) for HELZ2 RNA quantification were used to reverse transcribe instead. RNA was incubated at 65 °C for 5 min before the addition of Prime Script MMLV reverse transcriptase and the reverse transcription was performed as follows: 42 °C for 60 min followed by 70 °C for 15 min. RT-qPCR was performed using Luna Universal qPCR Master Mix (New England Biolabs). Amplification was performed using StepOnePlus Real-Time PCR System (Applied Biosystems) using the following parameters: 15 s at 95 °C; followed by 40 cycles of denaturation (95 °C for 15 s) and amplification (60 °C for 60 s). Technical duplicates were made for each sample. Quantification of cDNA for each reaction was determined by comparing the cycle threshold (Ct) with a standard curve generated from one of the samples using StepOne Software v2.2. All Ct readings fall within the range of the standard curve generated.

**Primers used for RT-qPCR.** HLTF_F: 5′-GTGCATGCTGCAGTACAGA-3′
   HLTF_R: 5′-GCTGTTCCCAGAATGGTGGA-3′
   SMC2_F: 5′-GCTTTTTGCTGGGCATCTCC-3′
   SMC2_R: 5′-ACCAGCCTGCCCATTTTTGT-3′
   L1 (SV40)_F: 5′-TCCAGACATGATAAGATACATTGATGAG-3′
   L1 (SV40)_R: 5′-GCAATAGCATCACAAATTTCACAAA-3′
   Luciferase_F: 5′-CGAGGCTACAAACGCTCTCA-3′
   Luciferase_R: 5′-CAGGATGCTCTCCAGTTCGG-3′
   IFN-α _F: 5′-CTGAATGACTTGGAAGCCTG-3′
   IFN-α _R: 5′-ATTTCTGCTCTGACAACCTC-3′
   HELZ2_F: 5′-GAGAAGGTGGTTCTTCTCGGAG-3′
   HELZ2_R: 5′-CTCATGCATGCGGTACTGAG-3′
   MOV10_F: 5′-CGTACCGGAAACAGGTGGAG-3′
   MOV10_R: 5′- TGAACCCACCTTCAAGTCCTTG-3′
   *mneoI* (Alu or L1)_F: 5′- ACCGGACAGGTCGGTCTTG-3′
   *mneoI* (Alu or L1)_R: 5′- CTGGGCACAACAGACAATCG-3′
   Beta-actin_F: 5′-CCTTTTTTGTCCCCCAACTTG-3′
   Beta-actin_R: 5′-TGGCTGCCTCCACCCA-3′
   GAPDH_F: 5′-GGAGTCCCTGCCACACTCAG-3′
   GAPDH_R: 5′-GGTCTACATGGCAACTGTGAGG-3′
   Oligo (dT): 5′-TTTTTTTTTTTTTTTTTTTTTVN-3′

## RNA-IP

RNA immunoprecipitation (RNA-IP) experiments were carried out as described previously with some modifications[27]. HeLa-JVM cells were plated in 10 cm tissue culture dishes at $1.5 \times 10^6$ cells per dish. On the following day (day 0), the cells were transfected with 5 μg of plasmid DNA (pJM101/L1.3, pJM101/L1.3FLAG or pALAF008_M8) using 15 μL of PEI-MAX-40K in 500 μL of Opti-MEM. Approximately 24 h post-transfection (day 1), the medium was replaced with fresh DMEM. On the following day (day 2), the medium was replaced daily with fresh DMEM containing 100 μg/mL hygromycin B and the cells were collected at day 5 post-transfection. The whole cell extracts were prepared by incubation in the Lysis150 buffer containing 0.2 mM PMSF and 1× cOmplete EDTA-free protease inhibitor cocktail for one hour at 4 °C. The lysate was separated from the insoluble fraction by centrifugation at $12,000 \times g$ for five minutes and transferred to a new tube. Ten microliters of the lysate were saved as the input fraction. Prior to immunoprecipitation, the anti-FLAG antibody-conjugated beads were prepared as described in "immunoprecipitation and western blotting" section of the Methods. The cleared lysate (input) was incubated with the anti-FLAG antibody-conjugated beads for 5 h at 4 °C. The beads were then washed four times with 150 μL of Lysis150 buffer without protease inhibitors. The RNA extraction was performed as described in "RNA extraction and RT-qPCR" in the Methods section with a slight modification: 200 μg/mL glycogen was added to the immunoprecipitated RNA fraction before ethanol precipitation. All of the RNA samples were resuspended in 30 μL of RNase-free water. Five microliters (one sixth) of the extracted RNA from the input and IP fractions were used to synthesize cDNA using PrimeScript MMLV reverse

transcriptase as described in the previous section. The ORF1p-associated RNA values were calculated by dividing the cDNA amount in the IP fraction by that in the input.

### In vitro RNase assay

The RNase assay was performed based on RNase II assay by Barbas A., et al.[92] with several modifications. To produce HELZ2-3xFLAG proteins, HEK293T cells were seeded on two 10-cm dishes at ~$5 \times 10^6$ cells per dish in DMEM. Approximately 24 h after cell seeding, the cells were transfected with 10 µg of plasmid pCEP4, pALAF071 (HELZ2-3xFLAG), or pALAF073 (HELZ2-3xFLAG_dRNase), which was preincubated in 500 µL of Opti-MEM with 30 µL of PEI-MAX-40K (1 mg/mL) for 10 min at room temperature. The medium was replaced with fresh DMEM ~24 h post-transfection. Approximately 48 h post-transfection, the transfected cells were collected, washed with 1x cold PBS, flash-frozen with liquid nitrogen, and stored at −80 °C.

To purify the recombinant proteins, 20 µL of Dynabeads Protein G was washed twice with PBS containing 0.5% (w/v) BSA and 0.1% (v/v) Triton X-100 followed by conjugation with 2 µg of mouse monoclonal anti-FLAG M2 antibody (Sigma-Aldrich, F1804) in PBS containing 0.5% (w/v) BSA and 0.1% (v/v) Triton X-100 at 4 °C for 1 h. After conjugation, the beads were washed with PBS containing 0.5% (w/v) BSA and 0.1% (v/v) Triton X-100 twice and resuspended in RIPA buffer (10 mM Tris-HCl [pH 7.5], 1 mM EDTA, 1% [v/v] TritonX-100, 0.1% [w/v] sodium deoxycholate, 0.1% SDS [w/v], 140 mM NaCl). Each frozen cell pellet was lysed using 1 mL RIPA buffer containing 0.2 mM PMSF and 1× cOmplete EDTA-free protease inhibitor cocktail. The resuspended cell pellets were incubated at 4 °C for 1 h and centrifuged at $12,000 \times g$ for 5 min to pellet the cell debris. The supernatant (~1 mL) was collected and incubated with anti-FLAG antibody-conjugated Dynabeads Protein G at 4 °C for 3 h with gentle rotation. The beads were then washed five times with 200 µL of RIPA buffer. The HELZ2-3xFLAG protein was eluted using 20 µL of 200 µg/mL of 3xFLAG peptide (Sigma-Aldrich) in RIPA buffer containing 0.2 mM PMSF and 1× cOmplete EDTA-free protease inhibitor cocktail by incubation at 4 °C for 1 h with gentle rotation. The eluate fraction (~20 µL) was mixed with 40 µL of TBST (20 mMTris-HCl [pH 8], 150 mM NaCl, 0.05% [v/v] Tween 20) containing 90% glycerol, subjected to SDS-PAGE, visualized by PlusOne Silver Staining Kit, and analyzed by western blotting using an anti-FLAG antibody (Sigma-Aldrich, F1804).

For the detection of 3′ to 5′ RNase activity, 2 µL of the purified HELZ2-3xFLAG or HELZ2 3xFLAG_dRNase were incubated with a single-strand poly(A)$_{30}$ RNA oligonucleotide labeled with IRDye800 at its 5′ end (poly[rA30], Integrated DNA Technologies [IDT]) for 0, 5, 10, and 60 min at 37 °C in 50 µL of the RNase buffer (20 mM Tris-HCl [pH 8], 100 mM KCl, 1 mM MgCl2, 1 mM DTT, 6 nM poly[rA30]). Only the labeled probe without recombinant protein served as a control to indicate the full-length poly(A)$_{30}$ RNA. Fifty microliters of 2× RNA loading dye (47.5% [v/v] formamide, 20 mM EDTA, 0.1% [w/v] Orange G) were added to the reaction and the resultant single-stranded (ss) RNA products were separated in a 5% acrylamide/TBE gel (45 mM Tris-HCl [pH 7.5], 45 mM boric acid, 2 mM EDTA, 5% acrylamide/bisacrylamide [37.5:1]). The gel image was captured by an Odyssey CLx imaging system (LI-COR Biosciences, Lincoln, NE, United States).

### Statistics and reproducibility

All western blots and immunofluorescence were independently replicated three times to ensure reproducibility. The RNase assay experiment in Supplementary Fig. 5c, d was performed twice with similar results observed. The rest of the experimental replicate numbers are indicated in the figure legends. One-way ANOVA followed by Bonferroni-Holm post hoc tests were performed for all statistical analyses unless stated otherwise in the figure legends. All analyses were performed using online website statistical calculator ASTATSA 2016 (https://www.astatsa.com/) or GraphPad Prism version 9.0.0 for

Windows (GraphPad Software, San Diego, California, United States; www.graphpad.com). The numbers of biological replicates are indicated in the figure legends. Where applicable, data are always shown as the mean ± standard errors of the means (SEM). The exact $p$-value of each pair was indicated in the figure legends. ns: not significant; * $p < 0.05$; ** $p < 0.01$; *** $p < 0.001$.

### Reporting summary

Further information on research design is available in the Nature Portfolio Reporting Summary linked to this article.

## Data availability

The crystal structure images of ORF1p are based on the 2ykp pdb file (https://www.rcsb.org/structure/2ykp). Uniprot database (https://www.uniprot.org/help/uniprotkb) was used for protein identification from the mass spectrometry data and the Mascot Server 2.7.0 database (https://www.matrixscience.com) was used as the search engine. The mass spectrometry proteomics data have been deposited to the ProteomeXchange Consortium via the PRIDE [1] partner repository with the dataset identifier PXD038851 and 10.6019/PXD038851. The mass spectrometry data will also be available at jPOST repository (https://repository.jpostdb.org/) with the accession number PXD032869 and PXD036759. Preranked GSEA analysis was performed using GSEA Molecular Signatures Database (MSigDB: https://www.gsea-msigdb.org/gsea/msigdb/), STRING database v11.5 (https://string-db.org/) was used for STRING analysis, and ISG screening was done using the interferome database v2.0 (www.interferome.org). The data supporting the findings of this study are available from the corresponding authors upon reasonable request. Source data including analyzed mass spectrometry data and uncropped western blot images are available in a Source Data file. Source data are provided with this paper.

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

## Acknowledgements

We thank K. H. Burns, D. Ardeljan, T. Heidmann, M. T. Hayashi, D. Trono, and Z. Izsvak for valuable reagents, all Ishikawa lab members (especially Theventhiran, T. Makino, K. Sugino, and K. Nishimori), K. Takahara, M. Miyoshi, and J. B. Moldovan for helpful discussions. A.L.-F. was supported by JASSO and MEXT Scholarships. F.I. was supported by JSPS KAKENHI (Grant Number JP19H05655). J.V.M. was supported, in part, by NIH grant GM060518. T.M. was supported by JSPS KAKENHI (Grant Number JP18K06180 and 21K19219), ISHIZUE 2021 of Kyoto University Research Development Programs, and research grants from the Takeda Science Foundation, the Japan Foundation for Applied Enzymology, the Sumitomo Foundation for Basic Science Research Projects, and Astellas Foundation for Research on Metabolic Disorders. A part of this study was conducted through the CORE Program of the Radiation Biology Center, Kyoto University and was supported by the Core-to-Core Program, JSPS.

## Author contributions

A.L.-F., J.V.M., and T.M. conceived and designed the experiments, analyzed data, and prepared the manuscript. A.L.-F. and T.M. performed experiments. Y.W. provided technical support and performed mass spectrometry. K.U. conducted the Bio-Plex cytokine assay. F.I., J.V.M., and T.M. contributed to critical discussions, writing, and editing the manuscript. All authors contributed to ideas.

## Competing interests

J.V.M. is an inventor on patent US6150160, is a paid consultant for Gilead Sciences, serves on the scientific advisory board to Tessera Therapeutics Inc. (where he is paid as a consultant and has equity options), has licensed reagents to Merck Pharmaceutical, and recently served on the American Society of Human Genetics Board of Directors. The other authors declare no competing interests.
