## [Peer review file · Nature Communications]

The interferon stimulated gene-encoded protein HELZ2 inhibits human LINE-1 retrotransposition and LINE-1 RNA-mediated type I interferon inductionREVIEWER COMMENTS

Reviewer #1 (Remarks to the Author):

The interferon stimulated gene-encoded protein HELZ2 inhibits human LINE-1 retrotransposition and LINE-1 RNA-mediated type I interferon induction

Luqman-Fatah et al.

In this elegant work, the authors identify novel L1 regulators by employing an RNA binding mutant (M8 RBM) of L1 ORF1 protein, in parallel to the wildtype L1 ORF1 protein to map the interactome of L1 ribonucleoprotein particles in an RNA-dependent manner. Both versions of the L1 ORF1 protein are equally expressed in cell lines but the M8 RBM fails to bind RNA, is not able to form RNP foci in cytoplasm and leads to greatly reduced engineered L1 retrotransposition efficiency. The authors perform IP-Mass spec with both L1 ORF1p constructs and find an enrichment of proteins indicative of interferon signalling specifically with the wild type protein, implying that their interaction with ORF1p is dependent on RNP foci formation/RNA binding. The interactome includes 5 ISGs not previously known to interact with L1 RNPs and the authors go on to confirm all these ORF1 binding partners by co-IP. The mechanism of how these ISGs regulate L1s is confirmed by assessing engineered L1s in their retrotransposition efficiency, ORF1p levels, L1 RNA and RNP foci formation. Data suggests that HELZ2 blocks L1 activity by acting on L1 RNA through its decreased stability or digestion. Further analyses identifies the helicase domains (particularly helicase 2) to be crucial to the anti-LINE-1 function of HELZ2 and the L1 5'UTR to be the target sequence of HELZ2 recognition. This is a remarkable, thorough and important work, relevant to scientists in diverse fields of research ranging from genome instability to immunology and I have almost no comments.

Minor comments

1.Line 89: It should be clarified at this point in the text that the WT ORF1 protein only and not the M8 RBM ORF1p associated with the cohort of ISGs identified.

2.Line 178: This sentence would be clearer if the word 'L1' was added after 'WT ORF1p-FLAG' : 'conducted with WT ORF1p-FLAG cell extracts, but was severely reduced in IP reactions conducted with M8/RBM ORF1p-FLAG L1 cell extracts'.

3.Line 322: This sentence is a bit unclear: 'The WA1 mutant demonstrated a low, but not statistically significant decrease in L1 retrotransposition efficiency'. Could be rephrased i.e. 'The WA1 mutant was still able to block retrotransposition almost as effectively as WT HELZ2'

Line 331 is also unclear: 'In general, the WA2 and WB2 mutants consistently exhibited a less severe inhibition of L1 retrotransposition'. Could change to 'whereas the WA2 and WB2 mutants were impaired in their ability to block retrotransposition' or similar.

4.Figure 1: The use of the mcherry-G3BP1 construct in this experiment was not well explained.

5.Figure 2: It would be interesting to know here what the genes are corresponding to these gene ontology terms (viral transcription and interferon-alpha/gamma response) in supplementary?

6.Figure 3a: From the legend it appears that L1 regulator candidates that were ISGs were focused on amongst the top 300 mass spec. candidates. Can a ranking be given to these candidates in the table to show how high up they appeared in the screen depending on the no. of peptides scored for each of these proteins? All the 300 protein hits should be listed in a supp table, included with the manuscript.

7. Figure 6E: It would be interesting to know if HELZ2 recognizes the L1 5'UTR through its RNA binding/helicase or other domain? However, this is such a complete study, it could be a discussion point.

8. Figure 7: In the model, I think the exclamation marks can be removed, especially as they detract from the main message of the paper (which is about novel ISGs not RNA sensors).

9. Supp Figure 5: It is interesting that some residues of HELZ2 are conserved in yeast. Since yeast do not have LINE-1 elements, the authors should comment on the possible role of this host protein in yeast in the discussion.

10. It would be interesting to note in the discussion that the main function of LINE-1 ORF1p may be to bind and shield L1 RNA from host ISG proteins, including RNA sensors, thereby explaining its requirement for retrotransposition. This is implied in the abstract but not really mentioned further. Also, IFN-alpha is mentioned several times but not IFN-beta, which is a fundamental type I IFN so it would be more appropriate to refer to 'type I IFNs' in general in the discussion.

Reviewer #2 (Remarks to the Author):

See attached text doc "review"

This manuscript explores the relationship between L1 RNPs and some interferon stimulated genes that are likely to be host-defenses against L1 expression. In the manuscript, the candidate HELZ2 is examined most closely and characterized as a candidate L1 RNA binder in the 5'UTR. A relationship between L1 expression / HELZ2 expression / and interferon response is demonstrated. Precise details of the chain of events, post-L1 RNA binding by HELZ, remain subject to further research. ORF1p is suggested to confer protection against IFN response, presumably by limiting the degree to which e.g. cytoplasmic pattern recognition receptors and related/connected ISGs can access the L1 RNA. By uncoupling ORF1p binding to the RNA, the authors infer certain protein interactions only co-IP in the context of ORF1p RNA-binding capacity and that IFN response is exacerbated (in support of the proposed model), presumably by the more 'naked' L1 RNA.

The connected topics of L1 expression and innate immunity / interferon response / inflammation / aging are presently of high interest and therefore subject to an intense cycle of research activity. This manuscript aligns with this high interest within the relevant research communities. However, it is my opinion that the manuscript is not structured efficiently. It is my opinion that the most interesting claim of this manuscript is the following one:

"Because the expression of each construct up regulates IFN- α expression, the data suggest that L1 RNA, but not L1 cDNA or L1 retrotransposition per se, are responsible for the modest induction of type I IFN expression."

^^ this is an unresolved aspect of L1 molecular (patho)physiology. Of course, these two possibilities are not mutually exclusive, and many investigators are not addressing the nuance that either or both can be true in a cell/tissue/state specific way. The fact that reverse transcriptase inhibitors reduce the IFN response has led others to conclude that the L1 cDNA (and/or the DNA:RNA hybrids) are the innate immune triggers - but this, in my opinion, is not a settled matter nor the only applicable mechanism. CpG content and dsRNA features may equally apply and this will become more apparent as more research is conducted. This manuscript contributes to understanding the paradigm of L1 innate immune triggers and to keeping the door open for other features than 'just' L1 cDNA as candidate targets of the related host machinery. On this basis, I am supportive of this

manuscript and would like to assist it to reach maturity and publication.

I suggest the authors consider leading this manuscript with the question as to **what** triggers the IFN response upon L1 expression (in this case ectopic over expression in model cells), and then narrate the findings in terms of the support for L1 RNA as a valid contributor - including the potential roles of their ISG hits and specifically HELZ2 and the 5'-UTR. Much of the data are in place, the manuscript just needs to be restructured.

I am agnostic to the data/claims concerning the stress granules due to the ectopic overexpression context of the manuscript. However, I believe it would be interesting to know the following: does ORF2p physical association with L1 RNA change upon loss of ORF1p L1 RNA binding (and loss of granule formation). From experiments we have done with HEK-293TLD and pLD561 - CMV::ORF2p only - (Taylor et al. 2013 and 2018) - we know that ORF2p does still bind back to its encoding RNA in the absence of ORF1p and has fractional LEAP activity. BUT, our construct lacked the WT 5'-UTR present in the authors construct (in addition to being broadly recoded) - and the authors point to this aspect of ORFeus in this manuscript. If it turns out that ORF2p assembly on L1 RNA is not significantly compromised in their system (this may even already be known from prior studies by JVMs group), it allows the following question to be asked: does interaction of HELZ and the other ISG candidates with ORF2p/L1 RNA RNP increase in the ORF1p M8 mutant? Presumably, (according to the proposed model) in the absence of ORF1p, L1 RNAs are better decorated by HELZ and connected ISGs/PRRs and this would be revealed as quantitative recovery increases in ORF2p IPs - the RT mutant ORF2p could be used to ensure that RNA is the target. I say all this because the functional significance of the granules is unclear and I do not believe it is advanced by the findings so far presented in the manuscript - this or similar kinds of experiments would allow RNP interrogation in the absence of (1) ORF1p binding and (2) granule formation; and therefore contribute to a more definitive proof that the binding occurs within the granules or not, or only within an L1 (.e.g ORF2p, L1 RNA) RNP regardless of granule status... this would mean L1 RNPs bring HELZ2 into granules as opposed to granules creating the macromolecular context for HELZ binding. I admit that, the area is complex, there is a lot to keep in mind at once - I apologize in advance if I have overlooked evidence provided by the authors that addresses my critiques, but perhaps my advice can anyway help to shape and clarify the text for a broad audience.

Aside from the above suggestions, I offer the following critiques:

LINE 77 — Can the authors clarify what they intend to be understood by the language “in close proximity to stress granules.” Although several studies point to stress granules as a place in which LINE-1 accumulates, this does not seem to be a certainty. Others have suggested they may be p-body like...Briggs, E. M. et al. RIP-seq reveals LINE-1 ORF1p association with p-body enriched mRNAs. *Mobile Dna* 12, 5 (2021).

... and our own work suggests that they may be IMP1 granule-like...

Taylor, M. S. et al. Dissection of affinity captured LINE-1 macromolecular complexes. *eLife* 7, e30094 (2018).

There are other examples (some cited in this manuscript). I am not sure who is ‘right’ - in fact all may be correct depending on the cell/tissue-type and circumstances. I think it is great that the author’s want to go down this road, but they should be specific in the use of the language (if they believe it is a stress granule, ok, then just say that), or instead be sure to remain unambiguously agnostic if they believe the nature of the granule in question is unclear. For the latter case, more details and clarity would be helpful. Depending on the level of expression and endogenous /ectopic nature could influence the results - so it is appreciated if the authors can keep this nuance in mind also for the reader.

LINE 123 — Regarding Sup Fig. 2b. Is this a representative blot? This result has been

reproduced more than once and the M9 and M10 mutants display the same behavior on repeat experiments? If so, please state for the confidence of the reader. Also, can this blot be re-probed (or a new blot run) using an anti-ORF1p antibody to rule out loss of the tag in these mutants (as well as M1). I realize that it is the view of the authors that these signals represent the steady state levels of the protein – and this is easily cross-checked - my understanding is that there's no (or very little) endogenous ORF1p expression in U2OS. So, the signals for anti-ORF1p and anti-FLAG would corroborate this and provide an additional replicate of the relative levels if this has not already been done. The Author's state cells were collected on day 5, 9, or 4 - post-transfection - can they please make the significance of this information more clear for the reader. I presume that these cells take different amounts of time under antibiotic selection to stabilize and express (?) - but for the reader, a clear mention of the specific reason for these details is appreciated - it is appreciated that the detail is given at all.

LINES 149 to 155 – Could the authors please rationalize why the activity assay is conducted in HeLa but the stress granule localization assay is carried out in U2OS cells? It's not clear the meaning of the cell line switch.

LINE 166 to 169 – WT ORF1p IPs a great number of RNAs in addition to L1 RNA: in an ectopic context, L1 RNA is a highly abundant one, but by no means the only enriched RNA. According to Fig. 1g, ORF1p does not measurably bind to GAPDH, neither in WT nor in mutant - this figure would be greatly improved if another mRNA that co-IPs with ORF1p were selected / added. The point that would be made is: ORF1p does not just lose binding activity for L1 RNA, it generally loses RNA binding activity - presumably either as a consequence of the mutations in the RRM which preclude granule formation (?). Comparison to the M7 mutant may shed further light on this matter as well - being that it is also in the RRM and has reduced but not eliminated activity. I see that the authors suggest that the loss of PABPC1 is evidence of general loss of RNA binding - and this makes sense - but I still maintain that it would be satisfying and congruent to see a non-L1 RNA as having signal in the WT and losing that signal in the mutant; GAPDH is inert in this context and in my opinion does not reveal much.

LINE 187 – section on immune-related proteins – This section leaves a lot to be desired in my opinion. It looks as though the analysis is conducted based on single replicates and therefore lack a statistical basis of confidence for the relative enrichments between the case and the control. By using GSEA, the author's side-step assessing the reliability of any particular differential protein enrichment (using standard MS-based quantitative methods) by looking at the grouping of proteins taken together. The authors compare the WT profile to the M8 profile with the reasoning that the WT provides the RNP-specific profile (which included granule localized and other localizations) and M8 provides the ORF1p-specific profile (granule excluded). The resulting NES scores are modest at best - 1.4 and 1.6 (typical label-free MS effect size-cut offs are 2 fold because label free is not usually able to reliably distinguish smaller effect sizes using 3 replicates - here we have 1 replicate). I think it would be useful for the author's to comment on these effect sizes in the text and if they should indeed be interpreted as borderline or if, for this kind of analysis, this is considered a substantial effect size.

The yield of total proteins looks higher in the WT than in the M8 (Fig. 2c) - this is expected as additional proteins associate in the context of an RNP when both ORF1p and L1 RNA are present - but the yield of ORF1p itself is either different (could be several fold) and/or is difficult to discern - this complicates the interpretation of the relative yield of ORF1p- vs L1 RNP-specific factors. I did try to check the relative intensity of L1 ORF1p in the provided spreadsheets but I did not find a protein-level intensity roll-up and the ORF1p peptides were divided across many fractions but I also did not see a column labeled intensity in either spreadsheet and this was confusing to me - I think the authors are using the number of peptides identified rather than the analyte signal-strength registered by the instrument – and if so, I would like this to be spelled out more clearly. The more common way to do this (instead of MS1 intensity) would be

spectral counting. Can the authors please also clarify if they are referring only to unique peptides (diagnostic for the proteins) or not. It would be reassuring to see quantitation on the ORF1p yield from both IPs -and generally, quant with stats from multiple replicates for the components of the IPs, which is standard practice.

Now, the authors have done some functional analysis of their selected hits (on the bases of ontological / pathway associations - caveats stated above) - so, as a journal, you may choose to prioritize this. Function is function, and it can be argued that it's less important how one got there than that effects themselves. Indeed, it is already established in the LINE-1 field that L1 induces e.g. IFN-I the responses, probably most prominently in...

Cecco, M. D. et al. L1 drives IFN in senescent cells and promotes age-associated inflammation. Nature 566, 73-+ (2019). [Ref 43 in this manuscript]

...others are published as well. And it is already established that LINE-1 is combated by comparable cellular machinery as viral-type defenses (numerous papers, including some cited in this manuscript). This prior knowledge justifies the choice of candidates on the basis of a qualitative comparisons and makes it unsurprising that these categories were represented as such.

If the journal wants to enforce best practices in quantitative mass spectrometry, a minimum of three replicates are required for a standard label-free quantitative analysis. This is easy and straightforward to execute. 3 replicates of case vs control. For this manuscripts the authors could include both M8 and no FLAG as controls in a ANOVA-style comparisons parse out the proteins that are significantly enriched in WT vs both. This would be the standard way to go about this with normal experimental and statistical rigor and should provide a nice clean ranked list. I want to make it clear that I am not saying that "I don't believe the author's findings," (because I do believe them) but I am saying that this experimental design does not conform to the rigor / norms of contemporary quantitative MS-based comparisons.

For these experiments the authors switch back to HeLa from U2OS - why? Can the rationale for cell type selection in different assays be laid out early in the manuscript? And then after this, IFN-alpha production is tested in HEK? I presume this is due to which IFN genes / pathways are active in which cell lines, but the reader needs to be informed about these choices and their rationale all along the way.

LINE 216 — "using a primer set that amplified the mneoI retrotransposition reporter 217 cassette" — since the L1 construct has it's own cleavage and polyadenylation site (SV40 I think), then I understand that this primer set amplified only read-through transcripts? (Is this common with SV40? I thought not). Presumably this choice was made to avoid amplifying other abundant endogenous L1-containing transcripts. Can the authors please make the rationale for this choice and the described the expected results compared to the obtained results - making it clear to the reader. It may be possible to use primers where one hybridizes across the sequence of the FLAG-tag which is unique to this construct and the other is common to L1 RCs to avoid reading out on the reporter.

LINE 240 — I think number of number of peptides (are we talking unique peptides?) is being used confusingly as a proxy for abundance. The authors should use the peptide-spectrum matches or spectral counts or something else here... a large protein that is low abundance could still provide more detectable peptides than a small protein that is higher abundance - I suggest the authors please clarify how they are using the peptides in greater details - for this a table in the text for the proteins of interest, extracting a few pieces of key info from the spreadsheets that are currently in the supplement would suffice. And again, I suggest some replicates and proper quant to make any kinds of claims about "fold-change."

LINE 331 — helicase domain 1 or domain 2 is more important? I am reading that WA1 and WA2 are more severe and WB1 and WB2 are less severe in terms of inhibition of transpositions. Doesn't this make the more severe WA domain more important (yet helicase 2 is emphasized in text, either I misunderstood this section or it is a typo, I think)? Shouldn't there be 4 open triangles on Fig. 5a, indicating the positions of 4 defined amino acid substitutions? I see two triangles but 4 positions? Have I misunderstood?

Looking at the data - mutants in WA1/2 abrogate the HELZ2 effects in HeLa, but function fine in HEK, a not dissimilar result is seen in WB1/2... the results seem rather inconclusive to me. I am not certain that Fig. 5b-3 contribute to the manuscript and could be moved to the supplement.

LINE 335 — the WA2 and WA1/2 appear hugely over expressed compared to WT and WA1, are these possible aggregating or otherwise not participating in the same molecular biology as the WT or WA1? I am surprised that WA1/2 does not show (at least) the same effect as WA1 - are we instead seeing an unexpected sign epistasis? considering the expression levels of WA2 and WA1/2 this seems difficult to interpret and I think goes a ways towards explaining the results in 5b-e and further suggests these experiments might be better in the supplements and claims in this section refined and pared down?

LINE 410 — (Fig. 6f) I believe this needs a control that is "no L1" (pCEP) + HELZ2 — does over-expression of HELZ drive IFN down regardless of L1 status?

Discussion

Is the HELZ2 exonuclease activity unambiguously shown? Did I miss the reference? If this has been shown - great, reference it (and my apologies if I missed that). If it has not been shown, I am not certain it is wise to presume on the basis of sequence / structure homology. I get that the word putative was used - and that is fine when discussing homologs, broadly, but in the discussion exonuclease activity is put forward as a mechanism to degrade L1. This raises the bar. Furthermore, HELZ2 is shown to rely on binding the 5' UTR, but its putative activity is 3'→5' so, this does not quite add up. Bottom line, 3'-5' exonucleases need access to free 3'-ends, and typically PABPC1/4 have to be cleared - I guess in a mechanism using CCR4-NOT and/or PAN2/3 - then by the RNA exosome... anyway, my point is that the authors may be making a few too many assumptions here. Since they have the Myc-tagged HELZ2 (and mutants thereof) they could easily test in vitro RNase activity (including against in vitro transcribed L1 RNAs +/- 5' UTR) - these don't have to be sophisticated assays, but POC that shows there is the expected activity - this will go a ways towards bridging some of the assumptions of the model.

Other comments

* Why ignore the significant GO term related to NMD that also would affect the level of L1 RNA? I guess this was simply to focus the paper on the viral defenses topic? Can the NMD nodes be highlighted in Fig. 3C so that we can see how the compositions of these enriched terms intersect / interact?

* Fig 5g - please validate siRNA KDs by western blotting not RT-qPCR (it is not that I don't believe the result, it is that the readout of a KD is the protein level, not the RNA level).

* If ORF2p is expression w/o the 5'-UTR then HELZ overexpression has no effect on Alu?

* how much acetone was used to precipitate proteins (not defined in the method)

* can you please us units for your % - BSA is no doubt at 0.5% w/v and Triton X-100 is at 1% v/v - but this should be explicitly defined as a matter of routine any time % by weight or volume is used.

*** not the author's fault but neither www.interferome.org nor the URL given in the reference (which is different) seem to load.**

**Signed,
John LaCava**

Reviewer #3 (Remarks to the Author):

The article "The interferon stimulated gene-encoded protein HELZ2 inhibits human LINE-1 retrotransposition and LINE-1 RNA-Mediated type I interferon induction" by Miyoshi and colleagues is an interesting study of proteins induced by the interferon response to LINE-1 RNA that can inhibit retrotransposition of LINE-1. This is an exciting and thorough study that gives insight into control of LINE-1 expression in the cell and links it with type I interferon signaling. In general the experiments are well controlled and the conclusions are well supported by the data presented. I have the following questions:

Major points

- 1. In Figure 2, the authors assess "IFN-a" RNA by qRT-PCR. They should be specific about which IFN-a as there are many and different ones are expressed by different cell types. They should also assess levels of interferon beta and interferon lambda (type III IFN) as both have previously been shown to be activated by transposable element RNA.**
- 2. The decrease in Alu retrotransposition (in Figure 6) when HELZ2 or MOV10 is overexpressed is interesting but the authors should put it in context. Are these proteins associating with Alu RNA as well?**
- 3. Besides IFN alpha RNA (qRT-PCR), the authors should assess secreted interferon alpha/beta and a panel of downstream interferon stimulated genes to ensure their L1 expression causes a bona fide interferon response. To link the type I IFNs with the ISGs (their model in Figure 7) they could also perform antibody blocking experiments of the IFNAR1 receptor to prove that type I interferon activation of ISG proteins is what limits L1 retrotransposition.**

Minor points

- 1. The authors should explain why they used arsenite (line 161) when studying stress granules**
- 2. Please show (in the supplement) the original flow plots for the quantified EGFP+ cell data (i.e. Figure 5e) for reference**

Reponses to the referees' comments of Nature Communications manuscript NCOMMS-22-10777 entitled, "The interferon stimulated gene-encoded protein HELZ2 inhibits human LINE-1 retrotransposition and LINE-1 RNA-mediated type I interferon induction," by Luqman-Fatah et al.

We thank the editors and reviewers for their time and scholarly comments. Please find our point-by-point responses to the reviewers' comments below. The original comments are noted in black text. Our responses are noted in blue text.

Reviewer #1 (Remarks to the Author):

The interferon stimulated gene-encoded protein HELZ2 inhibits human LINE-1 retrotransposition and LINE-1 RNA-mediated type I interferon induction

Luqman-Fatah et al.

In this elegant work, the authors identify novel L1 regulators by employing an RNA binding mutant (M8 RBM) of L1 ORF1 protein, in parallel to the wildtype L1 ORF1 protein to map the interactome of L1 ribonucleoprotein particles in an RNA-dependent manner. Both versions of the L1 ORF1 protein are equally expressed in cell lines but the M8 RBM fails to bind RNA, is not able to form RNP foci in cytoplasm and leads to greatly reduced engineered L1 retrotransposition efficiency. The authors perform IP-Mass spec with both L1 ORF1p constructs and find an enrichment of proteins indicative of interferon signaling specifically with the wild type protein, implying that their interaction with ORF1p is dependent on RNP foci formation/RNA binding. The interactome includes 5 ISGs not previously known to interact with L1 RNPs and the authors go on to confirm all these ORF1 binding partners by co-IP. The mechanism of how these ISGs regulate L1s is confirmed by assessing engineered L1s in their retrotransposition efficiency, ORF1p levels, L1 RNA and RNP foci formation. Data suggests that HELZ2 blocks L1 activity by acting on L1 RNA through its decreased stability or digestion. Further analyses identifies the helicase domains (particularly helicase 2) to be crucial to the anti-LINE-1 function of HELZ2 and the L1 5'UTR to be the target sequence of HELZ2 recognition. This is a remarkable, thorough, and important work, relevant to scientists in diverse fields of research ranging from genome instability to immunology and I have almost no comments.

We thank the reviewer for their kind words and scholarly comments on our study.

Minor comments

1.Line 89: It should be clarified at this point in the text that the WT ORF1 protein only and not the M8 RBM ORF1p associated with the cohort of ISGs identified.

We thank the reviewer for the suggestion. We have modified the text.

On page 5, we now state:

"Immunoprecipitation (IP) coupled with liquid chromatography-tandem mass spectrometry (LC-MS/MS) analyses followed by Gene Ontology (GO)^{55,56} and Gene Set Enrichment Analysis (GSEA)⁵⁷ that compared the proteins associated with **WT ORF1p vs. an ORF1p triple mutant that impairs RNA binding (M8/RBM)** revealed that a full-length RC-L1 containing a carboxyl-terminal epitope-tagged version of ORF1p (WT ORF1p-FLAG) preferentially associates with proteins encoded by several interferon stimulated genes (ISGs), including HERC5, HELZ2, OASL, DDX60L, and IFIT1."

2.Line 178: This sentence would be clearer if the word 'L1' was added after 'WT ORF1p-FLAG' : 'conducted with WT ORF1p-FLAG cell extracts, but was severely reduced in IP reactions conducted with M8/RBM ORF1p-FLAG L1 cell extracts'.

We thank the reviewer for this suggestion and added 'L1' in the sentence.

On page 9, we now state:

"Moreover, the Poly(A) Binding Protein Cytoplasmic 1 (PABPC1) was robustly detected in IP reactions conducted with cell extracts derived from WT ORF1p-FLAG **L1** transfected cells,...."

3.Line 322: This sentence is a bit unclear: 'The WA1 mutant demonstrated a low, but not statistically significant decrease in L1 retrotransposition efficiency'. Could be rephrased i.e. 'The WA1 mutant was still able to block retrotransposition almost as effectively as WT HELZ2'.

Line 331 is also unclear: 'In general, the WA2 and WB2 mutants consistently exhibited a less severe inhibition of L1 retrotransposition'. Could change to 'whereas the WA2 and WB2 mutants were impaired in their ability to block retrotransposition' or similar.

We apologize for the unclear phrasing; we have modified the paragraph as suggested to avoid confusion.

The paragraph on page 15 now read as follows:

"The WA1 mutant was able to inhibit L1 retrotransposition almost as effectively as WT HELZ2 in HEK293T (Fig. 5b), but not HeLa-JVM (Fig. 5c) cells. The WA2 and WA1&2 double mutants were significantly impaired in their ability to inhibit L1 retrotransposition in both HEK293T (Fig. 5b) and HeLa-JVM cells (Fig. 5c). A similar data trend was observed for the Walker B box mutations in HEK293T and HeLa-JVM cells (Supplementary Figs. 5g and 5h, respectively). In sum, mutations in the helicase 2 (WA2 and WB2) domains generally alleviated the HELZ2-mediated repression of L1 retrotransposition to a greater extent than mutations in the helicase 1 (WA1 and WB1) mutants, indicating the importance of the helicase 2 domain in the inhibition of L1 retrotransposition."

We also have moved **Figs. 5d and 5e** (WB1/2) to the Supplement (now **Supplementary Figs. 5g and 5h**, respectively).

4.Figure 1: The use of the mcherry-G3BP1 construct in this experiment was not well explained.

We apologize for the oversight and have clarified the text.

On page 8, we now state:

"..., G3BP1, which is a widely used stress granule marker, that is tagged at its amino terminus with a mCherry fluorescent protein (mCherry-G3BP1)...."

5.Figure 2: It would be interesting to know here what the genes are corresponding to these gene ontology terms (viral transcription and interferon-alpha/gamma response) in supplementary?

We thank the reviewer for this comment. Our label-free quantitative mass spectrometry analyses, which also were requested by Reviewer 2 (see below), revealed the following viral-related GO terms: host-virus interaction, innate immunity, and antiviral defense (see updated [Fig. 2d]). The list of proteins and respective GO terms can now be found in **Supplementary Table 1**. The list of proteins for GSEA interferon-alpha/gamma responses can be found in **Supplementary Table 2**.

6.Figure 3a: From the legend it appears that L1 regulator candidates that were ISGs were focused on amongst the top 300 mass spec. candidates. Can a ranking be given to these candidates in the table to show how high up they appeared in the screen depending on the no. of peptides scored for each of these proteins? All the 300 protein hits should be listed in a supp table, included with the manuscript.

Thank you for the suggestion. We have modified the text based upon our label-free quantitative mass spectrometry analyses. We have included all the protein hits with the respective log₂ abundance ratios (WT vs M8[RBM]) and rankings in the **Source data file, Comparisons tab**. The studied proteins (i.e., HELZ2, HERC5, OASL, IFIT1, and DDX60L) are annotated on the volcano plot and their respective p-values and log₂ abundance ratios (WT vs. M8/RBM) are noted in **Fig. 3b**.

7.Figure 6E: It would be interesting to know if HELZ2 recognizes the L1 5'UTR through its RNA binding/helicase or other domain? However, this is such a complete study, it could be a discussion point.

We appreciate the reviewer's suggestion and have added the following comment in the discussion.

On pages 21 and 22, we now state:

"RNB domains typically are flanked by cold shock and S1 domains that form an RNA-binding channel. However, HELZ2 appears to lack these domains, as well as conserved amino acids

associated with these domains⁹⁸; thus, it remains unclear which domain of HELZ2 recognizes the L1 5'UTR."

8. Figure 7: In the model, I think the exclamation marks can be removed, especially as they detract from the main message of the paper (which is about novel ISGs not RNA sensors).

We thank the reviewer for the suggestion and have removed the exclamation marks in the model.

9. Supp Figure 5: It is interesting that some residues of HELZ2 are conserved in yeast. Since yeast do not have LINE-1 elements, the authors should comment on the possible role of this host protein in yeast in the discussion.

We thank the reviewer for this suggestion. The best characterized RNB-containing protein in yeast (Rrp44/Dis3) is a component of the RNA exosome and functions in RNA quality control.

On page 20, we now state:

"Other RNB-containing proteins are known to function in RNA quality control (e.g., the yeast and human RNA exosome component Dis3, and prokaryotic cold shock inducible protein RNase R⁹⁹). A more in-depth analysis of HELZ2 revealed mechanistic similarities to other RNB-containing proteins, which can degrade highly structured RNAs through its concerted helicase and 3' to 5' exoribonuclease activities^{100,101}."

10. It would be interesting to note in the discussion that the main function of LINE-1 ORF1p may be to bind and shield L1 RNA from host ISG proteins, including RNA sensors, thereby explaining its requirement for retrotransposition. This is implied in the abstract but not really mentioned further. Also, IFN-alpha is mentioned several times but not IFN-beta, which is a fundamental type I IFN so it would be more appropriate to refer to 'type I IFNs' in general in the discussion.

We thank the reviewer for the suggestions. As suggested, we have changed IFN-alpha to type I IFN in the discussion.

On page 22, we have added the following sentence to explain the contribution of ORF1p to L1 retrotransposition:

"Our working model further suggests that ORF1p binding to L1 RNA may attenuate the type I interferon response, which, in turn, might reduce the expression of inhibitory ISG proteins."

Reviewer #2 (Remarks to the Author):

This manuscript explores the relationship between L1 RNPs and some interferon stimulated genes that are likely to be host-defenses against L1 expression. In the manuscript, the candidate HELZ2 is examined most closely and characterized as a candidate L1 RNA binder in the 5'UTR. A relationship between L1 expression /HELZ2 expression / and interferon response is demonstrated. Precise details of the chain of events, post-L1 RNA binding by HELZ, remain subject to further research. ORF1p is suggested to confer protection against IFN response, presumably by limiting the degree to which e.g. cytoplasmic pattern recognition receptors and related/connected ISGs can access the L1 RNA. By uncoupling ORF1p binding to the RNA, the authors infer certain protein interactions only co-IP in the context of ORF1p RNA-binding capacity and that IFN response is exacerbated (in support of the proposed model), presumably by the more 'naked' L1 RNA.

The connected topics of L1 expression and innate immunity / interferon response /inflammation / aging are presently of high interest and therefore subject to an intense cycle of research activity. This manuscript aligns with this high interest within the relevant research communities. However, it is my opinion that the manuscript is not structured efficiently. It is my opinion that the most interesting claim of this manuscript is the following one:

"Because the expression of each construct up regulates IFN- α expression, the data suggest that L1 RNA, but not L1 cDNA or L1 retrotransposition per se, are responsible for the modest induction of type I IFN expression."

^ this is an unresolved aspect of L1 molecular (patho)physiology. Of course, these two possibilities are not mutually exclusive, and many investigators are not addressing the nuance that either or both

can be true in a cell/tissue/state specific way. The fact that reverse transcriptase inhibitors reduce the IFN response has led others to conclude that the L1 cDNA (and/or the DNA:RNA hybrids) are the innate immune triggers - but this, in my opinion, is not a settled matter nor the only applicable mechanism. CpG content and dsRNA features may equally apply and this will become more apparent as more research is conducted. This manuscript contributes to understanding the paradigm of L1 innate immune triggers and to keeping the door open for other features than 'just' L1 cDNA as candidate targets of the related host machinery. On this basis, I am supportive of this manuscript and would like to assist it to reach maturity and publication.

We thank the reviewer for their scholarly review and enthusiastic comments. We agree with the reviewer that many aspects of L1 molecular (patho)physiology remain unresolved with respect to the extent to which presumptive L1 intermediates (e.g., L1 RNA, L1 cDNA, or L1 DNA:RNA hybrids) can trigger innate immune responses and how host factors might respond to these triggers. However, we hope the reviewer agrees that our manuscript provides additional evidence for how host factors involved in the innate immune response interact with L1 RNA and/or RNPs.

I suggest the authors consider leading this manuscript with the question as to *what* triggers the IFN response upon L1 expression (in this case ectopic over expression in model cells), and then narrate the findings in terms of the support for L1 RNA as a valid contributor - including the potential roles of their ISG hits and specifically HELZ2 and the 5'-UTR. Much of the data are in place, the manuscript just needs to be restructured.

We thank the reviewer for their comment and helpful suggestions. We understand that the relationship among retrotransposons and the innate immune response is a topic of high interest to the research community; however, the main focus of our study was to identify cellular proteins that preferentially interact with wild type vs. M8 (RBM) mutant ORF1p proteins and to elucidate how the identified proteins affect L1 retrotransposition. Our insights into the IFN response resulted from these efforts. Given the positive comments by other reviewers, we respectfully prefer to keep the current organizational structure of the manuscript, as it accurately describes the path and scientific logic we used to develop the story. As noted below, we have included additional experiments and edited the manuscript to address each of the reviewer's comments.

I am agnostic to the data/claims concerning the stress granules due to the ectopic overexpression context of the manuscript. However, I believe it would be interesting to know the following: does ORF2p physical association with L1 RNA change upon loss of ORF1p L1 RNA binding (and loss of granule formation). From experiments we have done with HEK-293TLD and pLD561 - CMV::ORF2p only - (Taylor et al. 2013 and 2018) - we know that ORF2p does still bind back to its encoding RNA in the absence of ORF1p and has fractional LEAP activity. BUT, our construct lacked the WT 5'-UTR present in the authors construct (in addition to being broadly recoded) - and the authors point to this aspect of ORF2p in this manuscript. If it turns out that ORF2p assembly on L1 RNA is not significantly compromised in their system (this may even already be known from prior studies by JVMs group), it allows the following question to be asked: does interaction of HELZ and the other ISG candidates with ORF2p/L1 RNA RNP increase in the ORF1p M8 mutant? Presumably, (according to the proposed model) in the absence of ORF1p, L1 RNAs are better decorated by HELZ and connected ISGs/PRRs and this would be revealed as quantitative recovery increases in ORF2p IPs - the RT mutant ORF2p could be used to ensure that RNA is the target. I say all this because the functional significance of the granules is unclear and I do not believe it is advanced by the findings so far presented in the manuscript - this or similar kinds of experiments would allow RNP interrogation in the absence of (1) ORF1p binding and (2) granule formation; and therefore contribute to a more definitive proof that the binding occurs within the granules or not, or only within an L1 (e.g. ORF2p, L1 RNA) RNP regardless of granule status... this would mean L1 RNPs bring HELZ2 into granules as opposed to granules creating the macromolecular context for HELZ binding. I admit that the area is complex, there is a lot to keep in mind at once - I apologize in advance if I have overlooked evidence provided by the authors that addresses my critiques, but perhaps my advice can anyway help to shape and clarify the text for a broad audience.

We thank the reviewer for the insightful comments.

To address the reviewer's question regarding ORF2p RNA binding, we performed additional RNA-immunoprecipitation (IP) experiment using the WT and M8/RBM L1 constructs that express L1 ORF2p-3xFLAG using parameters similar as those reported in **Fig. 1g** (RNA-IP with ORF1p). The results of these analyses (**shown below**) revealed that L1 RNA was enriched in the M8/RBM ORF2p-3xFLAG precipitate when compared to the WT ORF2p-3xFLAG control (**left panel**). Unexpectedly, ORF2p-3xFLAG also was more abundant in the M8/RBM L1 compared to the WT ORF2p-3xFLAG control (**right panel**), even though both constructs produced similar levels of ORF1p-T7 (please also see the L1 RNA levels in **Fig. 6b**), which could, in principle, explain the enrichment of L1 RNA in the M8/RBM RNA-IP experiment (**left panel**). We can envision at least two possibilities to account for these results: (1) ORF2p might be more stable when produced from the M8/RBM L1 construct; or (2) ORF2p translation might occur more efficiently, or is not subject to negative feedback inhibition, when ORF1p does not bind to L1 RNA. Given the available data, it currently is difficult to quantify and compare the binding of ISGs to L1 RNA in the WT vs. M8/RBM ORF2p-3xFLAG immunoprecipitation experiments. As we hope the reviewer would agree, future studies, which are beyond the scope of this manuscript, are required to distinguish between the models described above.

Review only Figure legend:

(Left): L1 RNA immunoprecipitation using ORF2p-3xFLAG. HeLa-JVM cells were co-transfected with either pJM101 (control, no tag), pTMF3 (WT, ORF2p-3xFLAG), or pTMF3_M8 (M8 [RBM], ORF2p-3xFLAG). The ORF2p-3xFLAG complexes were immunoprecipitated using an anti-FLAG antibody, the bound RNA was purified, and L1 RNA levels were assessed by RT-qPCR using either the SV40 primer set (L1) or the GAPDH primer set (GAPDH), which served as a control.

(Right): Western blot analysis of ORF1p-T7 and ORF2p-3xFLAG in the pJM101, pTMF3, or pTMF3_M8 transfected HeLa-JVM cells. Cell extracts were prepared as in the left panel. Anti-FLAG and anti-T7 antibodies were used to detect ORF2p-3xFLAG and ORF1p-T7, respectively. GAPDH served as a loading control.

To detect possible interactions between ISGs and L1 RNA, we used the same RNA-IP strategy described above, using an epitope-tagged version of HELZ2 with 3xFLAG at the carboxyl-terminus. However, we did not observe appreciable amounts of L1 RNA (data not shown), which likely is due to our finding that HELZ2 overexpression significantly reduces L1 RNA levels (see **Fig. 4c**).

Although it is difficult to investigate whether more HELZ2 binds to the L1 RNA in the absence of ORF1p binding, our results (**Fig. 6b**) revealed that HELZ2 overexpression reduced the steady levels of both WT L1 and M8/RBM L1 RNA relative to a pCEP4 mock control, suggesting that HELZ2 can recognize, bind, and perhaps degrade L1 RNA in the absence of ORF1p binding. We further demonstrate that HERC5 overexpression can reduce both WT ORF1p and M8/RBM ORF1p steady state levels to similar extents (**Supplementary Fig. 6a**). Thus, these data suggest that HELZ2 and HERC5 target L1 RNA instead of the L1 RNP, and that ORF1p cytoplasmic foci formation may not be necessary for the interactions between ISGs and L1 RNA, as the M8/RBM exhibited a severe impairment in ORF1p cytoplasmic foci formation (**Supplementary Fig. 3d**).

To account for the above results, we modified the manuscript. On page 17 in the Result section, we now state:

*“We observed a similar reduction in L1 WT ORF1p-FLAG and M8/RBM ORF1p-FLAG protein levels upon the co-expression of HERC5 in HeLa-JVM cells (**Supplementary Fig. 6a**). Thus, both HELZ2 and HERC5 overexpression appear to destabilize L1 RNA and ORF1p, respectively, independent of WT L1 RNP formation.”*

Our data further demonstrated that the type I IFN response induced by the expression of the M8/RBM L1 was slightly higher than that induced by the expression of the WT L1 (**Fig. 2f**), suggesting that ORF1p binding to L1 RNA may protect L1 RNPs from being recognized by innate immune response sensor(s).

To clarify the presentation of our results, we have added the following sentence that explains the ORF1p contribution to retrotransposition on page 22:

“Our working model further suggests that ORF1p binding to L1 RNA may attenuate the type I interferon response, which, in turn, might reduce the expression of inhibitory ISG proteins.”

We hope these additional experiments and explanations satisfy the reviewer’s concerns and thank them again for raising these important questions.

Aside from the above suggestions, I offer the following critiques:

LINE 77 — Can the authors clarify what they intend to be understood by the language “in close proximity to stress granules.” Although several studies point to stress granules as a place in which LINE-1 accumulates, this does not seem to be a certainty. Others have suggested they may be p-body like...Briggs, E. M. et al. RIP-seq reveals LINE-1 ORF1p association with p-body enriched mRNAs. *Mobile DNA* 12, 5 (2021). ... and our own work suggests that they may be IMP1 granule-like...Taylor, M. S. et al. Dissection of affinity captured LINE-1 macromolecular complexes. *eLife* 7, e30094 (2018).

There are other examples (some cited in this manuscript). I am not sure who is ‘right’ - in fact all may be correct depending on the cell/tissue-type and circumstances. I think it is great that the author’s want to go down this road, but they should be specific in the use of the language (if they believe it is a stress granule, ok, then just say that), or instead be sure to remain unambiguously agnostic if they believe the nature of the granule in question is unclear. For the latter case, more details and clarity would be helpful. Depending on the level of expression and endogenous /ectopic nature could influence the results - so it is appreciated if the authors can keep this nuance in mind also for the reader.

We thank the reviewer for the insightful comments,

Our data provide evidence that L1 cytoplasmic foci may be distinct from stress granules, as we only observed stress granules upon sodium arsenite treatment (**Supplementary Fig. 3b and 3c**). Moreover, data in Doucet *et al.*, 2010 indicate that ORF1p cytoplasmic foci can closely associate with stress granules. As requested by the reviewer, we have removed “in close proximity to stress granules” and on page 4 and now state:

“Previous studies revealed that ORF1p, ORF2p, and L1 RNA can localize within cytoplasmic foci that closely associate with stress granule (SG) proteins...”

LINE 123 — Regarding Sup Fig. 2b. Is this a representative blot? This result has been reproduced more than once and the M9 and M10 mutants display the same behavior on repeat experiments? If so, please state for the confidence of the reader. Also, can this blot be re-probed (or a new blot run) using an anti-ORF1p antibody to rule out loss of the tag in these mutants (as well as M1). I realize that it is the view of the authors that these signals represent the steady state levels of the protein – and this is easily cross-checked - my understanding is that there’s no (or very little) endogenous ORF1p expression in U2OS. So, the signals for anti-ORF1p and anti-FLAG would corroborate this and provide an additional replicate of the relative levels if this has not already been done. The Author’s state cells were collected on day 5, 9, or 4 - post-transfection - can they please make the significance of this information more clear for the reader. I presume that these cells take different amounts of time under antibiotic selection to stabilize and express (?) - but for the reader, a clear

mention of the specific reason for these details is appreciated - it is appreciated that the detail is given at all.

Thank you for the comments and suggestions.

The ORF1p steady state result is reproducible and we conducted four independent replicates of this experiment; the blot shown in **Supplementary Fig. 2b** is a representative blot. As pointed out by the reviewer, the M9 and M10 mutants exhibited some variability across the cell lines. Based on the reviewer's suggestion, we examined the ectopically expressed ORF1p-FLAG steady state level using both anti-FLAG and anti-ORF1p antibodies, where an ectopically expressed non-tagged ORF1p was included as a negative control for anti-FLAG antibody and a positive control for anti-ORF1p antibody. Since both antibodies showed similar steady state levels of ORF1p and only a single band of ORF1p was observed in the anti-ORF1p blot, it is unlikely that the FLAG epitope tag was removed or degraded from ORF1p. In addition, an anti-ORF1p antibody showed a relatively low amount of endogenous ORF1p in all three U-2 OS, HeLa-JVM, and HEK293T cell lines (**Supplementary Fig. 2b**)

On pages 6 and 7, we now state:

*"The steady state levels of ORF1p in the M9 mutant were reduced in each cell line and the steady state level of ORF1p in the M10 mutant was more mildly reduced in U-2 OS, but not HEK293T and HeLa-JVM cells, when compared to the WT ORF1p-FLAG control... Similar results were obtained in Western blots using an anti-ORF1p antibody, which can also detect endogenous ORF1p (**Supplementary Fig. 2b**)."*

As suggested, in the **Supplementary Fig. 2b legend**, we have added additional detail to clarify our methodology. The text now states:

*"U-2 OS, HeLa-JVM, or HEK293T cells were collected on day 5, day 9, or day 4 post-transfection, respectively, **which were determined to be the optimal days to observe ORF1p steady state levels in the respective cell lines.**"*

Notably, the difference in ORF1p kinetics was previously reported in HeLa-JVM cells in Kulpa *et al.*, 2005.

As requested, we also have compared our results to the ORF1p tri-alanine mutagenesis paper (Adney EM. *Et al.*, 2019). The severe reduction of ORF1p steady state level in the M1 mutant generally agrees with the comprehensive tri-alanine scanning mutation of ORF1p reported in Adney *et al.* study (11.4% of the WT, corresponding amino acids 158-160_LRL-AAA [in comparison to M1, N157A&R159A]). The mutations corresponding to M9 and M10 in the Adney *et al.* study also showed some reduction in ORF1p levels (81.2% of the WT, corresponding amino acids, 260-262_ARR-AAA [in comparison to M9, R261A]; and 68.4% of the WT; corresponding amino acids, 281-283_SYP-AAA [in comparison to M10, Y282A]). Although our results generally agree with the Adney *et al.*, study, it should be noted that the mutations we created are different from those in the tri-alanine scanning mutation study.

To highlight these comparisons, we now cited Adney *et al.*, 2019 and included the following text on page 7:

"These results are in general agreement with a previous ORF1p alanine scanning mutational analyses⁵⁹."

LINES 149 to 155 — Could the authors please rationalize why the activity assay is conducted in HeLa but the stress granule localization assay is carried out in U2OS cells? It's not clear the meaning of the cell line switch.

We thank the reviewer for the suggestion. We used U-2 OS cells for the ORF1p cytoplasmic foci and stress granule localization assays because our earlier experiments revealed that L1 cytoplasmic foci and stress granules are more easily detected in U-2 OS cells due to the larger observable surface area in these cells (see Doucet *et al.*, 2010). We subsequently performed most of our experiments in HeLa-JVM cells, as it is a commonly used cell line for L1 retrotransposition assays when compared to U-2 OS cells.

On page 8, we now state:

“A previous study revealed that U-2 OS cells allow the ready detection of L1 ORF1p cytoplasmic foci⁴⁹ and G3BP1-containing stress granules (Supplementary Fig. 3a).”

LINE 166 to 169 — WT ORF1p IPs a great number of RNAs in addition to L1 RNA: in an ectopic context, L1 RNA is a highly abundant one, but by no means the only enriched RNA. According to Fig. 1g, ORF1p does not measurably bind to GAPDH, neither in WT nor in mutant - this figure would be greatly improved if another mRNA that co-IPs with ORF1p were selected / added. The point that would be made is: ORF1p does not just lose binding activity for L1 RNA, it generally loses RNA binding activity - presumably either as a consequence of the mutations in the RRM which preclude granule formation (?). Comparison to the M7 mutant may shed further light on this matter as well - being that it is also in the RRM and has reduced but not eliminated activity. I see that the authors suggest that the loss of PABPC1 is evidence of general loss of RNA binding - and this makes sense - but I still maintain that it would be satisfying and congruent to see a non-L1 RNA as having signal in the WT and losing that signal in the mutant; GAPDH is inert in this context and in my opinion does not reveal much.

We agree with the reviewer and addressed this question by checking the enrichment of non-L1 RNAs in our WT and M8/RBM ORF1p RNA-IP experiments. Using the same WT and M8/RBM ORF1p RNA-IP sample (**Fig. 1g**), we have examined the enrichment of two different mRNAs, HLTF and SMC2, which were reported to be enriched in an ORF1p RIP-seq experiment (see Briggs EM. *et al.*, Mobile DNA, 2021). The data suggest that the M8/RBM mutant generally is compromised for RNA binding (**Supplementary Fig. 3e**).

On page 8, we cited Briggs et al., 2021, and now state:

*“RNA-immunoprecipitation (RNA-IP) experiments confirmed that the M8 mutant was impaired for its ability to bind L1 and **other cellular RNAs** when compared to WT ORF1p, **suggesting the M8 mutant exhibited a general loss in its ability to bind RNA** (Fig. 1g, **Supplementary Fig. 3e**, and see below), which is consistent with the previous study¹².”*

LINE 187 — section on immune-related proteins — This section leaves a lot to be desired in my opinion. It looks as though the analysis is conducted based on single replicates and therefore lack a statistical basis of confidence for the relative enrichments between the case and the control. By using GSEA, the author's side-step assessing the reliability of any particular differential protein enrichment (using standard MS-based quantitative methods) by looking at the grouping of proteins taken together. The authors compare the WT profile to the M8 profile with the reasoning that the WT provides the RNP-specific profile (which included granule localized and other localizations) and M8 provides the ORF1p-specific profile (granule excluded). The resulting NES scores are modest at best - 1.4 and 1.6 (typical label-free MS effect size-cut offs are 2 fold because label free is not usually able to reliably distinguish smaller effect sizes using 3 replicates - here we have 1 replicate). I think it would be useful for the author's to comment on these effect sizes in the text and if they should indeed be interpreted as borderline or if, for this kind of analysis, this is considered a substantial effect size.

We thank the reviewer for the insightful comments.

As requested, we have repeated the IP-MS experiment two more times to generate three replicate datasets to address the issue regarding the effect size (see below). We used preranked GSEA for the analysis using the calculated log₂ abundance ratios of proteins associated with WT vs. M8/RBM. Interferon alpha and interferon gamma responses remain in the top six list of most enriched gene sets (**Fig. 2e and Supplementary Table 2**). Importantly, we used GSEA to complement (not replace) the standard MS-based analysis (**see Source data**). We found that the NES scores are not comparable to MS effect size cut-offs, as the GSEA NES scores are calculated differently from the log₂ abundance ratios of WT ORF1p-FLAG vs. M8/RBM-FLAG in the mass spectrometry.

To clarify the text, on page 10, we now state:

*“We next performed a preranked Gene Set Enrichment Analysis (GSEA) using the log₂ abundance ratio of WT ORF1p-FLAG vs. M8/RBM-FLAG IP/LC-MS/MS protein hits to determine if there was an enrichment of hallmark gene set signatures in the Molecular Signatures Database (MsigDB) (**see***

Methods). *These analyses identified two interferon-related gene sets—the interferon alpha and interferon gamma responses—among the top six most significantly enriched gene sets (Fig. 2e and Supplementary Table 2, see Methods)."*

The yield of total proteins looks higher in the WT than in the M8 (Fig. 2c) - this is expected as additional proteins associate in the context of an RNP when both ORF1p and L1 RNA are present - but the yield of ORF1p itself is either different (could be several fold) and/or is difficult to discern - this complicates the interpretation of the relative yield of ORF1p- vs L1 RNP-specific factors. I did try to check the relative intensity of L1 ORF1p in the provided spreadsheets but I did not find a protein-level intensity roll-up and the ORF1p peptides were divided across many fractions but I also did not see a column labeled intensity in either spreadsheet and this was confusing to me - I think the authors are using the number of peptides identified rather than the analyte signal-strength registered by the instrument — and if so, I would like this to be spelled out more clearly. The more common way to do this (instead of MS1 intensity) would be spectral counting. Can the authors please also clarify if they are referring only to unique peptides (diagnostic for the proteins) or not. It would be reassuring to see quantitation on the ORF1p yield from both IPs -and generally, quant with stats from multiple replicates for the components of the IPs, which is standard practice.

Now, the authors have done some functional analysis of their selected hits (on the bases of ontological / pathway associations - caveats stated above) - so, as a journal, you may choose to prioritize this. Function is function, and it can be argued that it's less important how one got there than that effects themselves. Indeed, it is already established in the LINE-1 field that L1 induces e.g. IFN-I the responses, probably most prominently in...Cecco, M. D. et al. L1 drives IFN in senescent cells and promotes age-associated inflammation. *Nature* 566, 73+ (2019). [Ref 43 in this manuscript]...others are published as well. And it is already established that LINE-1 is combated by comparable cellular machinery as viral-type defenses (numerous papers, including some cited in this manuscript). This prior knowledge justifies the choice of candidates on the basis of a qualitative comparisons and makes it unsurprising that these categories were represented as such.

If the journal wants to enforce best practices in quantitative mass spectrometry, a minimum of three replicates are required for a standard label-free quantitative analysis. This is easy and straightforward to execute. 3 replicates of case vs control. For this manuscripts the authors could include both M8 and no FLAG as controls in an ANOVA-style comparisons parse out the proteins that are significantly enriched in WT vs both. This would be the standard way to go about this with normal experimental and statistical rigor and should provide a nice clean ranked list. I want to make it clear that I am not saying that "I don't believe the author's findings," (because I do believe them) but I am saying that this experimental design does not conform to the rigor / norms of contemporary quantitative MS-based comparisons.

Thank you for stating your concerns and suggestions to improve our experimental design. We agree with the reviewer's concerns regarding the experimental rigor and standard of practice in quantitative mass spectrometry. Thus, as noted above, we repeated the IP-MS experiments to obtain triplicate datasets. We also have run the results through the ANOVA label-free quantitative (LFQ) analysis using Proteome Discoverer software (Thermo Fisher Scientific) that was developed together with another well-established quantitative analysis suite, MaxQuant, which can be used to compare the \log_2 abundance ratios between WT and M8/RBM protein hits (Palomba *et al.*, *J. Proteome Res.* 2021). These results now are shown in a volcano plot in **Fig. 3b** (also see **Source Data**). The \log_2 protein abundance ratio then was normalized to the obtained ORF1p amount (shown in the middle of the volcano plot); we used a cutoff of 0.5 \log_2 (abundance ratio of WT ORF1p-FLAG vs. M8/RBM ORF1p-FLAG) to include the ISGs examined in this study. Notably, the interferome database and String analyses returned a larger pool of potential ISG proteins (**Fig. 3a**) and the studied ISG proteins are annotated in **Fig. 3b**. An updated GO analysis, using the cutoff of 0.5 \log_2 (abundance ratio of WT ORF1p-FLAG vs. M8/RBM ORF1p-FLAG) returned three immune-related terms: host-virus interaction, innate immunity, and antiviral defense (**Fig. 2d**).

To address the reviewer's suggestion, on page 9, we now state:

"To identify cellular proteins that differentially interact with the WT ORF1p-FLAG and M8/RBM-FLAG protein complexes, we conducted immunoprecipitation coupled with liquid chromatography-tandem mass spectrometry (IP/LC-MS/MS), followed by label-free quantification (LFQ) analyses (Fig. 2c).

We used the Database for Annotation, Visualization and Integrated Discovery (DAVID) to conduct gene ontology (GO) analyses using proteins that have $>0.5 \log_2$ abundance ratio in WT ORF1p-FLAG vs. M8/RBM-FLAG mutant IP/LC-MS/MS experiments (see Source Data). These analyses revealed an enrichment of viral-related GO terms, including “host-virus interaction,” “innate immunity,” and “antiviral defense” (Fig. 2d and Supplementary Table 1), associated with the WT ORF1p-FLAG vs. M8/RBM-FLAG protein complexes, suggesting a cohort of antiviral proteins preferentially associates with WT L1 RNPs.”

We also added the requested details in the Method section on pages 41-43.

We hope these additional experiments and explanations satisfy the reviewer’s concerns and thank them for raising these important points.

For these experiments the authors switch back to HeLa from U2OS - why? Can the rationale for cell type selection in different assays be laid out early in the manuscript? And then after this, IFN-alpha production is tested in HEK? I presume this is due to which IFN genes / pathways are active in which cell lines, but the reader needs to be informed about these choices and their rationale all along the way.

We thank the reviewer for the comments. In general, we used HeLa cells for most of the experiments. We switched to HEK293T cells for the IFN experiment due to the minimal interferon response caused by plasmid transfection in HEK293T (i.e., HEK293T previously were reported to have low cGAS levels in [Tunbak H. et al., Nat. Comm., 2020], as cited in the text). Notably, plasmid-induced interferon response can, in principle, mask the L1-mediated interferon response signal.

On page 10, we now state:

*“Because the overexpression of engineered L1s previously was reported to modestly induce type I IFN response, we next tested whether there was a difference in IFN- α induction in HEK293T cells transfected with WT and mutant L1 constructs. **Notably, HEK293T cells previously were reported to have a low amount of cyclic GMP-AMP synthase (cGAS, a DNA sensor), which can prevent a strong innate immune response by plasmid-based transfections and immunogenic DNAs⁴⁶.**”*

LINE 216 — “using a primer set that amplified the mneol retrotransposition reporter 1cassette” — since the L1 construct has its own cleavage and polyadenylation site (SV40 I think), then I understand that this primer set amplified only read-through transcripts? (Is this common with SV40? I thought not). Presumably this choice was made to avoid amplifying other abundant endogenous L1-containing transcripts. Can the authors please make the rationale for this choice and the described the expected results compared to the obtained results - making it clear to the reader. It may be possible to use primers where one hybridizes across the sequence of the FLAG-tag which is unique to this construct and the other is common to L1 RCs to avoid reading out on the reporter.

We thank the reviewer for the insightful comment. As the reviewer suggested, we choose the *mneol* primer set to avoid amplifying endogenous L1 transcripts. To clarify, the *mneol* primer set amplifies the retrotransposition reporter cassette in the 3’ UTR region of L1 RNA, not the read-through transcripts.

On page 10, we now state:

*“Controls revealed the L1 RNA levels of the RT-deficient and M8/RBM-FLAG mutants were similar to the WT L1 using a primer set that amplified the mneol retrotransposition reporter cassette, **thereby avoiding the amplification of endogenous L1 transcripts** (Fig. 2f).”*

LINE 240 — I think number of number of peptides (are we talking unique peptides?) is being used confusingly as a proxy for abundance. The authors should use the peptide-spectrum matches or spectral counts or something else here... a large protein that is low abundance could still provide more detectable peptides than a small protein that is higher abundance - I suggest the authors please clarify how they are using the peptides in greater details - for this a table in the text for the proteins of interest, extracting a few pieces of key info from the spreadsheets that are currently in the supplement would suffice. And again, I suggest some replicates and proper quant to make any kinds of claims about “fold-change.”

We thank the reviewer for this comment. We now listed the updated IP-MS triplicate data, which provides fold-change using MS1 intensities in **Fig. 2c** and the **Source data** file. We have also included descriptions of the IP-MS parameter results in the **Source data**.

On page 9, we now state:

*“To identify cellular proteins that differentially interact with the WT ORF1p-FLAG and M8/RBM-FLAG protein complexes, we conducted immunoprecipitation coupled with liquid chromatography-tandem mass spectrometry (IP/LC-MS/MS), followed by label-free quantification (LFQ) analyses (**Fig. 2c**). We used the Database for Annotation, Visualization and Integrated Discovery (DAVID) to conduct gene ontology (GO) analyses using proteins that have >0.5 log₂ abundance ratio in WT ORF1p-FLAG vs. M8/RBM-FLAG mutant IP/LC-MS/MS experiments (see **Source data**). These analyses revealed an enrichment of viral-related GO terms, including “host-virus interaction,” “innate immunity,” and “antiviral defense” (**Fig. 2d and Supplementary Table 1**), associated with the WT ORF1p-FLAG vs. M8/RBM-FLAG protein complexes, suggesting a cohort of antiviral proteins preferentially associates with WT L1 RNPs.”*

LINE 331 — helicase domain 1 or domain 2 is more important? I am reading that WA1 and WA2 are more severe and WB1 and WB2 are less severe in terms of inhibition of transpositions. Doesn't this make the more severe WA domain more important (yet helicase 2 is emphasized in text, either I misunderstood this section or it is a typo, I think)? Shouldn't there be 4 open triangles on Fig. 5a, indicating the positions of 4 defined amino acid substitutions? I see two triangles but 4 positions? Have I misunderstood?

We apologize that the phrasing in this section was confusing (as also noted by Reviewer 1). Both WA2 (a Walker A motif mutant in “helicase 2”) and WB2 (a Walker B motif mutant in “helicase 2”) HELZ2 mutants exhibited more marked losses in the inhibition of L1 retrotransposition than the WA1 (a Walker A motif mutant in “helicase 1”) and WB1 (a Walker B motif mutant in “helicase 1”) HELZ2 mutants, indicating that the “helicase 2” domain is more critical for inhibiting L1 retrotransposition.

As requested, we have clarified the text. The paragraph on page 15 now read as follows:

*“We mutated conserved amino acids in the Walker A and Walker B boxes thought to be required for ATP binding (WA1 [K550A] in the helicase 1 domain and WA2 [K2180A] in the helicase 2 domain) (**Fig. 5a**) or ATP hydrolysis (WB1 [E668A] in the helicase 1 domain and WB2 [E2361A] in the helicase 2 domain), respectively.”*

*“The WA1 mutant was able to inhibit L1 retrotransposition almost as effectively as WT HELZ2 in HEK293T (**Fig. 5b**), but not HeLa-JVM (**Fig. 5c**), cells. The WA2 and WA1&2 double mutants were significantly impaired in their ability to inhibit L1 retrotransposition in both HEK293T (**Fig. 5b**) and HeLa-JVM cells (**Fig. 5c**). A similar data trend was observed for the Walker B box mutations in HEK293T and HeLa-JVM cells (**Supplementary Figs. 5g and 5h, respectively**). In sum, mutations in the helicase 2 (WA2 and WB2) domains generally alleviated the HELZ2-mediated repression of L1 retrotransposition to a greater extent than mutations in the helicase 1 (WA1 and WB1) mutants, indicating the importance of the helicase 2 domain in the inhibition of L1 retrotransposition.”*

Also, as requested, we have moved **Figs. 5d and 5e** (WB1/2) to **Supplementary Fig. 5g and 5h**. The number of triangles should now reflect the number of mutations made (three mutations, three triangles).

Looking at the data - mutants in WA1/2 abrogate the HELZ2 effects in HeLa, but function fine in HEK, a not dissimilar result is seen in WB1/2... the results seem rather inconclusive to me. I am not certain that Fig. 5b-3 contribute to the manuscript and could be moved to the supplement.

We thank the reviewer for this comment. Although the HELZ2 WA2 mutant in HEK293T still inhibits L1 retrotransposition unlike in HeLa, a loss of L1 inhibition can be clearly seen between WA2 and the WT HELZ2 (**Fig. 5b**, WT ~10% vs. WA2 ~70% in comparison to the pCMV-3Tag-8-Barr control). This effect is similar in WT vs. WB2 in HEK293T (**Supplementary Fig. 5g**, WT ~10% vs. WB2 ~78% in comparison to the pCMV-3Tag-8-Barr control). The observed difference of the L1 retrotransposition inhibition by the HELZ2 mutants in HeLa and HEK293T cells may be due to a lower L1 retrotransposition inhibition by WT HELZ2 in HeLa cells compared to HEK293T cells (**Fig. 5b and 5c**, ~10% in HEK293T vs. ~50% in HeLa, in comparison to the pCMV-3Tag-8-Barr control).

However, the trend of HELZ2 mutant effects on L1 retrotransposition is consistent in both cell lines (i.e., WA1 > WA2 > WA1&2 in ascending order of a loss of L1 retrotransposition inhibition). As such, we modified the description of our results as noted above. In addition, we have moved **Fig. 5d and 5e** (WB1/2) to Supplementary **Fig. 5g and 5h** as suggested by the reviewer.

LINE 335 — the WA2 and WA1/2 appear hugely over expressed compared to WT and WA1, are these possible aggregating or otherwise not participating in the same molecular biology as the WT or WA1? I am surprised that WA1/2 does not show (at least) the same effect as WA1 - are we instead seeing an unexpected sign epistasis? considering the expression levels of WA2 and WA1/2 this seems difficult to interpret and I think goes a ways towards explaining the results in 5b-e and further suggests these experiments might be better in the supplements and claims in this section refined and pared down?

We appreciate the thoughtful question and suggestion. Although it is difficult to conclude whether the mutations completely change the biology of HELZ2 and if epistasis occurs, our results suggest that the HELZ2 WA2 and WA1&2 mutants act similarly to WT HELZ2 because (1) the mutated sites are consensus helicase sites that are known to impair the helicase activity without affecting the protein structure (Miller JM. and Enemark EJ., *Archaea*, 2016); and (2) immunofluorescence of the HELZ2 WA1&2 mutant exhibits a similar uniform cytoplasmic distributions in comparison to WT HELZ2 and we did not observe any aggregated HELZ2 signals in the WA1&2 mutant (**Fig. 4d**). We respectfully believe that our results reflect a loss in the HELZ2 helicase activity rather than a total change in HELZ2 behavior. As such, we think it is better to keep the WA1&2 mutant results in the main figure. We have moved the WB1/2 mutant results to the supplementary materials as suggested (see above).

LINE 410 — (Fig. 6f) I believe this needs a control that is “no L1” (pCEP) + HELZ2 — does over-expression of HELZ drive IFN down regardless of L1 status?

We agree with the reviewer suggestion and thank them for the comment. We added the requested pCEP4 + HELZ2 control (**Fig. 6f**). The overexpression of HELZ2 reduced IFN levels regardless of L1 status, suggesting that HELZ2 might degrade other immunogenic RNAs or reduces the IFN response through other mechanisms.

On pages 18-19, we now state:

*“Notably, this level of IFN- α induction was even lower than that observed in cells transfected with only the pCEP4 empty vector (Fig. 6f). **Co-expression of HELZ2 with the pCEP4 empty vector also reduced the IFN- α level when compared to the pCEP4 only empty vector (Fig. 6f), raising the possibility that HELZ2 overexpression may also reduce the stability of endogenous immunogenic RNAs, thereby reducing basal levels of IFN- α induction.**”*

Discussion

Is the HELZ2 exonuclease activity unambiguously shown? Did I miss the reference? If this has been shown - great, reference it (and my apologies if I missed that). If it has not been shown, I am not certain it is wise to presume on the basis of sequence / structure homology. I get that the word putative was used - and that is fine when discussing homologs, broadly, but in the discussion exonuclease activity is put forward as a mechanism to degrade L1. This raises the bar. Furthermore, HELZ2 is shown to rely on binding the 5' UTR, but its putative activity is 3'→5' so, this does not quite add up. Bottom line, 3'-5' exoribonucleases need access to free 3'-ends, and typically PABPC1/4 have to be cleared - I guess in a mechanism using CCR4-NOT and/or PAN2/3 - then by the RNA exosome... anyway, my point is that the authors may be making a few too many assumptions here. Since they have the Myc-tagged HELZ2 (and mutants thereof) they could easily test in vitro RNase activity (including against in vitro transcribed L1 RNAs +/- 5' UTR) - these don't have to be sophisticated assays, but POC that shows there is the expected activity – this will go a ways towards bridging some of the assumptions of the model.

We thank the reviewer for raising this concern, as the HELZ2 exonuclease activity had not yet been demonstrated biochemically. As per the reviewer's request, we used an *in vitro* assay to demonstrate that HELZ2 likely contains a 3' - 5' exonuclease activity and that this exonuclease

activity is lost in the dRNase mutant (**Supplementary Fig. 5d**). We cannot formally exclude the possibility that HELZ2 pairs with another RNase to degrade L1 RNA *in vivo*.

Notably, it remains to be seen if the HELZ2 helicase activity is able to “clear” proteins bound to the L1 RNA. We now focus our discussion on an RNA secondary structure that some RNB-containing proteins have been shown to unwind (Chu LY. et al., Structural insights into RNA unwinding and degradation by RNase R, *Nucleic Acids Res*, 2017). Thus, we respectfully submit that testing the stated hypotheses is beyond the scope of our current manuscript and is more suitable for future studies.

To address the reviewer’s comments, on page 15, we now state:

*“To examine whether the HELZ2 RNB domain has exoribonuclease activity, we purified the WT HELZ2-3xFLAG and dRNase HELZ2-3xFLAG mutant proteins from HEK293T cells (**Supplementary Fig. 5c**) and performed a ribonuclease assay using a poly(A)₃₀ RNA oligonucleotide labeled with IRDye800 at its 5’ end as an RNA substrate. The WT HELZ2-3xFLAG protein, but not the dRNase HELZ2-3xFLAG mutant protein, degraded the single-strand RNA substrate in a 3’ to 5’ direction (**Supplementary Fig. 5d**). However, the dRNase mutant generally only had minor effects (i.e., less than 2-fold) on L1 retrotransposition efficiency in HeLa-JVM and HEK293T cells when compared to the WT HELZ2 control (**Supplementary Figs. 5e and 5f, see discussion**).”*

On pages 20-21, we also added the following information:

*“Thus, it is tempting to suggest that HELZ2 might function in a similar stepwise manner, where its helicase activity initially unwinds L1 RNA secondary structures, allowing the subsequent degradation of L1 RNA by the HELZ2 3’ to 5’ exoribonuclease activity (**Fig. 7 and Supplementary Fig. 5d**)”*

Other comments

* Why ignore the significant GO term related to NMD that also would affect the level of L1 RNA? I guess this was simply to focus the paper on the viral defenses topic? Can the NMD nodes be highlighted in Fig. 3C so that we can see how the compositions of these enriched terms intersect / interact?

We thank the reviewer for the comments. The updated GO terms show a slightly different result than that reported in our original manuscript. We have mentioned other GO terms, including NMD, in the main text. We did not highlight the NMD node, as the STRING analyses were performed using the ISGs identified from the interferome database (**Fig. 3c in the previous manuscript**).

To clarify these points, on pages 9-10, we now state:

“We also observed an enrichment in the following GO terms: “nonsense-mediated mRNA decay” and “RNA-mediated gene silencing.” Proteins within these pathways, such as UPF1 and let-7 miRNA, respectively, previously were implicated in the regulation of L1 retrotransposition.”

* Fig 5g - please validate siRNA KDs by western blotting not RT-qPCR (it is not that I don’t believe the result, it is that the readout of a KD is the protein level, not the RNA level).

We acknowledge the suggestion. Notably, we tested two different HELZ2 antibodies (Abcam [AB129781] and Affinity Biosciences [DF4285]), but the antibodies were not able to detect the endogenous HELZ2 protein. Moreover, the Affinity Biosciences [DF4285] antibody does not detect the ectopically overexpressed HELZ2 protein in our experiments. Given these data, we believe that the RT-qPCR result is best kept in the main figure.

* If ORF2p is expression w/o the 5’-UTR then HELZ overexpression has no effect on Alu?

We thank the reviewer for the comment. We have tested co-overexpressing HELZ2 with an engineered Alu expression construct and observed a small reduction in Alu RNA (~40%), although this reduction is not as significant as the reduction in L1 RNA (~80%) (**Supplementary Fig. 6b**). Thus, it remains possible that HELZ2 affects Alu RNA levels to a lesser extent than L1 RNA levels. Notably, we did find that HELZ2 does not affect the retrotransposition efficiency of an L1 lacking the 5’UTR (**Supplementary Fig. 6c**).

On page 17, we now state:

*“That being stated, the co-expression of an Alu only expression plasmid (Alu_ neo^{Tet}) and HELZ2 in HeLa-HA cells still exhibited a ~40% reduction in Alu RNA levels, although this reduction was not as significant as observed with L1 RNA (**Supplementary Fig. 6b vs. Fig. 6d**)”*

In addition, on page 18, we also added the following sentence:

*“Consistently, HELZ2 overexpression did not significantly affect L1 retrotransposition efficiency in the L1 [Δ 5'UTR] construct (**Supplementary Fig. 6c**)”*

* how much acetone was used to precipitate proteins (not defined in the method)

We have now added the details to the Methods section.

On page 40, we now state:

*“This step was repeated once, and the protein was precipitated overnight by adding **three times the volume of cold acetone to the resultant eluate.**”*

* can you please us units for your % - BSA is no doubt at 0.5% w/v and Triton X-100 is at 1% v/v - but this should be explicitly defined as a matter of routine any time % by weight or volume is used.

We have added the units as requested throughout the Methods section.

* not the author's fault but neither www.interferome.org nor the URL given in the reference (which is different) seem to load.

The website was down for a while, but now it is up and accessible.

Reviewer #3 (Remarks to the Author):

The article “The interferon stimulated gene-encoded protein HELZ2 inhibits human LINE-1 retrotransposition and LINE-1 RNA-Mediated type I interferon induction” by Miyoshi and colleagues is an interesting study of proteins induced by the interferon response to LINE-1 RNA that can inhibit retrotransposition of LINE-1. This is an exciting and thorough study that gives insight into control of LINE-1 expression in the cell and links it with type I interferon signaling. In general the experiments are well controlled and the conclusions are well supported by the data presented. I have the following questions:

We thank the reviewer for their supportive and scholarly comments.

Major points

1. In Figure 2, the authors assess “IFN- α ” RNA by qRT-PCR. They should be specific about which IFN- α as there are many and different ones are expressed by different cell types. They should also assess levels of interferon beta and interferon lambda (type III IFN) as both have previously been shown to be activated by transposable element RNA.

We thank the reviewer for the insightful suggestions. The primer set that we used amplifies IFN- α 1 and IFN- α 13. We have indicated this point in the legends to **Fig. 2** and **Fig. 6** by adding the following text:

*“... (primer set: IFN- α , **which amplifies IFN- α 1 and IFN- α 13) ...”***

Moreover, we conducted a multiplex cytokine/chemokine assay (Bio-plex) to further examine the effects of L1 expression on the production of other cytokines/chemokines. We observed a modest increase in other IFNs, such as IFN- α 2, IFN beta, IFN gamma, IFN lambda (IL-28A and IL-29), in the M8/RBM L1-expressing culture media in comparison to the mock control (pCEP4) (**Supplementary Table 3**). As a positive control for this experiment, we included a poly(I:C) transfection control; this control exhibited higher levels of IFN beta and IL-28A expression in culture media when compared to culture media from cells transfected with the L1-expressing plasmids.

On pages 10-11, we state:

*“Finally, we conducted a Bio-Plex assay that allows the simultaneous assessment of 37 different cytokines and chemokines (see **Methods**). These analyses revealed that the M8/RBM mutant L1, in particular, exhibited a modest, but overall increase in secreted cytokines and chemokines, as well as other IFNs, when compared to a mock control. As a control, we also included polyinosinic:polycytidylic acid (poly[I:C]) in this assay, which is a double-stranded RNA analog known to strongly induce the innate immune response⁴⁶. We found that several cytokines exhibited a comparable level of upregulation in the poly(I:C) and M8/RBM transfected cells (**Supplementary Table 3**). WT or RT-deficient L1 transfected cells exhibited an increase in the secretion of several cytokines, including IFN- β , IL-27, and MMP-3, when compared to the controls; however, those levels were generally lower than those in the L1 M8/RBM-transfected cells. These results are consistent with the IFN- α RT-qPCR results (**Fig. 2f**), which demonstrated the M8/RBM L1-transfected cells induced a higher IFN- α expression when compared to either the WT or RT-deficient L1 transfected cells (**Supplementary Table 3**)”*

2. The decrease in Alu retrotransposition (in Figure 6) when HELZ2 or MOV10 is overexpressed is interesting but the authors should put it in context. Are these proteins associating with Alu RNA as well?

We thank the reviewer for the comment. As noted above in response to a similar comment made by Reviewer 2, we co-expressed HELZ2 with an engineered Alu expression construct and observed a reduction in Alu RNA (~40%), although this reduction is not as significant as the reduction in L1 RNA (~80%) (**Supplementary Fig. 6b**).

On page 17, we state:

*“That being stated, the co-expression of an Alu only expression plasmid (Alu_ neo^{Tet}) and HELZ2 in HeLa-HA cells still exhibited a ~40% reduction in Alu RNA levels, although this reduction was not as significant as observed with L1 RNA (**Supplementary Fig. 6b vs. Fig. 6d**).”*

It remains possible that HELZ2 directly effects Alu RNA levels, but the effect appears less than the effect on L1 RNA. That being stated, we think it is more likely that the HELZ2-mediated reduction in Alu retrotransposition efficiency is mainly due to the reduction of L1 RNA, which in turn leads to less ORF2p.

In regard to MOV10, previous literature has shown similarly that MOV10 severely reduced Alu retrotransposition efficiency (Goodier JL. et al., MOV10 RNA Helicase Is a Potent Inhibitor of Retrotransposition in Cells, PLoS Genet., 2012; Arjan-Odedra S. et al., Endogenous MOV10 inhibits the retrotransposition of endogenous retroelements but not the replication of exogenous retroviruses, Retrovirology, 2012), although the mechanism remains unclear. Notably, we mainly used MOV10 as a control in our retrotransposition assays. We believe that examining the effect of MOV10 on Alu retrotransposition is beyond the scope of our study, but do agree the topic is worthy of follow up experimentation.

3. Besides IFN alpha RNA (qRT-PCR), the authors should assess secreted interferon alpha/beta and a panel of downstream interferon stimulated genes to ensure their L1 expression causes a bona fide interferon response. To link the type I IFNs with the ISGs (their model in Figure 7) they could also perform antibody blocking experiments of the IFNAR1 receptor to prove that type I interferon activation of ISG proteins is what limits L1 retrotransposition.

We thank the reviewer for the suggestion. As detailed in our response to Reviewer 3 Point #1, we have run a cytokine assay (Bio-plex) to check for secreted cytokines/chemokines and observed a general increase cytokines and chemokines in the M8/RBM sample when compared to the mock control (pCEP4) (**Supplementary Table 3**).

On pages 10-11 of the Results, we now state:

*“Finally, we conducted a Bio-Plex assay that allows the simultaneous assessment of 37 different cytokines and chemokines (see **Methods**). These analyses revealed that the M8/RBM mutant L1, in particular, exhibited a modest, but overall increase in secreted cytokines and chemokines, as well as other IFNs, when compared to a mock control. As a control, we also included polyinosinic:polycytidylic acid (poly[I:C]) in this assay, which is a double-stranded RNA analog known to strongly induce the innate immune response. We found that several cytokines exhibited a*

comparable level of upregulation in the poly(I:C) and M8/RBM transfected cells (**Supplementary Table 3**). WT or RT-deficient L1 transfected cells exhibited an increase in the secretion of several cytokines, including IFN- β , IL-27, and MMP-3, when compared to the controls; however, those levels were generally lower than those in the L1 M8/RBM-transfected cells. These results are consistent with the IFN- α RT-qPCR results (**Fig. 2f**), which demonstrated the M8/RBM L1-transfected cells induced a higher IFN- α expression when compared to either the WT or RT-deficient L1 transfected cells (**Supplementary Table 3**)."

We report the results of our experiments in the Results section and have elaborated on the interpretation of these experiments in the Discussion. Since our RT-qPCR (some ISGs were modestly upregulated but the others were not) and antibody blocking experiments did not clearly show the link between type I IFNs increase with the ISG levels, we put a question mark besides the arrow pointing towards ISGs from the IFN receptor in **Fig. 7**. However, our cytokine assay and previous literature (*i.e.*, Tunbak H. et al., The HUSH complex is a gatekeeper of type I interferon through epigenetic regulation of LINE-1s, Nat. Comm., 2020 and Zhao K. et al., LINE1 contributes to autoimmunity through both RIG-I- and MDA5-mediated RNA sensing pathways, Journal of Autoimmunity, 2018) suggest that L1 RNA causes a *bona fide* interferon response. Future studies in a more suitable cell line (such as a human monocytic cell line) could elucidate the effects of overexpressed L1 RNA towards ISG expression.

On page 22 of the Discussion, we state:

"Because we observed a modest increase in some cytokines in M8/RBM mutant-transfected cells (**Supplementary Table 3**), our data are in general agreement with other reports, which demonstrated that higher L1 RNA expression levels can lead to the upregulation of the type I IFN response."

We hope the additional experiment and explanation satisfy the reviewer's concern and, once again, thank them for raising this important point.

Minor points

1. The authors should explain why they used arsenite (line 161) when studying stress granules

We thank the reviewer for the suggestion.

On page 8, we now state:

"...; however, we did not observe the formation of mCherry-G3BP1 foci. Thus, we next treated the U-2 OS cells with sodium arsenite, which strongly induces stress granule formation. Sodium arsenite treatment resulted in both an increase in size of the ORF1p cytoplasmic foci and co-localization of ORF1p with mCherry-G3BP1 foci (Supplementary Fig. 3c)."

2. Please show (in the supplement) the original flow plots for the quantified EGFP+ cell data (*i.e.* Figure 5e) for reference

We thank the reviewer for the suggestion and have included representative flow cytometry plots of HeLa-JVM and HEK293T experiments in **Supplementary Fig. 4a**. All the plots were obtained similarly and include the live cell gating, no EGFP control, L1 EGFP+ cells, and the transfection control (intronless EGFP reporter).

REVIEWERS' COMMENTS

Reviewer #1 (Remarks to the Author):

The authors have answered all of my queries thank you.

Reviewer #2 (Remarks to the Author):

I am generally satisfied that the Author's have addressed the majority of my concerns, or otherwise justified their alternative choices that may run contrary to my suggestions. I believe this manuscript is much improved and I recommend publication.

I have a few remaining comments - the editor should decide if these are important or not.

* The fact that the ORF2p level INCREASES in the M8 mutant is interesting - L1 RNA quant probably should be normalized to the ORF2p recovery in the IP. I realize you don't intend to include that in this manuscript. This is just a comment for the authors only.

* In Fig 2D: "Proteins having a >0.5 log₂ abundance ratio in the WT ORF1p vs. M8 [RBM]) complexes were subjected to DAVID gene ontology analysis." is this regardless of the p-value? As long as they were enriched by 1.4x or more they were included? Might add "at any p-value" if this is so.

* The authors point out antibodies that DID NOT work in their hands - it would be amazing to report that they did this in the supplement - readers want to know antibodies that may not work as much as they need to know ones that work. I would like to see this practice normalized, so I am suggesting it. Obviously it is not scientifically necessary.

* antibody dilutions are given as fractions - whenever possible, can they be given in ug / ml (units of concentration) - at least for commercial antibodies of known concentration? This kind of reporting makes experiments more reproducible. Again, I would like to see this practice normalized, so I am suggesting it.

Reviewer #3 (Remarks to the Author):

The authors have answered my queries with additional experiments and the manuscript is significantly improved. I have no further questions.

Reponses to the referees' comments of Nature Communications manuscript NCOMMS-22-10777A entitled, "The interferon stimulated gene-encoded protein HELZ2 inhibits human LINE-1 retrotransposition and LINE-1 RNA-mediated type I interferon induction," by Luqman-Fatah *et al.*

We thank again the editors and reviewers for their time and scholarly comments. Please find our point-by-point responses to the reviewers' comments below. The original comments are noted in black text. Our responses are noted in blue text.

Reviewer #1 (Remarks to the Author):

The authors have answered all of my queries thank you.

We again thank the reviewer for their time and comments.

Reviewer #2 (Remarks to the Author):

I am generally satisfied that the Author's have addressed the majority of my concerns, or otherwise justified their alternative choices that may run contrary to my suggestions. I believe this manuscript is much improved and I recommend publication.

We again thank the reviewer for their scholarly review and comments

I have a few remaining comments - the editor should decide if these are important or not.
* The fact that the ORF2p level INCREASES in the M8 mutant is interesting - L1 RNA quant probably should be normalized to the ORF2p recovery in the IP. I realize you don't intend to include that in this manuscript. This is just a comment for the authors only.

We appreciate the reviewer's insightful comment. Although we did not include the RNA-IP data with the ORF2p pull-down experiment, which is beyond the scope of this study, the precipitated RNA amounts will be normalized to ORF2p recovered after IP in our future study to conduct more functional validation of this data.

* In Fig 2D: "Proteins having a $>0.5 \log_2$ abundance ratio in the WT ORF1p vs. M8 [RBM]) complexes were subjected to DAVID gene ontology analysis." is this regardless of the p-value? As long as they were enriched by 1.4x or more they were included? Might add "at any p-value" if this is so.

We thank the reviewer for the insightful comments and have modified the text.

On page 68 in the figure legends, we now state:

"*Proteins having a \$>0.5 \log_2\$ abundance ratio at any p-value in the WT ORF1p vs. M8 [RBM]) complexes were subjected to DAVID gene ontology analysis.*"

* The authors point out antibodies that DID NOT work in their hands - it would be amazing to report that they did this in the supplement - readers want to know antibodies that may not work as much as they need to know ones that work. I would like to see this practice normalized, so I am suggesting it. Obviously it is not scientifically necessary.

We thank the reviewer for this suggestion.

On page 33 in the Methods section, we state:

"*Please note: We tested two different anti-HELZ2 antibodies (Abcam [AB129781] and Affinity Biosciences [DF4285]), but the antibodies were not able to detect the endogenous HELZ2 protein in our experimental conditions.*"

* antibody dilutions are given as fractions - whenever possible, can they be given in $\mu\text{g} / \text{ml}$ (units of concentration) - at least for commercial antibodies of known concentration? This kind of reporting

makes experiments more reproducible. Again, I would like to see this practice normalized, so I am suggesting it.

We thank the reviewer for this suggestion and have added the concentration of commercial antibodies used in our method section when the information is provided by the manufacturer.

Reviewer #3 (Remarks to the Author):

The authors have answered my queries with additional experiments and the manuscript is significantly improved. I have no further questions.

We again thank the reviewer for their time and comments.